# A Perturbation Approach to Unconstrained Linear Bandits

**Andrew Jacobsen** [* 1 2] **Dorian Baudry** [* 3] **Shinji Ito** [4] **Nicolò Cesa-Bianchi** [1 2]

## Abstract

We revisit the standard perturbation-based approach of Abernethy et al. (2008) in the context of unconstrained Bandit Linear Optimization (uBLO). We show the surprising result that in the unconstrained setting, this approach effectively reduces Bandit Linear Optimization (BLO) to a standard Online Linear Optimization (OLO) problem. Our framework improves on prior work in several ways. First, we derive expected-regret guarantees when our perturbation scheme is combined with comparator-adaptive OLO algorithms, leading to new insights about the impact of different adversarial models on the resulting comparator-adaptive rates. We also extend our analysis to dynamic regret, obtaining the first guarantees with optimal $\sqrt{P_T}$ path-length dependencies without prior knowledge of $P_T$. We then develop the first high-probability guarantees for both static and dynamic regret in uBLO. Finally, we discuss lower bounds on the static regret, and prove the folklore $\Omega(\sqrt{dT})$ rate for adversarial linear bandits on the Euclidean ball, which is of independent interest.

## 1. Introduction

Online convex optimization (OCO) provides a general framework for sequential decision-making under uncertainty, in which a learner repeatedly selects an action from a set $\mathcal{W} \subseteq \mathbb{R}^d$ and receives feedback generated by an adversarial environment (Shalev-Shwartz, 2012; Hazan et al., 2016; Orabona, 2019). The standard measure of performance is *regret*, which compares the learner's cumulative loss to that of some unknown benchmark strategy. The most

*Equal contribution [1]Università degli Studi di Milano [2]Politecnico di Milano [3]Inria, Univ. Grenoble Alpes, Grenoble INP, CNRS, LIG, 38000 Grenoble, France [4]The University of Tokyo and RIKEN. Correspondence to: Andrew Jacobsen <contact@andrew-jacobsen.com>, Dorian Baudry <dorian.baudry@inria.fr>.

*Proceedings of the 43$^{rd}$ International Conference on Machine Learning*, Seoul, South Korea. PMLR 306, 2026. Copyright 2026 by the author(s).

general formulation is *dynamic regret*, defined by

$$R_T(u_{1:T}) = \sum_{t=1}^{T} f_t(w_t) - \sum_{t=1}^{T} f_t(u_t),$$

where $f_t : \mathcal{W} \to \mathbb{R}$ denotes a $G$-Lipschitz convex loss function, each play $w_t \in \mathcal{W}$ of the learner is based solely on its past observations, and $u_{1:T} = (u_t)_{t \in [T]}$ is a *comparator sequence* in $\mathcal{W}$. The classical *static* regret is recovered as the special case $u_1 = \cdots = u_T$.

In this paper we focus on *Bandit Linear Optimization* (BLO) (Flaxman et al., 2005; Kalai & Vempala, 2005; Dani et al., 2007; Lattimore & Szepesvári, 2020; Lattimore, 2024), where $f_t$ is a *linear* function and the learner observes only the scalar output $f_t(w_t) = \langle \ell_t, w_t \rangle$ for some vector $\ell_t \in \mathbb{R}^d$ with $\|\ell_t\| \le G$, rather than the full gradient $\nabla f_t(w_t) = \ell_t$. We propose a modular reduction that enables the use of an arbitrary OLO learner under bandit feedback by feeding it suitably perturbed loss estimates. We focus in particular on *unconstrained* bandit linear optimization (uBLO), in which the action set is $\mathcal{W} = \mathbb{R}^d$ (van der Hoeven et al., 2020; Luo et al., 2022; Rumi et al., 2026). A central objective in uBLO is to obtain *comparator-adaptive* guarantees, in which the regret against a static comparator $u$ scales with $\|u\|$, while simultaneously enforcing a *risk-control* constraint of the form $R_T(0) \le \epsilon$, where $\epsilon > 0$ is a fixed, user-specified parameter.

This formulation is closely connected to parameter-free online learning and coin-betting (Mcmahan & Streeter, 2012; Orabona & Pál, 2016; Cutkosky & Orabona, 2018): controlling $R_T(0)$ can be interpreted as allowing the bettor to *reinvest* a data-dependent fraction of an initial budget and its accumulated gains, rather than committing to a fixed betting scale (Orabona, 2019, Chapter 5.3). This viewpoint highlights how uBLO complements the more standard linear bandit setting with a bounded action set. In the bounded case, the learner effectively plays with a fixed "unit budget" each round (since $\|w_t\|$ is uniformly bounded), whereas in the unconstrained case the learner may increase its effective scale over time, but only if its total gains permit it. In applications where exploration must be performed under budget constraints, this built-in risk control can make the unconstrained model (perhaps counterintuitively) the more natural abstraction.

In this work we also broaden the standard notion of comparator-adaptivity by allowing the adversary to choose the comparator norm more strategically, e.g., at the end of the interaction, refining what "adaptive" guarantees entail.

**Notation.** We denote by $\mathcal{F}_{t-1}$ the $\sigma$-field generated by the history up to the start of round $t$, and we define $\mathbb{E}_t[\cdot] = \mathbb{E}[\cdot|\mathcal{F}_{t-1}]$. We say a sequence of random variables $(X_t)_{t=1}^T$ is *adapted* to $(\mathcal{F}_t)_{t=0}^T$ if $X_t$ is $\mathcal{F}_t$ measurable for all $t$. For any $A, B \in \mathbb{R}$, we denote $A \wedge B = \min\{A, B\}$, $A \vee B = \max\{A, B\}$, and $(A)_+ = A \vee 0$. Then, for (multivariate) functions $f$ and $g$, we use $f = \mathcal{O}(g)$ (resp. $f = \Omega(g)$) when there exists a constant $c > 0$ s.t. $f \leq cg$ (resp. $\geq$). We also use $\widetilde{\mathcal{O}}$ and $\widetilde{\Omega}$ respectively to further hide polylogarithmic terms (though we will occasionally still highlight $\log(\|u\|)$ dependencies when relevant). Unless stated otherwise $\log$ is the natural logarithm and we denote $\log_+(x) = \log(x) \vee 0$. A matrix $A \in \mathbb{R}^{d \times d}$ is *positive definite* if it is symmetric and satisfies $\langle x, Ax \rangle > 0$ for any $x \in \mathbb{R}^d$.

## 1.1. Related Works

Bandit Linear Optimization (BLO), also known as *adversarial linear bandits*, has a long history (McMahan & Blum, 2004; Awerbuch & Kleinberg, 2004; Dani & Hayes, 2006; Dani et al., 2007; Abernethy et al., 2008; Bubeck et al., 2012). In these works, the action set $\mathcal{W}$ is typically *constrained* to a bounded set, and minimax-optimal guarantees on the *expected regret* are known to be of order $d\sqrt{T}$ or $\sqrt{dT}$ depending on the geometry of the decision set (Dani et al., 2007; Shamir, 2015). More recently, these results have also been extended to nearly-matching *high-probability* guarantees (Lee et al., 2020; Zimmert & Lattimore, 2022).

Our work is most closely related to the works of van der Hoeven et al. (2020); Luo et al. (2022); Rumi et al. (2026), which investigate linear bandit problems with *unconstrained* action sets (uBLO). van der Hoeven et al. (2020) provided the first approach for this setting, using a variant of the scale/direction decomposition from the parameter-free online learning literature (Cutkosky & Orabona, 2018). As remarked by Luo et al. (2022), using their approach with a direction learner admitting $\widetilde{\mathcal{O}}(\sqrt{dT})$ regret on the unit ball (Bubeck et al., 2012), one can obtain a $\widetilde{\mathcal{O}}(\|u\|\sqrt{(d \vee \log(\|u\|))T})$ static regret bound. Our work provides new insights into these results by highlighting a subtle dependency issue between the loss sequence and the comparator norm, addressed by our approach.

Later, Rumi et al. (2026) investigated *dynamic* regret in the uBLO setting, and achieved the first guarantees in the (oblivious) adversarial setting that adapt to the number of switches of the comparator sequence, $S_T = \sum_t \mathbb{I}\{u_t \neq u_{t-1}\}$, while guaranteeing a $\sqrt{S_T}$ dependence without prior knowledge of the comparator sequence. Besides this work, all existing works on dynamic regret under bandit feedback fail

to obtain the optimal $\sqrt{S_T}$ dependencies without leveraging prior knowledge of the comparator sequence (Agarwal et al., 2017; Marinov & Zimmert, 2021; Luo et al., 2022), and in fact Marinov & Zimmert (2021) show that $\sqrt{S_T}$ dependencies are impossible against an adaptive adversary in many constrained settings. [1]

Our work is also related to the recent line of work in online convex optimization on comparator-adaptive (sometimes called *parameter-free*) methods. These are algorithms which, for any fixed $\epsilon > 0$, achieve guarantees of the form

$$R_T(u) = \widetilde{\mathcal{O}}\left(\epsilon + \|u\|\sqrt{T \log\left(\frac{\|u\|\sqrt{T}}{\epsilon} + 1\right)}\right), \quad (1)$$

uniformly over all $u \in \mathbb{R}^d$ simultaneously, matching the bound that gradient descent would obtain with oracle tuning (up to logarithmic factors) (Mcmahan & Streeter, 2012; McMahan & Orabona, 2014; Orabona & Pál, 2016; Cutkosky & Orabona, 2018; Orabona & Pál, 2021; Mhammedi & Koolen, 2020; Zhang & Cutkosky, 2022). The key feature of Equation (1) is that the bound is *adaptive* to an arbitrary comparator norm, rather than the worst-case $D = \sup_{x,y \in \mathcal{W}} \|x - y\|$, making these methods crucial for unconstrained settings. Comparator-adaptive methods have also recently been extended to dynamic regret (Jacobsen & Cutkosky, 2022; Zhang et al., 2023; Jacobsen & Cutkosky, 2023; Jacobsen & Orabona, 2024; Jacobsen et al., 2025), with guarantees that adapt to the *path-length* $P_T := \sum_{t=2}^T \|u_t - u_{t-1}\|$ and *effective diameter* $M = \max_t \|u_t\|$ in unbounded domains

$$R_T(u_{1:T}) \leq \widetilde{O}\left(\sqrt{(M^2 + MP_T)T}\right). \quad (2)$$

In this work we use a less common generalization of this bound due to Jacobsen & Cutkosky (2022), which adapts to the path-length and *each* of the individual comparator norms to achieve

$$R_T(u_{1:T}) \leq \widetilde{O}\left(\sqrt{(\|u_T\| + P_T)\sum_{t=1}^T \|\ell_t\|^2 \|u_t\|}\right). \quad (3)$$

Our results are the first of this form in the bandit setting.

## 1.2. Contributions

In this paper we revisit the classic perturbation-based approach of (Abernethy et al., 2008) in the setting of unconstrained linear bandits (Section 2), under the name `PABLO` (*Perturbation Approach for Bandit Linear Optimization*). We show that this approach produces loss estimators with strong properties, enabling us to develop several novel results in the uBLO setting.

**Adaptive comparators in expected-regret bounds.** We propose a novel algorithm with expected-regret guarantees

---

[1]In particular, their lower bound holds for finite policy classes.

for both static regret (Section 3.1) and for dynamic regret (Section 3.2). A key novelty is that our bounds remain valid in an adversarial regime where the comparator may be *data-adaptive*. This exposes an oblivious comparator assumption that is often left implicit in prior work, which we discuss in Section 3.1. Notably, distinguishing between oblivious and data-adaptive comparator settings induces a $\sqrt{d}$ separation in the dimension dependence of our bounds, while preserving the same $\sqrt{T}$ scaling in the horizon. We show that this contrasts with direct adaptations of prior approaches, which can incur a worse dependence on $T$ in the adaptive comparator regime. We leave as an open question to determine if this gap is unavoidable.

$\sqrt{P_T}$-**adaptive dynamic regret.** By relying on the PABLO framework, we develop the first algorithm for uBLO with an expected dynamic-regret guarantee exhibiting the optimal $\sqrt{P_T}$ dependence without prior knowledge of $P_T$. This contrasts with Rumi et al. (2026), who derive a comparable guarantee only for the weaker switching measure $S_T := \sum_t \mathbb{I}\{u_t \neq u_{t-1}\}$. Moreover, as detailed in Section 3.2, our analysis yields an even stronger bound, inspired by recent advances in OCO (see Eq. (3)).

**High-probability bounds.** In Section 4, we develop the first high-probability bounds for static and dynamic regret in uBLO. Our static regret guarantee scales as $R_T(u) \leq \widetilde{\mathcal{O}}(\|u\|\sqrt{dT\log(1/\delta)})$, matching the best known rates from the bounded domain setting (Lee et al., 2020; Zimmert & Lattimore, 2022). Our dynamic regret bound generalizes this result, and leads to $R_T(u_{1:T}) \leq \widetilde{\mathcal{O}}(\sqrt{d(M^2 + MP_T)T} + M\sqrt{dT\log(T/\delta)})$, where $M = \max_t \|u_t\|$, again without prior knowledge of $P_T$ or $M$.

**Open discussion on lower bounds.** In Section 5 we discuss the largely open problem of proving regret lower bounds for uBLO, with a focus on static regret. We establish intermediate results that motivate the conjecture that, when the comparator norm is non-adaptive, the minimax lower bound scales as $\Omega(\|u\|\sqrt{T(d \vee \log\|u\|)})$, and we briefly comment on the more challenging norm-adaptive regime.

As part of this investigation, we also provide a self-contained proof (Theorem 5.2) that the $\widetilde{\mathcal{O}}(\sqrt{dT})$ static regret achievable on the unit Euclidean ball (see, e.g., Bubeck et al. (2012)) is minimax-optimal. This complements existing characterizations of dimension-dependent minimax rates for adversarial linear bandits on bounded action sets (Dani et al., 2007; Shamir, 2015).

## 2. Perturbation-based approach

In this section we introduce a simple reduction that turns any algorithm for *Online Linear Optimization* (OLO) into an algorithm for *Bandit Linear Optimization* (BLO), inspired by the SCRiBLe algorithm of (Abernethy et al., 2008;

---

**Algorithm 1** PABLO

**Input:** OLO algorithm $\mathcal{A}$
**for** $t = 1$ **to** $T$ **do**
    Get $w_t \in \mathcal{W}$ from $\mathcal{A}$
    Choose a positive definite matrix $H_t \in \mathbb{R}^{d \times d}$ and let $v_1, \ldots, v_d$ be its eigenvectors
    Sample $s_t$ uniformly from $\mathcal{S} = \{\pm v_i : i \in [d]\}$
    Play $\widetilde{w}_t = w_t + H_t^{-1/2} s_t$, observe $\langle \ell_t, \widetilde{w}_t \rangle$
    Send loss estimate $\widetilde{\ell}_t = d\,H_t^{1/2} s_t \langle \widetilde{w}_t, \ell_t \rangle$ to $\mathcal{A}$
**end for**

---

2012). We will refer to this approach as PABLO, short for *a Perturbed Approach to Bandit Linear Optimization*. The pseudo-code of PABLO can be found in Algorithm 1. On each round $t$, the algorithm operates in two steps. First, an OLO learner $\mathcal{A}$ outputs a decision $w_t \in \mathcal{W}$ based on past feedback. Then the algorithm applies a randomized perturbation to $w_t$, observes the bandit feedback, and constructs an unbiased estimator $\widetilde{\ell}_t$ of the loss $\ell_t$, which is then passed back to $\mathcal{A}$.

This is a modest generalization of SCRiBLe, decoupling the OLO update from the perturbation mechanism. In the original SCRiBLe algorithm, both components are tied to a single self-concordant barrier $\psi$: the OLO step is implemented via FTRL with regularizer $\psi$, and the perturbation is scaled using the local geometry induced by $\nabla^2 \psi(w_t)$, which is replaced by a matrix $H_t^{\frac{1}{2}}$ in PABLO. As we show in Section 3, this coupling is not strictly necessary in the unconstrained setting, and it can be advantageous to tune the OLO algorithm and the perturbation level separately.

**Properties.** We now introduce some general properties on the loss estimators produced by Algorithm 1, according to the matrix chosen for the perturbation. Notably, the estimator is unbiased and its norm $\|\widetilde{\ell}_t\|^2$ admits an almost-sure upper bound, as well as a (potentially) sharper bound in expectation. The almost-sure upper bound is the crucial property that allows us to apply sophisticated comparator-adaptive OLO algorithms as subroutines.

**Proposition 2.1.** *Let $H_t \in \mathbb{R}^{d \times d}$ be positive definite and let $v_1, \ldots, v_d$ be an orthonormal basis of eigenvectors of $H_t$. Consider the set $\mathcal{S} = \{\sigma v_i : \sigma \in \{-1, 1\},\ i \in [d]\}$. Let $s_t$ be sampled uniformly at random from $\mathcal{S}$, and define*

$$\widetilde{w}_t = w_t + H_t^{-\frac{1}{2}} s_t, \quad and \quad \widetilde{\ell}_t = d\,H_t^{\frac{1}{2}} s_t \langle \widetilde{w}_t, \ell_t \rangle.$$

*Then $\mathbb{E}[\widetilde{\ell}_t \mid \mathcal{F}_{t-1}] = \ell_t$ and the following hold:*

$$\mathbb{E}[\|\widetilde{\ell}_t\|_2^2 \mid \mathcal{F}_{t-1}] = d\|\ell_t\|_2^2 + d\langle \ell_t, w_t \rangle^2 \operatorname{Tr}(H_t),$$
$$\|\widetilde{\ell}_t\|_2^2 \leq d^2 \|\ell_t\|_2^2 (\sqrt{\lambda_t}\,\|w_t\| + 1)^2,$$

*where $\lambda_t$ is the eigenvalue of $H_t$ associated with the eigenvector $v_t$ sampled on round $t$.*

From this, we immediately get the following corollary, which will have important implications for the techniques we employ throughout the rest of the paper.

**Corollary 2.2.** *Under the same assumptions as Proposition 2.1, let $\varepsilon \in (0,1)$ and suppose that for all $t$ we set*

$$H_t \preceq \frac{1}{d(\|w_t\|^2 \vee \varepsilon^2)} I_d \,. \tag{4}$$

*Then the following hold almost-surely:*

$$\|\widetilde{\ell}_t\|^2 \leq 4d^2 \|\ell_t\|^2 \quad and \quad \mathbb{E}\big[\|\widetilde{\ell}_t\|^2 | \mathcal{F}_{t-1}\big] \leq 2d\|\ell_t\|^2 \,.$$

The parameter $\varepsilon > 0$ will play a minor role in developing our high-probability guarantees in Section 4, though otherwise serves only to prevent division by zero when $w_t = \mathbf{0}$ and can be set to any positive value.

Corollary 2.2 has two important implications for our purposes. First, since the loss estimates are bounded uniformly by $2d\|\ell_t\| \leq 2dG$, we will be able to apply modern comparator-adaptive OLO algorithms, which require uniformly bounded gradient norms. Second, the conditional second-moment bound yields a sharper bound which can, as we will see, translate into order-optimal minimax regret bounds. In the next sections, we see that this distinction leads to different guarantees depending on whether the adversary can adapt the comparator norm to the loss sequence, or must commit to it in advance.

In what follows, the **oblivious setting** refers to fully oblivious settings where *both* the loss sequence and the comparator sequence are $\mathcal{F}_0$ measurable (*e.g.*, determined before the start of the game). For ease of exposition, we also assume in the oblivious setting that all $\mathcal{F}_0$-measurable quantities are deterministic (equivalently, that $\mathcal{F}_0$ is the trivial $\sigma$-algebra).

### 2.1. Generic Expected Regret Analysis for BLO

We now state a generic reduction showing how the expected-regret guarantees of `PABLO` follow from those of the underlying OLO routine $\mathcal{A}$.

**Proposition 2.3.** *Let $\mathcal{U}$ be a class of sequences [2] in $\mathbb{R}^d$ and suppose that $\mathcal{A}$ guarantees that for any sequence $g_{1:T} = (g_t)_{t=1}^T$ in $\mathbb{R}^d$ and any sequence $u_{1:T} = (u_t)_{t=1}^T \in \mathcal{U}$,*

$$R_T^{\mathcal{A}}(u_{1:T}) \leq B_T^{\mathcal{A}}(u_{1:T}, g_{1:T})$$

*for some function $B_T^{\mathcal{A}} : (\mathbb{R}^d)^{2T} \to \mathbb{R}_{\geq 0}$. Then, for any sequence of losses $\ell_1, \ldots, \ell_T$ and any comparator sequence $u_{1:T} \in \mathcal{U}$, `PABLO` using $\mathcal{A}$ for its OLO learner guarantees*

$$\mathbb{E}\left[R_T(u_{1:T})\right] \leq \mathbb{E}\left[B_T^{\mathcal{A}}\left(u_{1:T}, \widetilde{\ell}_{1:T}\right) + B_T^{\mathcal{A}}\left(u_{1:T}, \delta_{1:T}\right)\right]$$

*where $\delta_t = \ell_t - \widetilde{\ell}_t$ for all $t$, and $\widetilde{\ell}_t$ is defined in Algorithm 1.*

---

[2]Concretely, common classes of sequences are the static comparator sequences, sequences satisfying some path-length or diameter constraint, or the class of all sequences in $\mathbb{R}^d$.

The proof is deferred to Appendix A, and is based on a standard *ghost-iterate* trick (see, e.g., (Nemirovski et al., 2009; Neu & Okolo, 2024)). More generally, the same reduction applies to any bandit algorithm that plays a randomized action whose conditional expectation equals the iterate produced by an OLO routine. Our perturbation scheme is one concrete instantiation of this principle, and has the particular advantage of having loss estimates that are both unbiased and bounded uniformly (for appropriately chosen $H_t$ as in Corollary 2.2), enabling us to apply comparator-adaptive OLO algorithms which require bounded gradient norms.

## 3. Novel expected regret bounds for uBLO

In this section we present several applications of Proposition 2.3, leading to new algorithms and regret guarantees for the uBLO setting within the `PABLO` framework. A key feature of uBLO is that the action domain is unbounded, so the perturbation matrices $(H_t)_{t \geq 1}$ are not constrained by feasibility considerations, so we are free to set $H_t$ according to the conditions of Corollary 2.2. In the remainder of the paper, we therefore adopt the simple isotropic choice in Equation (4), and focus on how different regret guarantees arise from different choices of the underlying OLO routine.

### 3.1. Static Regret via Parameter-free OLO

We first instantiate `PABLO` for uBLO by choosing, as an OLO subroutine, the *parameter-free mirror descent* (PFMD) algorithm of Jacobsen & Cutkosky (2022, Section 3). This choice is motivated by its strong *static* comparator-adaptive guarantee on the unconstrained domain: for any $\epsilon > 0$ and any sequence $(g_t)_{t \geq 1}$ with $\|g_t\| \leq G$ for all $t$, its regret satisfies $R_T^{\mathcal{A}}(u) = \widetilde{O}\left(G\epsilon + \|u\|\sqrt{V_T \log_+\left(\frac{\|u\|\sqrt{V_T}}{G\epsilon}\right)}\right)$, where $V_T = \sum_{t=1}^T \|g_t\|_2^2$, uniformly over $u \in \mathbb{R}^d$. To turn this into a guarantee for the bandit setting, we then apply Proposition 2.3 and obtain the following result.

**Theorem 3.1.** *For any $u \in \mathbb{R}^d$, `PABLO` equipped with Jacobsen & Cutkosky (2022, Algorithm 4) with parameter $\epsilon/d$ guarantees*

$$\mathbb{E}\left[R_T(u)\right] = \widetilde{\mathcal{O}}\left(G\epsilon + \frac{d}{\kappa}\mathbb{E}\left[\|u\|\sqrt{V_T \log_+\left(\frac{d\|u\|\Lambda_T}{G\epsilon}\right)}\right]\right),$$

*where $V_T = \sum_{t=1}^T \|\ell_t\|^2$, $\Lambda_T = G\sqrt{T}\log^2(1+T)$, and $\kappa = \sqrt{d}$ in the oblivious setting and $\kappa = 1$ otherwise.*

**Remark 3.2.** *Theorem 3.1 actually holds more generally for a* partially-oblivious *setting in which the comparator is $\mathcal{F}_0$ measurable but $\ell_{1:T}$ may be adaptive. In this setting, the guarantee scales as $\mathbb{E}\|u\|\sqrt{d\sum_t \mathbb{E}\left[\|\ell_t\|^2|\mathcal{F}_0\right]}$, which recovers the statement for the oblivious setting as a special case. We focus our discussion on the adaptive and oblivious*

*settings in the main text for ease of presentation but provide a more general statement of the result in Appendix B.*

Proof of Theorem 3.1 can be found in Appendix B. A key subtlety, and the reason Theorem 3.1 yields two distinct guarantees, is that using Jensen's inequality to obtain an upper bound of the form

$$\mathbb{E}\left[\|u\|\sqrt{\sum_{t=1}^{T}\|\widetilde{\ell}_t\|_2^2}\right] \leq \|u\|\sqrt{\sum_{t=1}^{T}\mathbb{E}\left[\|\widetilde{\ell}_t\|_2^2\right]}$$

is only justified when the *scale* $\|u\|$ is conditionally independent of the randomness generating $\widetilde{\ell}_t$ (and thus does not depend on the realized trajectory through the losses and actions). When such independence holds (e.g., $\|u\|$ is chosen obliviously at the start of the game), we can exploit the sharper in-expectation control of $\|\widetilde{\ell}_t\|_2^2$. Otherwise, for norm-adaptive comparators, we must rely on the more conservative almost-sure bound from Corollary 2.2 to bound $\|\widetilde{\ell}_t\|^2 = \mathcal{O}(d^2\|\ell_t\|^2)$.

Note that this is generally not a concern in constrained settings: in the standard bounded domain setting, the worst-case comparator is typically on the convex hull of $\mathcal{W}$, and its norm can be bounded by the diameter of $\mathcal{W}$. However, in an unconstrained linear setting, no such finite worst-case comparator exists, and the goal becomes to ensure $R_T(u) \leq B_T(u)$ *for all* $u \in \mathbb{R}^d$ *simultaneously*, where $B_T(u)$ is some non-negative function. This makes the natural worst-case comparator have a data-dependent norm, e.g., $\|u\| \propto \exp\left(\frac{\|\sum_{t=1}^{T}\ell_t\|^2}{G^2 T}\right)$ when choosing $B_T(u)$ to match the minimax optimal bound for OLO (see Appendix H for details). Because of this, from a comparator-adaptive perspective, it is not natural to treat the comparator norm as independent of the losses or the learner's decisions without additional explicit assumptions.

Interestingly the above observation does not seem to be accounted for in prior works. Indeed, as far as we are aware all prior works in this setting are implicitly making the assumption that the comparator norm is oblivious rather than adaptive (van der Hoeven et al., 2020; Luo et al., 2022; Rumi et al., 2026), and the stated guarantees can potentially be very different without this assumption, as detailed in the following discussion.

**Comparison with existing work.** We proved that PABLO combined with PFMD yields comparator-adaptive *expected* regret bounds under two adversarial regimes depending on when the norm of the comparator is selected. It is instructive to compare these guarantees to what can be obtained from the scale/direction decomposition of van der Hoeven et al. (2020). Concretely, consider a decomposition in which the *scale* is learned by Algorithm 1 of Cutkosky & Orabona (2018) and the *direction* is learned by OSMD

specialized to the unit Euclidean ball (Bubeck et al., 2012, Section 5). If the norm $\|u\|$ is oblivious, this combination yields the bound

$$R_T(u) = \mathcal{O}\left(\|u\|\sqrt{T\left(\log\left(\frac{\|u\|\sqrt{T}}{\epsilon}\right)\vee d\right)}\right), \quad (5)$$

which can improve over Theorem 3.1 by up to a factor $\sqrt{d}$ when the log term matches the dimension.

However, this advantage hinges on applying the *in-expectation* second-moment control for the direction estimator, and therefore does not extend to the norm-adaptive regime. In particular, the standard OSMD guarantee on the unit ball (namely, the $\mathcal{O}(\sqrt{dT})$ term) does not directly translate when the comparator scale is allowed to be chosen adaptively and may be coupled with the realized trajectory. As we show in Appendix G, obtaining regret bounds against such norm-adaptive adversaries requires re-tuning the algorithm, and the resulting rate degrades to $\widetilde{\mathcal{O}}((dT)^{2/3})$. This highlights a key benefit of PABLO : it maintains $\sqrt{T}$-type regret guarantees in the horizon uniformly across both regimes, without requiring regime-dependent tuning.

### 3.2. Dynamic Regret in Expectation

To achieve dynamic regret guarantees, we now apply PABLO with a suitable OLO algorithm for unconstrained dynamic regret. The following result shows that the optimal $\sqrt{P_T}$ dependence can be obtained by leveraging the parameter-free dynamic regret algorithm of Jacobsen & Cutkosky (2022). We provide a modest refinement of their result which removes $M = \max_t \|u_t\|$ completely from the main term in the bound, and showcases a refined measure of comparator variability: the *log-linear path-length*

$$P_T^{\Phi} = \sum_{t=2}^{T}\|u_t - u_{t-1}\|\log\left(\|u_t - u_{t-1}\|T^3/\epsilon + 1\right)$$

which is an adaptive refinement of the $P_T\log\left(MT^3/\epsilon + 1\right)$ dependence reported by Jacobsen & Cutkosky (2022). This result is of independent interest and is provided in Appendix E.2. Then applying PABLO with this dynamic regret algorithm leads to the expected dynamic regret guarantee for uBLO presented in Theorem 3.3 below.

Proof of the following theorem can be found in Appendix C. The closest result to ours is the recent work of Rumi et al. (2026), which also achieves adaptivity to the comparator sequence without prior knowledge. However, their result scales with the *switching number* $S_T = \sum_{t=2}^{T}\mathbb{I}\{u_t \neq u_{t-1}\}$, which is closely related to $P_T$ but is a weaker measure of variation, failing to account for the potentially heterogeneous magnitudes of the increments $\|u_t - u_{t-1}\|$. Our result is therefore the first to achieve $\sqrt{P_T}$

adaptivity to the genuine path-length $P_T$. Moreover, their result is restricted to the oblivious adversarial setting, whereas our results remain meaningful even against fully-adaptive comparator and loss sequences.

**Theorem 3.3.** *For any sequence $u_{1:T} = (u_t)_{t=1}^T$ in $\mathbb{R}^d$, PABLO equipped with Algorithm 6 tuned with $\epsilon/d$ guarantees*

$$\mathbb{E}\left[R_T(u_{1:T})\right] = \mathcal{O}\left(\mathbb{E}\left[\frac{d}{\kappa}\sqrt{(\Phi_T + P_T^\Phi)\sum_{t=1}^T \|\ell_t\|^2 \|u_t\|}\right.\right.$$

$$\left.\left. + dG(\epsilon + \max_t \|u_t\| + \Phi_T + P_T^\Phi)\right]\right).$$

*where $\kappa = \sqrt{d}$ in the oblivious setting and $\kappa = 1$ otherwise, and we define $\Phi_T = \|u_T\|\log\left(\frac{\|u_T\|T}{\epsilon} + 1\right)$ and $P_T^\Phi = \sum_{t=2}^T \|u_t - u_{t-1}\|\log\left(\frac{\|u_t - u_{t-1}\|T^3}{\epsilon} + 1\right)$.*

Besides achieving the optimal $\sqrt{P_T}$ dependence, we inherit another novelty from the algorithm of (Jacobsen & Cutkosky, 2022): the bound of Theorem 3.3 is also adaptive to the *individual* comparator norms, with a variance penalty scaling with $\sum_{t=1}^T \|\ell_t\|^2 \|u_t\|$. This leads to a property which is similar in spirit to a strongly-adaptive guarantee (Daniely et al., 2015), in the sense that if the comparator sequence is only active (non-zero) within a sub-interval $[a, b]$, then the regret automatically restricts to that same sub-interval, $R_T(u_{1:T}) = \widetilde{\mathcal{O}}(\sqrt{P_{[a,b]}|b-a|})$, where $P_{[a,b]} = \sum_{t=a+1}^b \|u_t - u_{t-1}\|$ is the path-length over the interval. While Jacobsen & Cutkosky (2022) show that one cannot obtain comparator-adaptive guarantees on all sub-intervals *simultaneously* (thereby extending the impossibility result of Daniely et al. (2015) to unbounded domains), the per-comparator adaptivity in Theorem 3.3 can be viewed as a natural, achievable analogue of strong adaptivity in the unbounded setting.

Finally, we again observe a $\sqrt{d}$ discrepancy between the upper bounds obtained against a norm-oblivious or norm-adaptive adversary, leading to similar insights as observed in the previous section. Likewise, our result more generally holds for the partially-oblivious setting wherein $u_{1:T}$ is $\mathcal{F}_0$ measurable but $\ell_{1:T}$ may be adaptive, in which case the $\|\ell_t\|^2$ dependencies are replaced by the more general $\mathbb{E}\left[\|\ell_t\|^2|\mathcal{F}_0\right]$, as discussed in Remark 3.2.

## 4. High-probability Bounds

In this section we derive novel high-probability bounds for both static and dynamic regret, for new instances of PABLO. As in the previous setting, we begin with a general reduction to unconstrained OLO, and study the additional penalties that emerge due to the loss estimates. We again use $H_t$ from Eq. (4) in all applications presented in this section,

though in this section we will choose $\varepsilon^2 \propto 1/T$; this will ensure that the perturbations $H_t^{-1/2}s_t$ in Algorithm 1 have sufficiently nice concentration properties in the following reduction.

**Proposition 4.1.** *Let $\mathcal{A}$ be an OLO learner, $\delta \in (0, 1/3]$, and for all $t$ set $H_t$ as in Equation (4) for $\varepsilon^2 \propto 1/T$. Let $u_{1:T} = (u_t)_{t=1}^T$ be an arbitrary $(\mathcal{F}_t)_{t=0}^T$-adapted sequence in $\mathbb{R}^d$. Then, PABLO guarantees that with probability at least $1 - 3\delta$,*

$$R_T(u_{1:T}) \leq \widetilde{\mathcal{O}}\left(\widetilde{R}_T^{\mathcal{A}}(u_{1:T}) + G\sqrt{d\sum_{t=1}^T \|w_t\|^2 \log\left(\tfrac{1}{\delta}\right)}\right.$$

$$\left. + G\sqrt{d\sum_{t=1}^T \|u_t\|^2 \log\left(\tfrac{1}{\delta}\right)} + dGP_T\right),$$

*where $\widetilde{R}_T^{\mathcal{A}}(u_{1:T})$ is the regret of $\mathcal{A}$ against the losses $(\widetilde{\ell}_t)_t$.*

**Remark 4.2.** *Each of the results in this section in fact generalize to arbitrary comparator sequences via the same "ghost-iterate" trick as in Section 3.1, though the statement of the result above becomes a bit more involved to state. We focus on the $(\mathcal{F}_t)_{t=0}^T$-adapted case here for ease of exposition and discuss the extension to general comparator sequences in Appendix D.3.1.*

Proof of the theorem can be found in Appendix D.1. It shows that uBLO can also be effectively reduced to uOCO (with regard to high-probability bounds) at the expense of three additional terms. The latter two comparator-dependent terms are fairly benign and amount to a lower-order $\mathcal{O}(GP_T)$ and a $\widetilde{\mathcal{O}}(M\sqrt{dT\log(T/\delta)})$, both of which are expected in this setting. However, the term $G\sqrt{d\sum_{t=1}^T \|w_t\|^2 \log(1/\delta)}$ is algorithm dependent and could be arbitrarily large in an unbounded domain. Thus, the main difficulty to achieve high-probability bounds for PABLO stems from controlling the stability of the iterates $w_t$ from the OLO learner $\mathcal{A}$.

Fortunately, a similar concern was recently addressed by Zhang & Cutkosky (2022) in the context of unconstrained stochastic optimization with heavy-tailed noise. Their approach is based on adding an additional composite penalty $\varphi_t$ to the losses, which introduces an extra term $\sum_{t=1}^T \varphi_t(u) - \varphi_t(w_t)$ into the regret bound. With this, the goal is to choose $\varphi_t$ in such a way that $-\sum_{t=1}^T \varphi_t(w_t)$ is large enough to cancel with the $\|w_t\|$-dependent terms above, while also ensuring that $\sum_{t=1}^T \varphi_t(u)$ is not too large. Zhang & Cutkosky (2022) provide a Huber-like penalty which satisfies both of these conditions (see Lemma E.4).

The difficulty with the above approach is that by introducing the composite penalty, we change the OLO learner's feedback: on round $t$, the learner's feedback becomes

$\widetilde{g}_t = g_t + \nabla\varphi_t(w_t)$ instead of just $g_t \in \partial f_t(w_t)$, and the $\nabla\varphi_t(w_t)$ dependence may itself lead to $\|w_t\|$-dependent penalties in the bound. This issue can be fixed using an optimistic update by setting hints $h_t = \nabla\varphi_t(w_t)$, so that the usual $\sum_{t=1}^{T} \|\widetilde{g}_t\|^2$ penalties in the final bound become $\sum_{t=1}^{T} \|\widetilde{g}_t - h_t\|^2 = \sum_{t=1}^{T} \|g_t\|^2$, thus removing the problematic dependence on $\nabla\varphi_t(w_t)$.

Plugging this approach into our framework, and composing Proposition 4.1 with Zhang & Cutkosky (2022, Theorem 3), we obtain the following high-probability guarantee. Notably, the result matches the best-known results from the constrained setting (Zimmert & Lattimore, 2022) up to polylogarithmic terms.

**Theorem 4.3.** *Let* `PABLO` *be implemented with Zhang & Cutkosky (2022, Algorithm 1), and $\delta \in (0, 1/3]$. Then for any $\mathcal{F}_0$-measurable $u \in \mathbb{R}^d$, with probability at least $1 - 3\delta$,*

$$R_T(u) \leq \widetilde{\mathcal{O}}\Big(dG(\epsilon + \|u\|)\log\big(\tfrac{T}{\delta}\big) + G\|u\|\sqrt{dT\log\big(\tfrac{T}{\delta}\big)}\Big).$$

Interestingly, a similar strategy leveraging composite regularization and optimism can also be used to obtain high-probability dynamic regret bounds. In Appendix E.1 (Theorem E.2), we provide an algorithm $\mathcal{A}_\eta$ that guarantees that for any sequences $u_{1:T}$ and $\widetilde{\ell}_{1:T}$ in $\mathbb{R}^d$,

$$\sum_{t=1}^{T} \big\langle \widetilde{\ell}_t, w_t^\eta - u_t \big\rangle \leq \widetilde{\mathcal{O}}\Big(\frac{\|u_T\| + P_T}{\eta} + \eta\sum_{t=1}^{T} \|\widetilde{\ell}_t\|^2\|u_t\|$$
$$+ \sum_{t=1}^{T} \varphi_t(u_t) - \varphi_t(w_t^\eta)\Big), \quad (6)$$

for any convex and $H$-Lipschitz function $\varphi_t : \mathbb{R}^d \to \mathbb{R}_{\geq 0}$ and $\eta$ satisfying $\eta(\|\widetilde{\ell}_t\| + H) \leq 1$. This bound matches the regret bound obtained in the expected dynamic regret setting when $\eta$ is optimally tuned, but additionally exhibits a term $\sum_{t=1}^{T} \varphi_t(u_t) - \varphi_t(w_t^\eta)$. Note that achieving this bound requires developing a novel black-box optimistic reduction which obtains adaptivity to the individual comparator norms $\|u_t\|$, which is not possible using the optimistic reductions in Zhang & Cutkosky (2022) or Cutkosky (2019), so this result is of independent interest. We then obtain the optimal trade-off in $\eta$ using a standard technique for combining comparator-adaptive guarantees: by running the algorithm in parallel over a grid of values of $\eta$ and playing $w_t = \sum_\eta w_t^\eta$, we obtain the tuned bound

$$\sum_{t=1}^{T} \big\langle \widetilde{\ell}_t, w_t - u_t \big\rangle \leq \widetilde{\mathcal{O}}\Big(d\sqrt{(\|u_T\| + P_T)\sum_{t=1}^{T} \|\ell_t\|^2\|u_t\|}$$
$$+ \sum_{t=1}^{T} \varphi_t(u_t) - \sum_\eta\sum_{t=1}^{T} \varphi_t(w_t^\eta)\Big).$$

Finally, we show that for the Huber-like composite penalty $\varphi_t$ defined by Zhang & Cutkosky (2022), the aggregate-iterate $\|w_t\| = \|\sum_\eta w_t^\eta\|$ dependencies from Proposition 4.1 are canceled out by the aggregate penalty $-\sum_\eta\sum_{t=1}^{T} \varphi_t(w_t^\eta)$, while also ensuring that $\sum_{t=1}^{T} \varphi_t(u_t) = \widetilde{O}\big(\sqrt{d\sum_{t=1}^{T} \|u_t\|^2\log(T/\delta)}\big)$. A detailed description of the algorithm, along with proof of its regret guarantee stated below, can be found in Appendix D.3.

**Theorem 4.4.** *Let $\delta \in (0, 1/4]$. Then* `PABLO` *applied with Algorithm 3 and appropriately-chosen parameters (depending only on $G$, $\delta$, $T$, and $d$, given explicitly in Appendix D.3) guarantees that for any $(\mathcal{F}_t)_{t=0}^{T}$-adapted sequence $u_{1:T}$ in $\mathbb{R}^d$, with probability at least $1 - 4\delta$ it holds that*

$$R_T(u_{1:T}) \leq \widetilde{\mathcal{O}}\Big(\sqrt{d(\Phi_T + P_T^\Phi)\big[d\mathcal{V}_T \wedge \Omega_T\big]}$$
$$+ G\sqrt{d\sum_{t=1}^{T} \|u_t\|^2\log\big(\tfrac{T}{\delta}\big)}$$
$$+ dG(\epsilon + M + \Phi_T + P_T^\Phi)\log\big(\tfrac{T}{\delta}\big)\Big),$$

*where $M = \max_t \|u_t\|$, $\mathcal{V}_T = \sum_{t=1}^{T} \|\ell_t\|^2\|u_t\|$, $\Omega_T = MG^2(T + d\log(\tfrac{1}{\delta}))$, and we denote $\Phi_T = \|u_T\|\log\big(\tfrac{\|u_T\|T}{\epsilon} + 1\big)$ and $P_T^\Phi = \sum_{t=2}^{T} \|u_t - u_{t-1}\|\log\big(\tfrac{\|u_t-u_{t-1}\|T^3}{\epsilon} + 1\big)$.*

The full proof can be found in Appendix D.3. To understand the bound, consider first taking $[d\mathcal{V}_T \wedge \Omega_T] \leq \Omega_T$. In this case the bound reduces to

$$R_T(u_{1:T}) \leq \widetilde{\mathcal{O}}\Big(dG(\epsilon + M + P_T)\log\big(\tfrac{T}{\delta}\big)$$
$$+ GM\sqrt{dT\log\big(\tfrac{T}{\delta}\big)} + \sqrt{d(M^2 + MP_T)T}\Big).$$

Therefore, the bound captures the same worst-case $G\|u\|\sqrt{dT}$ bound as Theorem 4.3 in the static regret ($P_T = 0$) setting, matching the best-known result from the constrained setting up to poly-logarithmic terms as a special case (Zimmert & Lattimore, 2022). At the same time, if instead we bound $[d\mathcal{V}_T \wedge \Omega_T] \leq d\mathcal{V}_T$, we obtain a per-comparator adaptivity similar to the expected regret guarantee in Theorem 3.1, with $R_T(u_{1:T})$ bounded by

$$\widetilde{O}\Big(G\sqrt{d\sum_{t=1}^{T} \|u_t\|^2\log\big(\tfrac{T}{\delta}\big)} + d\sqrt{(\Phi_T + P_T^\Phi)\sum_{t=1}^{T} \|\ell_t\|^2\|u_t\|}$$
$$+ dG(\epsilon + M + P_T)\log\big(\tfrac{T}{\delta}\big)\Big).$$

Hence the bound retains the strong-adaptivity-like property discussed in Section 3.2, in which the bound automatically restricts to a sub-interval $[a, b]$ when comparing against comparator sequences which are only active on $[a, b]$, and also avoids $M = \max_t \|u_t\|$ in all but lower-order terms in the bound.

## 5. Towards lower bounds for uBLO

This section provides some insights on lower bounds for unconstrained adversarial linear bandits. We present a conjecture for the static-regret minimax rate, guided by known OCO lower bounds and by a self-contained proof of the folklore $\widetilde{\Theta}(\sqrt{dT})$ minimax bound on the unit Euclidean ball, which captures the intrinsic difficulty of identifying a favorable direction under mildly biased losses. We then discuss post-hoc comparator norm adaptivity, motivating lower-bound formulations that simultaneously control both the expected comparator norm and its worst-case magnitude.

We recall the scale/direction regret decomposition from van der Hoeven et al. (2020, Lemma 1). Although none of our algorithms use this approach, it is central for the discussions in this section: at each step $t$ we decompose the learner's action $x_t$ as $x_t = v_t z_t$, where $v_t \in \mathbb{R}$ is a scalar ("scale") and $z_t \in \mathbb{B}_d$ is a unit vector ("direction"). Then, it can be shown (Cutkosky & Orabona, 2018) that the full (unconstrained) regret $R_T(u)$ can be written as

$$R_T(u) = R_T^{\mathcal{V}}(\|u\|) + \|u\| \, R_T^{\mathcal{Z}}\left(\frac{u}{\|u\|}\right), \text{ with } \quad (7)$$

$R_T^{\mathcal{V}}(\|u\|) := \sum_{t=1}^{T} \big(v_t - \|u\|\big) \langle z_t, \ell_t \rangle$ (scale regret), and $R_T^{\mathcal{Z}}\left(\frac{u}{\|u\|}\right) := \sum_{t=1}^{T} \left\langle z_t - \frac{u}{\|u\|}, \ell_t \right\rangle$ (direction regret).

**"Scale" lower bound from uOLO.** We observe that one-dimensional online linear optimization (1D-OLO) is embedded in uBLO, if the adversary chooses to provide losses supported on a single coordinate. We may therefore invoke an existing 1D-OLO lower bound, stated below as a mild simplification of Thm. 7 in (Streeter & McMahan, 2012).

**Theorem 5.1** (Streeter & McMahan (2012, Theorem 7)). *For any uBLO algorithm $\mathcal{A}$ satisfying $R_T(0) \leq \epsilon$, if $\|u\|$ is $\mathcal{F}_0$-measurable and satisfies $\|u\| \leq \frac{\epsilon}{\sqrt{T}} 10^{\frac{T}{4}}$, then there exists a sequence $\ell_1, \ldots, \ell_T$ such that*

$$R_T(u) \geq \frac{1}{3} \cdot \|u\| \sqrt{T \log\left(\frac{\|u\|\sqrt{T}}{\epsilon}\right)} \, .$$

Moreover, in the norm-adaptive case the same bound holds with $\|u\|$ replaced by $\mathbb{E}\|u\|$, by the same arguments as in (Streeter & McMahan, 2012), replacing the radius parameter (denoted $R$ therein) by $\mathbb{E}\|u\|$, since $\mathbb{E}\|u\|$ is $\mathcal{F}_0$-measurable.

**Lower bound on the direction regret.** Because it comes from a hard instance for *scale learning*, the lower bound from the previous paragraph does not help explain about the $\sqrt{dT}$ component of the regret upper bounds obtained for the static regret (Thm. 3.1, Eq. (5)). It is thus natural to assume that this $d$-dependency might come from the direction regret. To support this, we prove the folklore conjecture on the minimax regret for linear bandits constrained in the

Euclidean ball, which is thus a result of independent interest. In the context of uBLO, it applies to the direction regret in the scale-direction regret decomposition (Eq. (7)).

**Theorem 5.2** (Lower bound on the unit ball). *Assume $T \geq 4d$. Then, for any algorithm $\mathcal{A}$ playing actions $z_1, \ldots, z_T$ in the unit Euclidean ball it holds that*

1. *There exists a parameter $\theta \in \mathbb{R}^d$ and a sub-Gaussian distribution $\mathbb{P}_\theta$, such that $\mathbb{E}_{\ell \sim \mathbb{P}_\theta}[\ell] = \theta$, $\mathbb{E}_{\ell \sim \mathbb{P}_\theta}[\|\ell\|^2] \leq 1$, for which it holds that*

$$R_T^{sto}(\mathcal{A}, \theta) := \mathbb{E}_{(\ell_t)_{t=1}^T \sim \mathbb{P}_\theta^T}\left[R_T^{\mathcal{Z}}\left(\frac{\theta}{\|\theta\|}\right)\right] \geq \frac{\sqrt{dT}}{64} \wedge \frac{T}{12d} \, .$$

2. *There exists a sequence of losses $\ell_1, \ldots, \ell_T$ satisfying $\|\ell_t\| \leq 1$ for all $t \geq 1$, a comparator $u \in \mathbb{B}_d$, and an absolute constant $C$ such that*

$$R_T^{\mathcal{Z}}(u) \geq C \cdot \left\{\sqrt{dT} \wedge \frac{T}{d}\right\} \cdot \sqrt{\frac{d}{d \vee \log(T)}} \, .$$

*Proof sketch.* We follow the outline of the proof of Theorem 24.2 in (Lattimore & Szepesvári, 2020), based on the difficulty of distinguishing problem instances indexed by a parameter $\theta$ drawn from a small hypercube around the origin, $\theta \in \{\pm\Delta\}^d$ with $\Delta = \Theta(T^{-1/2})$. We consider losses generated as $\ell_t = \theta + \varepsilon_t$ with $\varepsilon_t \sim \mathcal{N}(0, (2d)^{-1}I_d)$, and then compare pairs of environments $\theta$ and $\theta'$ that differ only in the sign of a single coordinate, and relate the resulting regret contributions via a Pinsker/KL argument up to a suitable stopping time. In our feedback model, the KL term involves the ratio $x_{ti}^2/\|x_t\|^2$, the Gaussian noise being inside the inner product, which we control by noting that $\|x_t\|$ shouldn't be bounded away from 1 on many rounds, otherwise the learner incurs a linear regret $\frac{T}{12d}$. Otherwise, $\|x_t\|$ is typically close to 1 and the KL analysis proceeds. The $1/d$ factor in the noise variance (which is 1 in (Lattimore & Szepesvári, 2020)) is exactly what yields the $\sqrt{dT}$ scaling (rather than $d\sqrt{T}$), and a standard randomization argument over $\theta$ concludes that $\sup_\theta R_T^{sto}(\mathcal{A}, \theta) \gtrsim \sqrt{dT}$.

The second lower bound follows from an analogous construction, with an added truncation to ensure losses lie in the unit ball. Truncation induces a bias term in the analysis, and alters the information structure. Thus, using a chi-squared concentration bound from (Laurent & Massart, 2000), we calibrate the noise so that truncation occurs with negligible probability, keeping the model close enough to the Gaussian case for the argument to proceed. We show that this is sufficient to also make the bias become negligible, and the result follows by extending the randomization argument to an adversarial loss sequence. □

The complete proof can be found in Appendix F. The key takeaway of the construction is that the improved $\sqrt{d}$ depen-

dence on the Euclidean ball (as opposed to the $d$ dependence in other geometries) is driven by a tighter noise-variance constraint, making a gap of order $\sqrt{1/T}$ as hard as a gap of order $\sqrt{d/T}$ in the model studied in Lattimore & Szepesvári (2020, Theorem 24.2).

This lower bound transfers directly to the *direction* term $R_T^{\mathcal{Z}}(u/\|u\|)$ in (7). When $\|u\|$ is $\mathcal{F}_0$-measurable, the decomposition immediately yields a scale-up by $\|u\|$. Moreover, it also yields a lower bound by $\mathbb{E}[\|u\|]\,\mathbb{E}\left[R_T^{\mathcal{Z}}(u/\|u\|)\right]$ in the norm-adaptive setting, as the adversary can correlate $\|u\|$ positively with the realized regret. However, we emphasize that this does not yield a lower bound on the full regret $R_T(u)$, as nothing prevents the scale regret $R_T^{\mathcal{V}}(\|u\|)$ from being large and negative.

**Conjecture on the lower bound for uBLO.** Based on the previous results, we conjecture that the regret bound presented in Equation (5)—obtained by combining a coin bettor with OSMD—is minimax optimal if the comparator norm is oblivious.

**Conjecture 5.3.** *If the comparator norm is oblivious, the minimax static regret guarantee for uBLO is*

$$R_T(u) \;=\; \Theta\!\left(\|u\|\sqrt{T\big(d \vee \log\|u\|\big)}\right)\;.$$

We leave a formal proof as an open problem, and briefly explain why it does not follow from the results of this section. The two lower bounds above capture complementary difficulties that can be interpreted through a stochastic-adversary lens: when losses exhibit only a weak average bias of order $T^{-1/2}$ in some direction, the learner must both (i) control risk and refrain from scaling up too aggressively, and (ii) remain uncertain about the true direction, since the losses could plausibly be pure noise or biased elsewhere.

However, these statements do not directly combine. Theorem 5.2 only ensures the existence of a loss sequence that forces $\Omega(\sqrt{dT})$ *direction* regret, but it does not guarantee that the same sequence is simultaneously hard for *scale* learning. For example, an algorithm may enforce $\Omega(\sqrt{dT})$ exploration on every sequence, yet still accumulate enough gains on some sequences to scale up. In fact, it could even use different scales during exploration and exploitation.

Thus, proving the conjecture seems to require constructing a single loss sequence that simultaneously forces $\Omega(\sqrt{dT})$ regret due to uniform exploration across all directions and prevents overly aggressive exploitation, in the sense of limiting how much the learner can scale up. Achieving this joint property appears non-trivial with standard randomization-hammer techniques, which underlie both lower bounds here.

**Norm adaptivity.** Our preliminary results do not explain the $\sqrt{d}$ gap in the upper bounds between the oblivious and norm-adaptive comparator settings. Whether this gap is intrinsic remains open. More fundamentally, it is still unclear what assumptions on the adversary are appropriate for deriving lower bounds in the norm-adaptive setting.

To illustrate the difficulty, consider the idea of controlling only the expected comparator norm, say $\mathbb{E}\|u\| = M$. We argue that this assumption is too weak to yield lower bounds that are meaningfully comparable with our upper bounds.

Consider, for simplicity, a policy that enforces uniform exploration with probability at least $\gamma$ at each round[3], and assume that the loss sequence and post-hoc comparator norms can be coupled with the realized learner's internal randomness. Let $E$ be the event that all rounds are exploratory, and write $p := \mathbb{P}(E) \geq \gamma^T$. Then, we can observe that the adversary can both enforce the learner's expected cumulative reward to be zero[4], and guarantee that

$$\mathbb{E}\|u\| = M \quad \text{and} \quad \mathbb{E}\left[R_T(u)\right] \geq \mathbb{E}\left[\frac{M}{p}\mathbb{I}(E)\cdot T\right] = MT.$$

This does not contradict comparator-dependent upper bounds such as (5), since on the event $E$ the realized comparator norm is $M/p$, so $\log\|u\|$ may be of order $T$. However, the construction is uninformative, as it obtains a linear lower bound only by coupling an exponentially large comparator norm with an exponentially unlikely event. This suggests that sharper restrictions on $\|u\|$ are needed to characterize the difficulty of uBLO in the norm-adaptive setting.

# 6. Future Directions

We leave open several directions for future work. As discussed in Section 5, novel techniques seem necessary to prove complete lower bounds for unconstrained BLO, both against norm-oblivious and norm-adaptive adversaries. It also remains to understand whether a dimension-dependent gap between the two settings can be avoided.

Our dynamic-regret guarantees also raise a natural question: under what conditions can one obtain non-trivial dynamic regret bounds in bandit settings *without* prior knowledge of, e.g., $P_T$? In many constrained problems, such guarantees are known to be impossible against adaptive adversaries (see for instance, Marinov & Zimmert (2021)). The uBLO setting may be an extreme regime that evades these lower bounds by removing the domain constraints. An interesting direction is to identify more general assumptions under which $\sqrt{P_T}$-type dependencies remain achievable.

Finally, follow-up work could seek to extend our approach to the more general Bandit Convex Optimization setting, in which the losses are arbitrary convex functions.

---

[3]The same idea can be extended to broader policy classes, but this assumption keeps the example transparent.

[4]e.g. starting with a constant unit loss at first, and switching to 0 after the first non-exploratory round

## Acknowledgements

NCB and AJ acknowledge the financial support from the EU Horizon CL4-2022-HUMAN-02 research and innovation action under grant agreement 101120237, project ELIAS (European Lighthouse of AI for Sustainability). This work was initiated while DB was visiting Università degli Studi di Milano. The visit was supported by Inria Grenoble and UK Research and Innovation (UKRI) under the UK government's Horizon Europe funding guarantee [grant number EP/Y028333/1]. SI was supported by JSPS KAKENHI Grant Number JP25K03184 and by JST PRESTO, Japan, Grant Number JPMJPR2511.

## Impact Statement

This paper presents work whose goal is to advance the field of Machine Learning. There are many potential societal consequences of our work, none which we feel must be specifically highlighted here.

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

# A. General Reduction to OLO for Expected Regret

In this section we detail the proof of our general reduction for expected regret guarantees. The statement is framed in terms of the regret guarantee which holds for a given class of comparator sequences $\mathcal{U}$. For instance, Algorithms $\mathcal{A}$ which only make static regret guarantees can also be applied in Proposition 2.3 by considering the class of sequences with $u_1 = \ldots = u_T = u$ for some $u \in \mathbb{R}^d$; algorithms which make dynamic regret guarantees in a bounded domain or under a budget constraint can be captured by the class of sequences such that $\|u_t\| \leq D$ for all $t$ or sequences satisfying $\sum_t \|u_t - u_{t-1}\| \leq \tau$ for some $\tau$; algorithms for unconstrained dynamic regret such as Jacobsen & Cutkosky (2022, Algorithm 2) can be applied with $\mathcal{U}$ being the class of all sequences in $\mathbb{R}^d$.

**Proposition 2.3.** *Let $\mathcal{U}$ be a class of sequences [5] in $\mathbb{R}^d$ and suppose that $\mathcal{A}$ guarantees that for any sequence $g_{1:T} = (g_t)_{t=1}^T$ in $\mathbb{R}^d$ and any sequence $u_{1:T} = (u_t)_{t=1}^T \in \mathcal{U}$,*

$$R_T^{\mathcal{A}}(u_{1:T}) \leq B_T^{\mathcal{A}}(u_{1:T}, g_{1:T})$$

*for some function $B_T^{\mathcal{A}} : (\mathbb{R}^d)^{2T} \to \mathbb{R}_{\geq 0}$. Then, for any sequence of losses $\ell_1, \ldots, \ell_T$ and any comparator sequence $u_{1:T} \in \mathcal{U}$, `PABLO` using $\mathcal{A}$ for its OLO learner guarantees*

$$\mathbb{E}\left[R_T(u_{1:T})\right] \leq \mathbb{E}\left[B_T^{\mathcal{A}}\left(u_{1:T}, \widetilde{\ell}_{1:T}\right) + B_T^{\mathcal{A}}\left(u_{1:T}, \delta_{1:T}\right)\right]$$

*where $\delta_t = \ell_t - \widetilde{\ell}_t$ for all $t$, and $\widetilde{\ell}_t$ is defined in Algorithm 1.*

*Proof.* The proposition is just a statement of the standard "ghost-iterate" trick. We have that $w_t$ is $\mathcal{F}_{t-1}$-measurable and that $\mathbb{E}\left[\ell_t - \widetilde{\ell}_t | \mathcal{F}_{t-1}\right] = 0$ via Proposition 2.1, so

$$
\begin{aligned}
\mathbb{E}\left[R_T(u_{1:T})\right] &= \mathbb{E}\left[\sum_{t=1}^T \langle \ell_t, \widetilde{w}_t - u_t \rangle\right] \\
&= \mathbb{E}\left[\sum_{t=1}^T \left\langle \widetilde{\ell}_t, w_t - u_t \right\rangle + \sum_{t=1}^T \left\langle \ell_t - \widetilde{\ell}_t, w_t - u_t \right\rangle\right] \\
&\leq \mathbb{E}\left[B_T^{\mathcal{A}}(u_{1:T}, \widetilde{\ell}_{1:T}) + \sum_{t=1}^T -\left\langle \ell_t - \widetilde{\ell}_t, u_t \right\rangle\right],
\end{aligned}
$$

where the last line uses the regret guarantee of $\mathcal{A}$ applied to losses $w \mapsto \left\langle \widetilde{\ell}_t, w \right\rangle$. Now let $\widehat{w}_t$ be the iterates of a "virtual instance" of $\mathcal{A}$ which is applied to losses $w \mapsto \left\langle \ell_t - \widetilde{\ell}_t, w \right\rangle$. Note that this virtual instance exists only in the analysis and doesn't need to be implemented, so there is no issue running the algorithm against the losses $w \mapsto \left\langle \ell_t - \widetilde{\ell}_t, w \right\rangle$, which would otherwise be unobservable to $\mathcal{A}$. Then, since $\widehat{w}_t$ is $\mathcal{F}_{t-1}$-measurable, we have via tower-rule that

$$
\begin{aligned}
\mathbb{E}\left[R_T(u_{1:T})\right] &\leq \mathbb{E}\left[B_T^{\mathcal{A}}(u_{1:T}, \widetilde{\ell}_{1:T}) + \sum_{t=1}^T \left\langle \ell_t - \widetilde{\ell}_t, \pm\widehat{w}_t - u_t \right\rangle\right] \\
&\leq \mathbb{E}\left[B_T^{\mathcal{A}}(u_{1:T}, \widetilde{\ell}_{1:T}) + B_T^{\mathcal{A}}\left(u_{1:T}, \left(\ell_t - \widetilde{\ell}_t\right)_{1:T}\right) - \sum_{t=1}^T \left\langle \ell_t - \widetilde{\ell}_t, \widehat{w}_t \right\rangle\right] \\
&\leq \mathbb{E}\left[B_T^{\mathcal{A}}(u_{1:T}, \widetilde{\ell}_{1:T}) + B_T^{\mathcal{A}}\left(u_{1:T}, \left(\ell_t - \widetilde{\ell}_t\right)_{1:T}\right) - \sum_{t=1}^T \left\langle \mathbb{E}\left[\ell_t - \widetilde{\ell}_t | \mathcal{F}_{t-1}\right], \widehat{w}_t \right\rangle\right] \\
&\leq \mathbb{E}\left[B_T^{\mathcal{A}}(u_{1:T}, \widetilde{\ell}_{1:T}) + B_T^{\mathcal{A}}\left(u_{1:T}, \left(\ell_t - \widetilde{\ell}_t\right)_{1:T}\right)\right]
\end{aligned}
$$

$\square$

---

[5]Concretely, common classes of sequences are the static comparator sequences, sequences satisfying some path-length or diameter constraint, or the class of all sequences in $\mathbb{R}^d$.

# B. Proof of Theorem 3.1

As discussed in Remark 3.2, the proof of the following result would allow a slightly more general statement which covers a more general *partially*-oblivious setting, wherein the comparator norm $\|u\|$ is $\mathcal{F}_0$ measurable but the loss sequence may be adaptive.

**Theorem 3.1.** *For any $u \in \mathbb{R}^d$, `PABLO` equipped with Jacobsen & Cutkosky (2022, Algorithm 4) with parameter $\epsilon/d$ guarantees*

$$\mathbb{E}\left[R_T(u)\right] = \widetilde{\mathcal{O}}\left(G\epsilon + \mathbb{E}\left[d\|u\|\sqrt{\sum_{t=1}^{T}\|\ell_t\|^2 \log\left(\tfrac{d\|u\|\Lambda_T}{G\epsilon} + 1\right)}\right]\right),$$

*where $\Lambda_T = \sqrt{\sum_{t=1}^{T}\|\ell_t\|^2 \log^2\left(1 + \sum_{t=1}^{T}\|\ell_t\|^2/G^2\right)}$. Moreover, if $\|u\|$ is $\mathcal{F}_0$ measurable, then*

$$\mathbb{E}\left[R_T(u)\right] = \widetilde{\mathcal{O}}\left(G\epsilon + \mathbb{E}\left[\|u\|\sqrt{d\sum_{t=1}^{T}\mathbb{E}\left[\|\ell_t\|^2|\mathcal{F}_0\right]\log\left(\tfrac{d\|u\|\Lambda_T^+}{G\epsilon} + 1\right)}\right]\right).$$

*where $\Lambda_T^+ = G\sqrt{T}\log^2(T+1)$.*

*Proof.* Let $\mathcal{A}$ be an instance of Jacobsen & Cutkosky (2022, Algorithm 4). Then via Jacobsen & Cutkosky (2022, Theorem 1) we have that for any sequence of linear losses $(g_t)_t$ satisfying $\|g_t\| \leq G$ for all $t$ and any $u \in \mathbb{R}^d$,

$$\sum_{t=1}^{T}\langle g_t, w_t - u\rangle = \mathcal{O}\left(\underbrace{\epsilon G + \|u\|\left[\sqrt{V_T \log\left(\frac{\|u\|\sqrt{V_T}\log^2(V_T/G^2)}{G\epsilon} + 1\right)}\right] \vee G\log\left(\frac{\|u\|\sqrt{V_T}\log^2(V_T/G^2)}{\epsilon} + 1\right)}_{=:B_T(u, g_{1:T})}\right)$$

where $V_T = G^2 + \sum_{t=1}^{T}\|g_t\|^2$. Hence, applying Proposition 2.3 and observing that $\|\widetilde{\ell}_t\| \leq 2d\|\ell_t\| \leq 2dG$ and $\|\ell_t - \widetilde{\ell}_t\| \leq \|\ell_t\| + \|\widetilde{\ell}_t\| \leq (2d+1)\|\ell_t\| \leq (2d+1)G$ for all $t$, letting $V_T = G^2 + \sum_{t=1}^{T}\|\ell_t\|^2$, $\widetilde{V}_T = \widetilde{G}^2 + \sum_{t=1}^{T}\|\widetilde{\ell}_t\|^2$ and $\widetilde{G} = 3dG$ we have

$$\mathbb{E}\left[R_T(u)\right] \leq \mathbb{E}\left[B_T(u, \widetilde{\ell}_{1:T}) + B_T\left(u, (\ell - \widetilde{\ell})_{1:T}\right)\right]$$

$$= \mathcal{O}\left(\mathbb{E}\left[\epsilon\widetilde{G} + \|u\|\left[\sqrt{\widetilde{V}_T \log\left(\frac{\|u\|\sqrt{\widetilde{V}_T}\log^2(\widetilde{V}_T/\widetilde{G}^2)}{\widetilde{G}\epsilon} + 1\right)}\right] \vee \widetilde{G}\log\left(\frac{\|u\|\sqrt{\widetilde{V}_T}\log^2(\widetilde{V}_T/\widetilde{G}^2)}{G\epsilon} + 1\right)\right]\right)$$

$$= \mathcal{O}\left(\mathbb{E}\left[\epsilon dG + \|u\|\sqrt{\widetilde{V}_T \log\left(\frac{\|u\|\sqrt{V_T}\log^2(V_T/G^2)}{G\epsilon} + 1\right)} + dG\|u\|\log\left(\frac{\|u\|\sqrt{V_T}\log^2(V_T/G^2)}{G\epsilon} + 1\right)\right]\right),$$

$$\tag{8}$$

where we've used the fact that $\sum_{t=1}^{T}\|\ell_t - \widetilde{\ell}_t\|^2 = \mathcal{O}(\sum_{t=1}^{T}\|\ell_t\|^2 + \|\widetilde{\ell}_t\|^2) = \mathcal{O}\left(d^2\sum_{t=1}^{T}\|\ell_t\|^2\right) = \mathcal{O}(d^2 V_T)$ via

Corollary 2.2. Now, if $\|u\|$ is $\mathcal{F}_0$-measurable, we have via tower rule and Jensen's inequality that

$$
\mathbb{E}\left[R_T(u)\right] = \mathcal{O}\left(\mathbb{E}\left[\epsilon dG + \|u\|\sqrt{\sum_{t=1}^{T}\mathbb{E}\left[\|\widetilde{\ell}_t\|^2|\mathcal{F}_0\right]\log\left(\frac{\|u\|\Lambda_T^+}{G\epsilon}+1\right)}\right.\right.
$$
$$
\left.\left.+ dG\|u\|\log\left(\frac{\|u\|\Lambda_T^+}{G\epsilon}+1\right)\right]\right)
$$
$$
= \mathcal{O}\left(\mathbb{E}\left[\epsilon dG + \|u\|\sqrt{d\sum_{t=1}^{T}\mathbb{E}\left[\|\ell_t\|^2|\mathcal{F}_0\right]\log\left(\frac{\|u\|\Lambda_T^+}{G\epsilon}+1\right)}\right.\right.
$$
$$
\left.\left.+ dG\|u\|\log\left(\frac{\|u\|\Lambda_T^+}{G\epsilon}+1\right)\right]\right),
$$

where we've bound $\sqrt{V_T}\log^2(1+V_T/G^2) \le G\sqrt{T}\log^2(1+T) =: \Lambda_T^+$ and used Corollary 2.2 and tower rule to bound $\mathbb{E}\left[\|\widetilde{\ell}_t\|^2|\mathcal{F}_0\right] = \mathbb{E}\left[\mathbb{E}\left[\|\widetilde{\ell}_t\|^2|\mathcal{F}_{t-1}\right]|\mathcal{F}_0\right] = \mathcal{O}\left(d\mathbb{E}\left[\|\ell_t\|^2|\mathcal{F}_0\right]\right)$. Otherwise, for arbitrary $u \in \mathbb{R}^d$ (possibly data-dependent), we can naively bound $\widetilde{V}_T = \mathcal{O}(d^2 V_T)$ to get

$$
\mathbb{E}\left[R_T(u)\right] = \mathcal{O}\left(\mathbb{E}\left[\epsilon dG + d\|u\|\sqrt{V_T\log\left(\frac{\|u\|\sqrt{V_T}\log^2(V_T/G^2)}{G\epsilon}+1\right)} + dG\|u\|\log\left(\frac{\|u\|\sqrt{V_T}\log^2(V_T/G^2)}{G\epsilon}+1\right)\right]\right).
$$

The bound in the theorem statement follows via change of variables $d\epsilon \mapsto \epsilon$. □

## C. Proof of Theorem 3.3

In this section we prove our result for expected dynamic regret. As in our expected static regret result, here we state a more general form of the theorem which allows for *partially*-oblivious settings where the comparator sequence $u_{1:T}$ is oblivious but the loss sequence $\ell_{1:T}$ may be adaptive, in which case the variance penalties become $\sum_{t=1}^{T}\mathbb{E}\left[\|\ell_t\|^2|\mathcal{F}_0\right]\|u_t\|$. Note that this captures the fully-oblivious setting from the statement in the main text as a special case, since when the losses are also $\mathcal{F}_0$ measurable we have $\mathbb{E}\left[\|\ell_t\|^2|\mathcal{F}_0\right] = \|\ell_t\|^2$ almost-surely. As discussed in the main text, the following result also shows a refined path-length adaptivity, scaling with the log-linear penalties $\sum_t \|u_t - u_{t-1}\|\log\left(\|u_t - u_{t-1}\|T^3/\epsilon + 1\right)$ instead of the worst-case $P_T\log\left(\max_t\|u_t\|T^3/\epsilon + 1\right)$ that would otherwise be obtained via direct application of Jacobsen & Cutkosky (2022, Theorem 4). As a result, our bound exhibits a dependence on $\max_t\|u_t\|$ only in the *lower-order term*, avoiding this worst-case factor entirely in the main term of the bound.

**Theorem 3.3.** *For any sequence $u_{1:T} = (u_t)_{t=1}^{T}$ in $\mathbb{R}^d$, PABLO applied with Algorithm 6 tuned with $\epsilon/d$ guarantees*

$$
\mathbb{E}\left[R_T(u_{1:T})\right] = \mathcal{O}\left(\mathbb{E}\left[dG(\epsilon + \max_t\|u_t\| + \Phi_T + P_T^\Phi) + d\sqrt{(\Phi_T + P_T^\Phi)\sum_{t=1}^{T}\|\ell_t\|^2\|u_t\|}\right]\right)
$$

*where we denote $\Phi_T = \|u_T\|\log\left(\frac{\|u_T\|T}{\epsilon}+1\right)$ and $P_T^\Phi = \sum_{t=2}^{T}\|u_t - u_{t-1}\|\log\left(\frac{4\|u_t - u_{t-1}\|T^3}{\epsilon}+1\right)$. Moreover, if the comparator sequence $u_{1:T}$ is $\mathcal{F}_0$ measurable, then*

$$
\mathbb{E}\left[R_T(u_{1:T})\right] = \mathcal{O}\left(\mathbb{E}\left[dG(\epsilon + \max_t\|u_t\| + \Phi_T + P_T^\Phi) + \sqrt{d(\Phi_T + P_T^\Phi)\sum_{t=1}^{T}\mathbb{E}\left[\|\ell_t\|^2|\mathcal{F}_0\right]\|u_t\|}\right]\right)
$$

*Proof.* Given any sequence $f_1, \ldots, f_T$ of $G$-Lipschitz convex losses, Algorithm 6 guarantees (Theorem E.7) that

$$
R_T^A(u_{1:T}) = \sum_{t=1}^{T}f_t(w_t) - f_t(u_t) \le 4G(\epsilon|\mathcal{S}| + M + \Phi_T + P_T^\Phi) + 2\sqrt{2(\Phi_T + P_T^\Phi)\sum_{t=1}^{T}\|g_t\|^2\|u_t\|},
$$

where $g_t \in \partial f_t(w_t)$, $M = \max_t \|u_t\|$, $\mathcal{S} = \left\{ \eta_i = \frac{2^i}{GT} \wedge \frac{1}{G} : i = 0, 1, \ldots \right\}$, and we denote $\Phi_T = \Phi\left(\|u_T\|, \frac{T}{\epsilon}\right)$ and $P_T^\Phi = \sum_{t=2}^T \Phi\left(\|u_t - u_{t-1}\|, \frac{4T^3}{\epsilon}\right)$ for $\Phi(x, \lambda) = x \log(\lambda x + 1)$. Hence, applying this algorithm to the $3dG$ Lipschitz loss sequences $(\widetilde{\ell}_t)_t$ and $(\ell_t - \widetilde{\ell}_t)_t$ and plugging into Proposition 2.3, we have

$$\mathbb{E}[R_T(u_{1:T})] = \mathcal{O}\left(\mathbb{E}\left[dG(\epsilon|\mathcal{S}| + M + \Phi_T + P_T^\Phi) + \sqrt{(\Phi_T + P_T^\Phi)\sum_{t=1}^T \|\widetilde{\ell}_t\|^2 \|u_t\|}\right. \right.$$
$$\left. \left. + \sqrt{(\Phi_T + P_T^\Phi)\sum_{t=1}^T \|\widetilde{\ell}_t - \ell_t\|^2 \|u_t\|}\right]\right).$$

Now via Corollary 2.2, we have that $\|\widetilde{\ell}_t\|^2 \le 4d^2\|\ell_t\|^2$ and likewise, $\|\ell_t - \widetilde{\ell}_t\|^2 \le 2\|\ell_t\|^2 + 2\|\widetilde{\ell}_t\|^2 = \mathcal{O}(d^2\|\ell_t\|^2)$, so we can always bound

$$\mathbb{E}[R_T(u_{1:T})] = \mathcal{O}\left(\mathbb{E}\left[dG(\epsilon|\mathcal{S}| + M + \Phi_T + P_T^\Phi) + d\sqrt{(\Phi_T + P_T^\Phi)\sum_{t=1}^T \|\ell_t\|^2 \|u_t\|}\right]\right).$$

On the other hand, if the comparator sequence is $\mathcal{F}_0$ measurable, we can apply Jensen's inequality and tower rule (twice) to get

$$\mathbb{E}[R_T(u_{1:T})] \le \mathcal{O}\left(\mathbb{E}\left[dG(\epsilon|\mathcal{S}| + M + \Phi_T + P_T^\Phi) + \sqrt{(\Phi_T + P_T^\Phi)\mathbb{E}\left[\sum_{t=1}^T \|\widetilde{\ell}_t\|^2 \Big| \mathcal{F}_0\right]\|u_t\|}\right. \right.$$
$$\left. \left. + \sqrt{(\Phi_T + P_T^\Phi)\sum_{t=1}^T \mathbb{E}\left[\|\widetilde{\ell}_t - \ell_t\|^2 \Big| \mathcal{F}_0\right]\|u_t\|}\right]\right)$$
$$\le \mathcal{O}\left(\mathbb{E}\left[dG(\epsilon|\mathcal{S}| + M + \Phi_T + P_T^\Phi) + \sqrt{d(\Phi_T + P_T^\Phi)\sum_{t=1}^T \mathbb{E}\left[\|\ell_t\|^2 \Big| \mathcal{F}_0\right]\|u_t\|}\right]\right),$$

where we've applied Corollary 2.2 to bound $\mathbb{E}\left[\|\widetilde{\ell}_t\|^2|\mathcal{F}_{t-1}\right] = \mathcal{O}(d\|\ell_t\|^2)$ and $\mathbb{E}\left[\|\widetilde{\ell}_t - \ell_t\|^2|\mathcal{F}_{t-1}\right] = \mathcal{O}\left(\mathbb{E}\left[\|\ell_t\|^2 + \|\widetilde{\ell}_t\|^2\right]\right) = \mathcal{O}(d\|\ell_t\|^2)$. Combining the bounds for the two cases gives the stated result.

$\square$

## D. High-probability Guarantees

### D.1. Reduction to OLO

In this section, we prove Proposition 4.1, which shows that with high-probability, the regret of PABLO scales with the regret of the OLO algorithm $\mathcal{A}$ deployed against the loss estimates $\widetilde{\ell}_t$, plus some additional stability terms. The result is framed in terms of a $(\mathcal{F}_t)_{t=0}^T$-adapted comparator sequence, though in Appendix D.3.1 we show that the results generalize straightforwardly to the fully-adaptive case using the same ghost-iterate trick as Proposition 2.3.

In the following theorem, we note that the lower-order $G\omega\sqrt{\log(16/\delta)}$ can be replaced by a $\delta$-independent penalty of $G\epsilon$ by setting $\omega = \epsilon/\sqrt{\log(16/\delta)}$, and the resulting trade-off can be bound in terms of $\log(1/\delta)$ since the remaining $\omega$-dependent terms are doubly-logarithmic in $1/\omega$ and can be naively bound as $\log\left(\frac{4}{\delta}\log\frac{C}{\omega}\right) = \mathcal{O}\left(\log\left(\frac{4\log(C/\epsilon) + 2\log(1/\delta)}{\delta}\right)\right) = \mathcal{O}\left(\log\left(\frac{\log(C/\epsilon)}{\delta}\right)\right)$ using $\log(1/\delta) \le \frac{1}{\delta}$ for $\delta > 0$. Hence, in the main text we drop the lower-order dependence on $G\omega\sqrt{\log(16/\delta)}$ but leave $\omega > 0$ free here for generality.

**Proposition 4.1.** *Let $\delta \in (0, 1/3]$, $\omega > 0$, and $\varepsilon^2 = \omega^2/T$. Let $u_{1:T} = (u_t)_{t=1}^T$ be an arbitrary $(\mathcal{F}_t)_{t=0}^T$-adapted sequence in $\mathbb{R}^d$. Then* `PABLO` *with $H_t = \frac{1}{d[\|w_t\|^2 \vee \varepsilon^2]} I_d$ guarantees that with probability at least $1 - 3\delta$,*

$$R_T(u_{1:T}) \leq \widetilde{R}_T^{\mathcal{A}}(u_{1:T}) + 2\Sigma_T(w_{1:T}) + \Sigma_T(u_{1:T}) + 3dGP_T + 2G\omega\sqrt{2\log\left(\frac{16}{\delta}\right)}$$

*where we define*

$$\Sigma_T(x_{1:T}) := 3G\sqrt{d\sum_{t=1}^T \|x_t\|^2 \log\left(\frac{4}{\delta}\left[\log_+\left(\frac{\sqrt{\sum_{t=1}^T \|x_t\|^2}}{\omega}\right) + 2\right]^2\right)}$$

$$+ 24dG \max\left(\omega, \max_{t \leq T} \|x_t\|\right)\log\left(\frac{28}{\delta}\left[\log_+\left(\frac{\max_t \|x_t\|}{\omega}\right) + 2\right]^2\right)$$

*Proof.* Recalling that Algorithm 1 plays $\widetilde{w}_t = w_t + H_t^{-\frac{1}{2}} s_t$ with $s_t$ drawn uniformly from $\{\pm e_i : i \in [d]\}$, we have

$$R_T(u_{1:T}) = \sum_{t=1}^T \langle \ell_t, \widetilde{w}_t - u_t \rangle$$

$$= \sum_{t=1}^T \langle \ell_t, w_t - u_t \rangle + \sum_{t=1}^T \left\langle \ell_t, H_t^{-\frac{1}{2}} s_t \right\rangle$$

$$= \underbrace{\sum_{t=1}^T \left\langle \widetilde{\ell}_t, w_t - u_t \right\rangle}_{\widetilde{R}_T^{\mathcal{A}}(u_{1:T})} + \underbrace{\sum_{t=1}^T \left\langle \ell_t - \widetilde{\ell}_t, w_t \right\rangle}_{\text{(A)}} + \underbrace{\sum_{t=1}^T \left\langle \widetilde{\ell}_t - \ell_t, u_t \right\rangle}_{\text{(B)}} + \underbrace{\sum_{t=1}^T \left\langle \ell_t, H_t^{-\frac{1}{2}} s_t \right\rangle}_{\text{(C)}}. \tag{9}$$

We proceed by bounding each of the noise terms (A), (B), and (C) with high probability.

**Bounding (A):** Let $X_t = \left\langle \ell_t - \widetilde{\ell}_t, w_t \right\rangle$ and observe by Proposition 2.1, tower rule, and the fact that $w_t$ is $\mathcal{F}_{t-1}$ measurable we have

$$\mathbb{E}[X_t|\mathcal{F}_{t-1}] = \mathbb{E}\left[\left\langle \ell_t - \widetilde{\ell}_t, w_t \right\rangle | \mathcal{F}_{t-1}\right] = 0.$$

Moreover, again by Proposition 2.1 we have

$$\mathbb{E}\left[X_t^2|\mathcal{F}_{t-1}\right] \leq \mathbb{E}\left[\|\ell_t - \widetilde{\ell}_t\|^2 \|w_t\|^2 | \mathcal{F}_{t-1}\right]$$

$$\leq \|w_t\|^2\left(\mathbb{E}\left[\|\widetilde{\ell}_t\|^2 | \mathcal{F}_{t-1}\right] - \|\ell_t\|^2\right) \leq 2d\|w_t\|^2\|\ell_t\|^2$$

$$\leq 2dG^2\|w_t\|^2.$$

and likewise,

$$|X_t| \leq \|\ell_t - \widetilde{\ell}_t\|\|w_t\| \leq (\|\ell_t\| + \|\widetilde{\ell}_t\|)\|w_t\| \leq (1 + 2d)\|\ell_t\|\|w_t\| \leq 3dG\|w_t\|$$

almost-surely. Therefore, applying Theorem I.4 with $\sigma_t^2 = 2dG^2\|w_t\|^2$ and $b_t = 3dG\|w_t\|$, we have that with probability at least $1 - \delta$,

$$\sum_{t=1}^T \left\langle \ell_t - \widetilde{\ell}_t, w_t \right\rangle \leq 2G\sqrt{2d\sum_{t=1}^T \|w_t\|^2 \log\left(\frac{4}{\delta}\left[\log_+\left(G\sqrt{2d\sum_{t=1}^T \|w_t\|^2/(2\nu^2)}\right) + 2\right]^2\right)}$$

$$+ 8\max\left(\nu, \max_{t \leq T} 3dG\|w_t\|\right)\log\left(\frac{28}{\delta}\left[\log_+\left(3dG\max_t \|w_t\|/\nu\right) + 2\right]^2\right)$$

and hence setting $\nu = 3dG\omega$,

$$
\leq 3G\sqrt{d\sum_{t=1}^{T}\|w_t\|^2 \log\left(\frac{4}{\delta}\left[\log_+\left(\frac{\sqrt{2}dG}{3\sqrt{2}dG\omega}\sqrt{\sum_{t=1}^{T}\|w_t\|^2}\right)+2\right]^2\right)}
$$

$$
+ 24dG\,\max\left(\omega, \max_{t\leq T}\|w_t\|\right)\log\left(\frac{28}{\delta}\left[\log_+\left(\max_t\|w_t\|/\omega\right)+2\right]^2\right)
$$

$$
\leq 3G\sqrt{d\sum_{t=1}^{T}\|w_t\|^2 \log\left(\frac{4}{\delta}\left[\log_+\left(\frac{\sqrt{\sum_{t=1}^{T}\|w_t\|^2}}{\omega}\right)+2\right]^2\right)}
$$

$$
+ 24dG\,\max\left(\omega, \max_{t\leq T}\|w_t\|\right)\log\left(\frac{28}{\delta}\left[\log_+\left(\frac{\max_t\|w_t\|}{\omega}\right)+2\right]^2\right)
$$

$$
=: \Sigma_T(w_{1:T})
$$

**Bounding $\boxed{\text{B}}$:** For $\mathcal{F}_t$-measurable $u_t$, we could have correlations between $u_t$ and $\ell_t - \widetilde{\ell}_t$, so we first shift the comparator sequence by one index:

$$
\sum_{t=1}^{T}\left\langle\widetilde{\ell}_t - \ell_t, u_t\right\rangle = \sum_{t=1}^{T}\left\langle\widetilde{\ell}_t - \ell_t, u_{t-1}\right\rangle + \sum_{t=1}^{T}\left\langle\widetilde{\ell}_t - \ell_t, u_t - u_{t-1}\right\rangle
$$

$$
\leq \sum_{t=1}^{T}\left\langle\widetilde{\ell}_t - \ell_t, u_{t-1}\right\rangle + 3dGP_T
$$

where we've used Corollary 2.2 to bound $\|\ell_t - \widetilde{\ell}_t\| \leq G + \|\widetilde{\ell}_t\| \leq G(1 + 2d) \leq 3dG$, and defined $u_0 = \mathbf{0}$. Now applying the same arguments as $\boxed{\text{A}}$, the first summation can be bound with probability at least $1 - \delta$ as

$$
\sum_{t=1}^{T}\left\langle\widetilde{\ell}_t - \ell_t, u_{t-1}\right\rangle \leq 3G\sqrt{d\sum_{t=1}^{T}\|u_{t-1}\|^2 \log\left(\frac{4}{\delta}\left[\log_+\left(\frac{\sqrt{\sum_{t=1}^{T}\|u_{t-1}\|^2}}{2\omega}\right)+2\right]^2\right)}
$$

$$
+ 16dG\,\max\left\{\omega, \max_{t\leq T-1}\|u_t\|\right\}\log\left(\frac{28}{\delta}\left[\log_+\left(\frac{\max_{t\leq T-1}\|u_t\|}{\omega}\right)+2\right]^2\right)
$$

$$
\leq 3G\sqrt{d\sum_{t=1}^{T}\|u_t\|^2 \log\left(\frac{4}{\delta}\left[\log_+\left(\frac{\sqrt{\sum_{t=1}^{T}\|u_t\|^2}}{2\omega}\right)+2\right]^2\right)}
$$

$$
+ 24dG\,\max\left\{\omega, \max_{t\leq T}\|u_t\|\right\}\log\left(\frac{28}{\delta}\left[\log_+\left(\frac{\max_{t\leq T}\|u_t\|}{\omega}\right)+2\right]^2\right)
$$

$$
= \Sigma_T(u_{1:T}),
$$

hence, with probability at least $1 - \delta$,

$$
\boxed{\text{B}} \leq \Sigma_T(u_{1:T}) + 3dGP_T.
$$

**Bounding $\boxed{\text{C}}$:** By definition we have $H_t^{-\frac{1}{2}} = \sqrt{d}[\|w_t\| \vee \varepsilon]I_d$ and $X_t := \left\langle\ell_t, H_t^{-\frac{1}{2}}s_t\right\rangle = \sqrt{d}[\|w_t\| \vee \varepsilon]\langle\ell_t, s_t\rangle$. Hence, since $s_t$ is drawn uniform random from $\{\pm e_i : i \in [d]\}$, we have $\mathbb{E}[X_t|\mathcal{F}_{t-1}] = 0$ and

$$
\mathbb{E}\left[X_t^2|\mathcal{F}_{t-1}\right] = \mathbb{E}\left[d[\|w_t\|^2 \vee \varepsilon^2]\ell_t^\top s_t s_t^\top \ell_t|\mathcal{F}_{t-1}\right] = d[\|w_t\|^2 \vee \varepsilon^2]\|\ell_t\|^2\frac{1}{d} = [\|w_t\|^2 \vee \varepsilon^2]\|\ell_t\|^2 \leq [\|w_t\|^2 \vee \varepsilon^2]G^2
$$

and $|X_t| \le \sqrt{d}[\|w_t\| \vee \varepsilon]\|\ell_t\| \le \sqrt{d}G(\|w_t\| \vee \varepsilon)$ almost surely. Thus, we can again apply Theorem I.4 with $\sigma_t^2 = G^2[\|w_t\|^2 \vee \varepsilon^2]$ and $b_t = \sqrt{d}[\|w_t\| \vee \varepsilon]$ to get

$$\boxed{C} \le 2G\sqrt{\sum_{t=1}^{T}(\|w_t\|^2 + \varepsilon^2)\log\left(\frac{4}{\delta}\left[\log_+\left(G\sqrt{\sum_{t=1}^{T}\frac{\|w_t\|^2 + \varepsilon^2}{2\nu^2}}\right) + 2\right]^2\right)}$$

$$+ 8\max\left\{\nu, \sqrt{d}G[\max_t \|w_t\| \vee \varepsilon]\right\}\log\left(\frac{28}{\delta}\left[\log_+\left(\frac{\sqrt{d}G[\max_t \|w_t\| \vee \varepsilon]}{\nu}\right) + 2\right]^2\right)$$

and recalling $\varepsilon^2 = \omega^2/T$,

$$\le 2G\sqrt{\omega^2 + \sum_{t=1}^{T}\|w_t\|^2 \log\left(\frac{4}{\delta}\left[\log_+\left(G\sqrt{\frac{\omega^2 + \sum_{t=1}^{T}\|w_t\|^2}{2\nu^2}}\right) + 2\right]^2\right)}$$

$$+ 8\max\left\{\nu, \sqrt{d}G[\max_t \|w_t\| \vee \tfrac{\omega}{\sqrt{T}}]\right\}\log\left(\frac{28}{\delta}\left[\log_+\left(\frac{\sqrt{d}G[\max_t \|w_t\| \vee \tfrac{\omega}{\sqrt{T}}]}{\nu}\right) + 2\right]^2\right)$$

and setting $\nu = \sqrt{d}G\omega$,

$$\le 2G\sqrt{\omega^2 + \sum_{t=1}^{T}\|w_t\|^2 \log\left(\frac{4}{\delta}\left[\log_+\left(\sqrt{\frac{\omega^2 + \sum_{t=1}^{T}\|w_t\|^2}{2d\omega^2}}\right) + 2\right]^2\right)}$$

$$+ 8\max\left\{\sqrt{d}G\omega, \sqrt{d}G[\max_t \|w_t\| \vee \tfrac{\omega}{\sqrt{T}}]\right\}\log\left(\frac{28}{\delta}\left[\log_+\left(\frac{\sqrt{d}G[\max_t \|w_t\| \vee \tfrac{\omega}{\sqrt{T}}]}{\sqrt{d}G\omega}\right) + 2\right]^2\right)$$

$$\le \underbrace{2G\sqrt{2\left[\omega^2 \vee \sum_{t=1}^{T}\|w_t\|^2\right]\log\left(\frac{4}{\delta}\left[\log_+\left(\sqrt{\frac{\omega^2 \vee \sum_{t=1}^{T}\|w_t\|^2}{d\omega^2}}\right) + 2\right]^2\right)}}_{\boxed{V}}$$

$$+ 8\sqrt{d}G\max\left\{\omega, [\max_t \|w_t\| \vee \tfrac{\omega}{\sqrt{T}}]\right\}\log\left(\frac{28}{\delta}\left[\log_+\left(\frac{\sqrt{d}G[\max_t \|w_t\| \vee \tfrac{\omega}{\sqrt{T}}]}{\sqrt{d}G\omega}\right) + 2\right]^2\right)$$

$$\le \boxed{V} + 8\sqrt{d}G\max\left\{\omega, \max_t \|w_t\|\right\}\log\left(\frac{28}{\delta}\left[\log_+\left(\frac{\max_t \|w_t\|}{\omega}\right) + 2\right]^2\right),$$

where the last line observes that if $\max_t \|w_t\| \le \omega/\sqrt{T}$ then the $\log_+$ in the last term simplifies to $\log_+\left(\frac{1}{\sqrt{T}}\right) = 0$. Now consider two cases: first, if $\sum_{t=1}^{T}\|w_t\|^2 \le \omega^2$, we also have $\max_t \|w_t\| \le \omega$ and we can bound

$$\boxed{V} \le 2G\omega\sqrt{2\log\left(\frac{4}{\delta}\left[\log_+\left(\sqrt{\frac{1}{d}}\right) + 2\right]^2\right)}$$

$$\le 2G\omega\sqrt{2\log\left(\frac{16}{\delta}\right)}$$

and otherwise, when $\sum_{t=1}^{T} \|w_t\|^2 \geq \omega^2$, we also have $\frac{\omega}{\sqrt{T}} \leq \sqrt{\sum_{t=1}^{T} \|w_t\|^2 / T} \leq \max_t \|w_t\|$ and

$$
\text{(V)} \leq 2G \sqrt{2 \sum_{t=1}^{T} \|w_t\|^2 \log \left( \frac{4}{\delta} \left[ \log_+ \left( \sqrt{\frac{\sum_{t=1}^{T} \|w_t\|^2}{d\omega^2}} \right) + 2 \right]^2 \right)}
$$

hence, combining these two cases, with probability at least $1 - \delta$ we have

$$
\begin{aligned}
\text{(C)} &\leq \text{(V)} + 8\sqrt{d}G \max \left\{ \omega, \max_t \|w_t\| \right\} \log \left( \frac{28}{\delta} \left[ \log_+ \left( \frac{\max_t \|w_t\|}{\omega} \right) + 2 \right]^2 \right) \\
&\leq 2G\omega\sqrt{2 \log \left( \frac{16}{\delta} \right)} + 2G \sqrt{2 \sum_{t=1}^{T} \|w_t\|^2 \log \left( \frac{4}{\delta} \left[ \log_+ \left( \sqrt{\frac{\sum_{t=1}^{T} \|w_t\|^2}{d\omega^2}} \right) + 2 \right]^2 \right)} \\
&\quad + 8\sqrt{d}G \max \left\{ \omega, \max_t \|w_t\| \right\} \log \left( \frac{28}{\delta} \left[ \log_+ \left( \frac{\max_t \|w_t\|}{\omega} \right) + 2 \right]^2 \right) \\
&\leq 2G\omega\sqrt{2 \log \left( \frac{16}{\delta} \right)} + \Sigma_T(w_{1:T})
\end{aligned}
$$

Thus, combining the bounds for $\text{(A)}$, $\text{(B)}$, and $\text{(C)}$, we have with probability at least $1 - 3\delta$

$$
R_T(u_{1:T}) \leq \widetilde{R}_T^{\mathcal{A}}(u_{1:T}) + 2\Sigma_T(w_{1:T}) + \Sigma_T(u_{1:T}) + 3dGP_T + 2G\omega\sqrt{2 \log \left( \frac{16}{\delta} \right)}
$$

$\square$

## D.2. High-probability Static Regret Guarantees

---

**Algorithm 2** Sub-exponential Noisy Gradients with Optimistic Online Learning

---

**Require:** $\mathbb{E}[g_t] = \nabla \ell_t(w_t)$, $\|g_t\| \leq b$, $\mathbb{E}[\|g_t\|^2 | w_t] \leq \sigma^2$ almost surely. Algorithms $\mathcal{A}_1$ and $\mathcal{A}_2$ with domains $\mathbb{R}^d$ and $\mathbb{R}_{\geq 0}$. Time horizon $T$, $0 < \delta \leq 1$.
**Initialize:** Constants $\{c_1, c_2, p_1, p_2, \alpha_1, \alpha_2\}$, $H = c_1 p_1 + c_2 p_2$
**for** $t = 1 : T$ **do**
    Receive $x_t'$ from $\mathcal{A}_1$, $y_t'$ from $\mathcal{A}_2$
    Rescale $x_t = x_t'/(b + H)$, $y_t = y_t'/(H(b + H))$
    Solve for $w_t$: $w_t = x_t - y_t \nabla \varphi_t(w_t)$
    Play $w_t$ and suffer loss $\ell_t(w_t)$
    Receive loss $g_t$ with $\mathbb{E}[g_t] \in \partial \ell_t(w_t)$
    Compute $\varphi_t(w) = r_t(w; c_1, \alpha_1, p_1) + r_t(w; c_2, \alpha_2, p_2)$ and $\nabla \varphi_t(w_t)$
    Send $(g_t + \nabla \varphi_t(w_t))/(b + H)$ to $\mathcal{A}_1$
    Send $-\langle g_t + \nabla \varphi_t(w_t), \nabla \varphi_t(w_t) \rangle / H(b + H)$ to $\mathcal{A}_2$
**end for**

---

In this section we briefly review the approach developed by Zhang & Cutkosky (2022) for developing high-probability guarantees in unconstrained settings.

In unconstrained settings, the usual Martingale concentration arguments alone are not enough to control the bias terms $\sum_{t=1}^{T} \left\langle \ell_t - \widetilde{\ell}_t, w_t \right\rangle$, since they will lead to terms on the order of $\widetilde{\mathcal{O}}(\sqrt{\sum_{t=1}^{T} \|w_t\|^2})$, which could be arbitrarily large in an unbounded domain. The main idea is of Zhang & Cutkosky (2022) is to add an additional composite penalty $\varphi_t$ to the update, which introduces an extra term $\sum_{t=1}^{T} \varphi_t(u) - \varphi_t(w_t)$ into the regret bound; with this, the goal is to choose $\varphi_t$ in such a way

that $-\sum_{t=1}^{T} \varphi_t(w_t)$ is large enough to cancel with the $\|w_t\|$-dependent terms left over from the Martingale concentration argument, while also ensuring that $\sum_{t=1}^{T} \varphi_t(u)$ is not too large. Zhang & Cutkosky (2022) provide a Huber-like penalty which satisfies both of these conditions (see Lemma E.4).

The difficulty with the above approach is that by introducing the composite penalty, we change the OLO learner's feedback: on round $t$, the learner's feedback becomes $\widetilde{g}_t = g_t + \nabla\varphi_t(w_t)$ instead of just $g_t \in \partial\ell_t(w_t)$, and the $\nabla\varphi_t(w_t)$ dependence would itself lead to $\|w_t\|$-dependent penalties in the bound. This issue can be fixed using an optimistic update by setting hints $h_t = \nabla\varphi_t(w_t)$, so that the usual $\sum_{t=1}^{T} \|\widetilde{g}_t\|^2$ penalties in the final bound become $\sum_{t=1}^{T} \|\widetilde{g}_t - h_t\|^2 = \sum_{t=1}^{T} \|g_t\|^2$. Note that setting $h_t$ this way requires solving an implicit equation for $w_t = x_t - y_t\nabla\varphi_t(w_t)$. The full pseudocode is provided in Algorithm 2, and it makes the following guarantee. [6]

**Theorem D.1.** *(Zhang & Cutkosky, 2022, Theorem 3) Suppose $\{g_t\}$ are stochastic subgradients such that $\mathbb{E}[g_t] \in \partial\ell_t(w_t)$, $\|g_t\| \le b$, and $\mathbb{E}[\|g_t\|^2 | w_t] \le \sigma$ almost surely for all $t$. Set the constants for $\varphi_t(w)$ as*

$$c_1 = 2\sigma\sqrt{\log\left(\frac{32}{\delta}\left[\log\left(2^{T+1}\right) + 2\right]^2\right)}, \ c_2 = 32b\log\left(\frac{224}{\delta}\left[\log\left(1 + \frac{b}{\sigma}2^{T+2}\right) + 2\right]^2\right)$$

$$p_1 = 2, \ p_2 = \log(T), \ \alpha_1 = \epsilon/c_1, \ \alpha_2 = \epsilon\sigma/(4b(b+H)),$$

*where $H = c_1 p_1 + c_2 p_2$, and $\|\nabla\varphi_t(w_t)\| \le H$. Then with probability at least $1 - \delta$, Algorithm 2 guarantees*

$$R_T(u) = \widetilde{\mathcal{O}}\left[\epsilon\log(T/\delta) + b\|u\|\log(T/\delta) + \|u\|\sigma\sqrt{T\log(T/\delta)}\right]$$

Note that via Proposition 2.1, we have that $\|\widetilde{\ell}_t\|^2 \le 4d^2G^2$, and $\mathbb{E}[\|\widetilde{\ell}_t\|^2 | w_t] \le 2dG^2$. Hence, to achieve static regret guarantees we can immediately apply the algorithm of (Zhang & Cutkosky, 2022) to get the following guarantee. The only modification we need to make is that we should multiply the $c_1$ and $c_2$ in Theorem D.1 by 2, to account for the fact that we have an extra $\|w_t\|$-dependent concentration penalty coming from our perturbation $\left\langle \ell_t, H_t^{\frac{1}{2}} s_t \right\rangle$.

**Theorem 4.3.** *Let `PABLO` be implemented with Zhang & Cutkosky (2022, Algorithm 1), and $\delta \in (0, 1/3]$. Then for any $\mathcal{F}_0$-measurable $u \in \mathbb{R}^d$, with probability at least $1 - 3\delta$,*

$$R_T(u) \le \widetilde{\mathcal{O}}\left(dG(\epsilon + \|u\|)\log\left(\frac{T}{\delta}\right) + G\|u\|\sqrt{dT\log\left(\frac{T}{\delta}\right)}\right).$$

### D.3. High-Probability Dynamic Regret Guarantees

In this section we provide an algorithm, detailed in Algorithm 3, which achieves the bound stated in Theorem 4.4. The algorithm can be understood as a special case of a more general algorithm characterized in Theorem E.3, which we develop separately in Appendix E.1. The approach is similar to the one described in the previous section: the algorithm adds additional composite penalties $\varphi_t$ to the update, which lead to additional factors of $\sum_{t=1}^{T} \varphi_t(u_t) - \varphi_t(w_t)$ in the regret, which are used to cancel out the $\|w_t\|$-dependent penalties from Proposition 4.1. In particular, Algorithm 3 is constructed by combining several instances of a base algorithm (Algorithm 4), each applied with different learning rates $\eta$ and with the Huber-like penalties discussed in Appendix E.1.1, which ensure a negative penalty of $\sum_{t=1}^{T} -\varphi_t(w_t) = \widetilde{\mathcal{O}}\left(-\left(\sum_{t=1}^{T} \|w_t\|^p\right)^{1/p}\right)$ appears in the bound at the expense of a problem-dependent penalty of $\sum_{t=1}^{T} \varphi_t(u_t) = \widetilde{\mathcal{O}}\left(\left(\sum_{t=1}^{T} \|u_t\|^p\right)^{1/p}\right)$. Applying `PABLO` with Algorithm 3 as the base algorithm then leads to the following high-probability regret guarantee in the uBLO setting.

---

[6] The theorem statement in Zhang & Cutkosky (2022) is given in terms of $\widetilde{\mathcal{O}}(\log(1/\delta))$, which hides a $T$ dependency inside the logarithm, as can be observed from the parameter settings for $c_1$ and $c_2$ in Theorem D.1. We highlight this $T$ dependence in our re-statement to facilitate a more direct comparison to the high-probability bounds of Zimmert & Lattimore (2022).

**Algorithm 3** Dynamic Algorithm for Heavy-tailed Noise

---

**Require:** $\|\widetilde{\ell}_t\| \leq L$ for all $t$, $\mathbb{E}[\|\widetilde{\ell}_t\|^2 | \mathcal{F}_{t-1}] \leq \sigma^2$ almost surely
**Input:** $0 < \delta \leq 1/3$, constants $\{c_1, c_2, p_1, p_2, \alpha_1, \alpha_2\}$
**Initialize:** $H = c_1 p_1 + c_2 p_2$, step-sizes $\mathcal{S} = \left\{ \eta_i \leq \frac{1}{L+H}, i = 0, 1, \ldots \right\}$, algorithms $\mathcal{A}_x^\eta$ and $\mathcal{A}_y^\eta$ implementing
Algorithm 5 on $\mathbb{R}^d$ and $\mathbb{R}_{\geq 0}$ respectively for each $\eta \in \mathcal{S}$
**for** $t = 1 : T$ **do**
  **for** $\eta \in \mathcal{S}$ **do**
    Get $x_t \in \mathbb{R}^d$ from $\mathcal{A}_x^\eta$ and $y_t \in R_{\geq 0}$ from $\mathcal{A}_y^\eta$
    Solve for $w_t^\eta = x_t - y_t \eta \nabla \varphi_t^\eta(w_t^\eta)$
      where $\varphi_t^\eta(w) = r_t^\eta(w; c_1, \alpha_1, p_1) + r_t^\eta(w; c_2, \alpha_2, p_2)$ for $r_t^\eta(\cdot; c, \alpha, p)$ in Equation (19), defined w.r.t. $(w_\tau^\eta)_{\tau \in [t]}$
  **end for**
  Play $w_t = \sum_{\eta \in \mathcal{S}} w_t^\eta$, observe $\widetilde{\ell}_t \in \partial f_t(w_t)$
  Send $\widetilde{\ell}_t + \nabla \varphi_t^\eta(w_t^\eta)$ to $\mathcal{A}_x^\eta$ for each $\eta \in \mathcal{S}$
  Send $-\eta \left\langle \widetilde{\ell}_t + \nabla \varphi_t^\eta(w_t^\eta), \nabla \varphi_t^\eta(w_t^\eta) \right\rangle$ to $\mathcal{A}_y^\eta$ for each $\eta \in \mathcal{S}$
**end for**

---

**Theorem 4.4.** *Let $\mathcal{A}$ be an instance of Algorithm 3 applied with $L = 2dG$, $\sigma^2 = 4dG^2$, and hyperparameters*

$$c_1 = 6G\sqrt{d|\mathcal{S}| \log\left(\frac{4}{\delta}\left[T + \log_+\left(\frac{4\epsilon\sqrt{|\mathcal{S}|}}{\omega}\right)\right]^2\right)}$$

$$c_2 = 48dG \log\left(\frac{28}{\delta}\left[T + \log_+\left(\frac{2\epsilon\sqrt{|\mathcal{S}|}}{\omega}\right)\right]^2\right)$$

$$\alpha_1 = \epsilon, \qquad \alpha_2 = \omega, \qquad p_1 = 2, \qquad p_2 = \log(T+1).$$

*Let $u_{1:T} = (u_1, \ldots, u_T)$ be an arbitrary $(\mathcal{F}_t)_{t=0}^T$-adapted sequence in $\mathbb{R}^d$. Then with probability at least $1 - 4\delta$, PABLO applied with $\mathcal{A}$ guarantees*

$$R_T(u_{1:T}) \leq \min\left\{ 8d\sqrt{(\Phi_T + P_T^\Phi)\sum_{t=1}^T \|\ell_t\|^2 \|u_t\|}, \; 8G\sqrt{2M(\Phi_T + P_T^\Phi)}\left[\sqrt{dT} + d\sqrt{\log(1/\delta)}\right]\right\}$$

$$+ 2G\sqrt{d\sum_{t=1}^T \|u_t\|^2 \log\left(\frac{4}{\delta}\left[\log_+\left(\frac{\sqrt{\sum_{t=1}^T \|u_t\|^2}}{\omega}\right) + 2\right]^2\right)}$$

$$+ 4c_1\sqrt{\left(\epsilon^2 + \sum_{t=1}^T \|u_t\|^2\right)\log\left(e + \frac{e\sum_{t=1}^T \|u_t\|^2}{\epsilon^2}\right)}$$

$$+ 3c_2 \log^2(T+1) \max\{\omega, M\}\left[\log_+\left(\frac{3M}{\omega}\right) + 3\right]$$

$$+ 24dG \max\left\{\omega, \max_{t \leq T} \|u_t\|\right\} \log\left(\frac{28}{\delta}\left[\log_+\left(\frac{\max_{t \leq T} \|u_t\|}{\omega}\right) + 2\right]^2\right)$$

$$+ 32(dG + H)(\epsilon|\mathcal{S}| + M + \Phi_T + P_T^\Phi)$$

$$+ c_1 \epsilon + c_2 \omega + 3dGP_T + 2G\omega\sqrt{2\log\left(\frac{16}{\delta}\right)}$$

*where $M = \max_t \|u_t\|$, $\Phi_T = \|u_T\| \log\left(\frac{\|u_T\|T}{\epsilon} + 1\right)$ and $P_T^\Phi = \sum_{t=2}^T \|u_t - u_{t-1}\| \log\left(\frac{4\|u_t - u_{t-1}\|T^3}{\epsilon} + 1\right)$.*

*Proof.* By Proposition 4.1, we have that with probability at least $1 - 3\delta$,

$$R_T(u_{1:T}) \leq \widetilde{R}_T^{\mathcal{A}}(u_{1:T}) + 2\Sigma_T(w_{1:T}) + \Sigma_T(u_{1:T}) + 3dGP_T + 2G\omega\sqrt{2\log\left(\frac{16}{\delta}\right)} \tag{10}$$

where $\widetilde{R}_T^{\mathcal{A}}(u_{1:T}) = \sum_{t=1}^T \langle \widetilde{\ell}_t, w_t - u_t \rangle$ and

$$\Sigma_T(x_{1:T}) := 3G\sqrt{d\sum_{t=1}^T \|x_t\|^2 \log\left(\frac{4}{\delta}\left[\log_+\left(\frac{\sqrt{\sum_{t=1}^T \|x_t\|^2}}{\omega}\right) + 2\right]^2\right)}$$

$$+ 24dG\max\left(\omega, \max_{t\leq T}\|x_t\|\right)\log\left(\frac{28}{\delta}\left[\log_+\left(\frac{\max_t\|x_t\|}{\omega}\right) + 2\right]^2\right) . \tag{11}$$

To bound the regret $\widetilde{R}_T^{\mathcal{A}}(u_{1:T})$, we note that Algorithm 3 is an instance of the algorithm characterized in Theorem E.3 applied with a specific composite penalties $\varphi_t$ and with Lipschitz constant $L = 2dG$. In particular, for each $\eta \in \mathcal{S}$, let $\mathcal{A}_\eta$ denote an instance of Algorithm 4 applied with composite penalty $\varphi_t^\eta(w) = r_t^\eta(w; c_1, \alpha_1, p_1) + r_t^\eta(w; c_2, \alpha_2, p_2)$ (for parameters $\{c_1, c_2, \alpha_1, \alpha_2, p_1, p_2\}$ to be determined) where $r_t^\eta$ is defined by

$$r_t^\eta(w; c, \alpha, p) = \begin{cases} c\left(p\|w\| - (p-1)\|w_t^\eta\|\right)\frac{\|w_t^\eta\|^{p-1}}{\left(\alpha^p + \sum_{s=1}^t \|w_s^\eta\|^p\right)^{1-1/p}} & \text{if } \|w\| > \|w_t^\eta\| \\ c\frac{\|w\|^p}{\left(\alpha^p + \sum_{s=1}^t \|w_t^\eta\|^p\right)^{1-1/p}} & \text{if } \|w\| \leq \|w_t^\eta\| \end{cases},$$

and $w_t^\eta$ is the output of $\mathcal{A}_\eta$ on round $t$. Algorithm 3 is constructed by combining the outputs of the $\mathcal{A}_\eta$ by playing $w_t = \sum_{\eta \in \mathcal{S}} w_t^\eta$ on round $t$, and its regret guarantee is given by Theorem E.3. Now observe that via Corollary 2.2, we have that $\|\widetilde{\ell}_t\|^2 \leq 4d^2\|\ell_t\|^2$ almost-surely, so we can apply Theorem E.3 with Lipschitz constant $L = 2dG \geq \|\widetilde{\ell}_t\|$ to bound

$$\widetilde{R}_T^{\mathcal{A}}(u_{1:T}) \leq 16(2dG + H)\left(\epsilon|\mathcal{S}| + M + \Phi(\|u_T\|, \tfrac{T}{\epsilon}) + P_T^\Phi\left(\tfrac{4T^3}{\epsilon}\right)\right) + 4\sqrt{\left[\Phi\left(\|u_T\|, \tfrac{T}{\epsilon}\right) + P_T^\Phi\left(\tfrac{4T^3}{\epsilon}\right)\right]\sum_{t=1}^T \|g_t\|^2\|u_t\|}$$

$$+ \max_{\eta^* \in \mathcal{S}}\sum_{\eta \in \mathcal{S}}\sum_{t=1}^T \varphi_t^\eta\left(u_t\mathbb{I}(\eta = \eta^*)\right) - \varphi_t^\eta(w_t^\eta),$$

where $M = \max_t \|u_t\|$, $\Phi(x, \lambda) = x\log(\lambda x + 1)$, and $P_T^\Phi(\lambda) = \sum_{t=2}^T \Phi(\|u_t - u_{t-1}\|, \lambda)$. The terms in the last line can be bound by observing that for any $\eta \in \mathcal{S}$ we have $\varphi_t^\eta(0) = 0$ and that via Lemma E.5 we have

$$\sum_{t=1}^T \varphi_t^\eta(u_t) \leq 4c_1\sqrt{\left(\alpha_1^2 + \sum_{t=1}^T \|u_t\|^2\right)\log\left(e + \frac{e\sum_{t=1}^T \|u_t\|^2}{\alpha_1^2}\right)}$$

$$+ 3c_2\log^2(T+1)\max\{\alpha_2, M\}\left[\log_+(3M/\alpha_2) + 3\right] .$$

Plugging these in above yields

$$\widetilde{R}_T^{\mathcal{A}}(u_{1:T}) \leq 4\sqrt{\left[\Phi\left(\|u_T\|, \tfrac{T}{\epsilon}\right) + P_T^\Phi\left(\tfrac{4T^3}{\epsilon}\right)\right]\sum_{t=1}^T \|g_t\|^2\|u_t\|}$$

$$+ 4c_1\sqrt{\left(\alpha_1^2 + \sum_{t=1}^T \|u_t\|^2\right)\log\left(e + \frac{e\sum_{t=1}^T \|u_t\|^2}{\alpha_1^2}\right)}$$

$$+ 16(2dG + H)\left(\epsilon|\mathcal{S}| + M + \Phi(\|u_T\|, \tfrac{T}{\epsilon}) + P_T^\Phi\left(\tfrac{4T^3}{\epsilon}\right)\right)$$

$$+ 3c_2\log^2(T+1)\max\{\alpha_2, M\}\left[\log_+(3M/\alpha_2) + 3\right]$$

$$- \sum_{\eta \in \mathcal{S}}\sum_{t=1}^T \varphi_t(w_t^\eta),$$

and plugging this back into the full regret bound in Equation (10) we have

$$R_T(u_{1:T}) \leq 16(2dG + H)\left(\epsilon|\mathcal{S}| + M + \Phi_T + P_T^\Phi\right)$$

$$+ 4\sqrt{\left[\Phi_T + P_T^\Phi\right] \sum_{t=1}^T \|\widetilde{\ell}_t\|^2 \|u_t\|}$$

$$+ 4c_1\sqrt{\left(\alpha_1^2 + \sum_{t=1}^T \|u_t\|^2\right) \log\left(e + \frac{e\sum_{t=1}^T \|u_t\|^2}{\alpha_1^2}\right)}$$

$$+ 3c_2 \log^2(T+1) \max\{\alpha_2, M\}\left[\log_+(3M/\alpha_2) + 3\right]$$

$$+ \Sigma_T(u_{1:T}) + 3dGP_T + 2\omega G\sqrt{2\log\left(\frac{16}{\delta}\right)}$$

$$+ 2\Sigma_T(w_{1:T}) - \sum_{\eta\in\mathcal{S}}\sum_{t=1}^T \varphi_t^\eta(w_t^\eta).$$

Hence, The main terms to control are the $\|w_t\|$-dependent terms in the last line, $2\Sigma_T(w_{1:T}) - \sum_{\eta\in\mathcal{S}}\sum_{t=1}^T \varphi_t^\eta(w_t^\eta)$, and $4\sqrt{(\Phi_T + P_T^\Phi)\sum_{t=1}^T \|\widetilde{\ell}_t\|^2 \|u_t\|}$ from the regret of $\mathcal{A}$. We begin by bounding the latter term, and in particular we will bound it in two different ways. First, by Corollary 2.2, we have that $\|\widetilde{\ell}_t\|^2 \leq 4d^2\|\ell_t\|^2$ almost-surely, so

$$4\sqrt{(\Phi_T + P_T^\Phi)\sum_{t=1}^T \|\widetilde{\ell}_t\|^2 \|u_t\|} \leq 8d\sqrt{(\Phi_T + P_T^\Phi)\sum_{t=1}^T \|\ell_t\|^2 \|u_t\|}.$$

Alternatively, we can first bound $\sum_{t=1}^T \|\widetilde{\ell}_t\|^2 \|u_t\| \leq M\sum_{t=1}^T \|\widetilde{\ell}_t\|^2$ and then apply a concentration inequality to bound the summation with high-probability. In particular, by Zhang & Cutkosky (2022, Lemma 24), we have that with probability at least $1 - \delta$

$$4\sqrt{(\Phi_T + P_T^\Phi)\sum_{t=1}^T \|\widetilde{\ell}_t\|^2 \|u_t\|} \leq 4\sqrt{M(\Phi_T + P_T^\Phi)\sum_{t=1}^T \|\widetilde{\ell}_t\|^2}$$

$$\leq 4\sqrt{M(\Phi_T + P_T^\Phi)\left(\frac{3}{2}(2dG^2)T + \frac{5}{3}(4d^2G^2)\log(1/\delta)\right)}$$

$$\leq 4G\sqrt{M(\Phi_t + P_T^\Phi)\left(3dT + \frac{20}{3}d^2\log(1/\delta)\right)}$$

$$\leq 4G\sqrt{dM(\Phi_T + P_T^\Phi)(3T + 8d\log(1/\delta))}$$

$$\leq 8G\sqrt{2dM(\Phi_T + P_T^\Phi)[T + d\log(1/\delta)]}.$$

Hence, combining these two bounds, we have that with probability $1 - \delta$,

$$4\sqrt{(\Phi_T + P_T^\Phi)\sum_{t=1}^T \|\widetilde{\ell}_t\|^2 \|u_t\|} \leq \min\left\{8d\sqrt{(\Phi_T + P_T^\Phi)\sum_{t=1}^T \|\ell_t\|^2 \|u_t\|},\right.$$

$$\left. 8G\sqrt{2dM(\Phi_T + P_T^\Phi)[T + d\log(1/\delta)]}\right\}$$

$$\leq \min\left\{8d\sqrt{(\Phi_T + P_T^\Phi)\sum_{t=1}^T \|\ell_t\|^2 \|u_t\|},\right.$$

$$\left. 8G\sqrt{2M(\Phi_T + P_T^\Phi)}\left[\sqrt{dT} + d\sqrt{\log(1/\delta)}\right]\right\} \quad (12)$$

Next, we simplify the terms $\Sigma(w_{1:T})$. First, notice that by Cauchy-Schwarz inequality, it holds that $\sum_{\eta \in \mathcal{S}} \|w_t^\eta\| \leq \sqrt{|\mathcal{S}| \sum_{\eta \in \mathcal{S}} \|w_t^\eta\|^2}$, and so

$$\left\| \sum_{\eta \in \mathcal{S}} w_t^\eta \right\|^2 \leq \left( \sum_{\eta \in \mathcal{S}} \|w_t^\eta\| \right)^2 \leq |\mathcal{S}| \sum_{\eta \in \mathcal{S}} \|w_t^\eta\|^2,$$

so using this and the fact that $\sqrt{a+b} \leq \sqrt{a} + \sqrt{b}$, we can break $\Sigma_T(w_{1:T})$ apart as

$$
\begin{aligned}
\Sigma_T(w_{1:T}) &\leq \sum_{\eta \in \mathcal{S}} 3G \sqrt{d |\mathcal{S}| \sum_{t=1}^T \|w_t^\eta\|^2 \log\left(\frac{4}{\delta} \left[ \log_+\left( \frac{\sqrt{|\mathcal{S}| \max_{\eta \in \mathcal{S}} \sum_{t=1}^T \|w_t^\eta\|^2}}{\omega} \right) + 2 \right]^2 \right)} \\
&\quad + \sum_{\eta \in \mathcal{S}} 16 dG \max\left( \omega, \max_{t \leq T} \|w_t^\eta\| \right) \log\left( \frac{28}{\delta} \left[ \log_+\left( \frac{\max_{\eta \in \mathcal{S}, t \leq T} \sqrt{|\mathcal{S}|} \|w_t^\eta\|}{\omega} \right) + 2 \right]^2 \right) \\
&\overset{(a)}{\leq} \sum_{\eta \in \mathcal{S}} 3G \sqrt{d |\mathcal{S}| \sum_{t=1}^T \|w_t^\eta\|^2 \log\left(\frac{4}{\delta} \left[ \log_+\left( \frac{\epsilon\sqrt{|\mathcal{S}| \sum_{t=1}^T 2^{2(t-1)}}}{\omega} \right) + 2 \right]^2 \right)} \\
&\quad + \sum_{\eta \in \mathcal{S}} 24 dG \max\left( \omega, \max_{t \leq T} \|w_t^\eta\| \right) \log\left( \frac{28}{\delta} \left[ \log_+\left( \frac{\epsilon\sqrt{|\mathcal{S}|} 2^{T-1}}{\omega} \right) + 2 \right]^2 \right) \\
&\overset{(b)}{\leq} \sum_{\eta \in \mathcal{S}} 3G \sqrt{d |\mathcal{S}| \sum_{t=1}^T \|w_t^\eta\|^2 \log\left(\frac{4}{\delta} \left[ \log_+\left( \frac{\epsilon\sqrt{|\mathcal{S}| 4^T}}{\omega} \right) + 2 \right]^2 \right)} \\
&\quad + \sum_{\eta \in \mathcal{S}} 24 dG \max\left( \omega, \max_{t \leq T} \|w_t^\eta\| \right) \log\left( \frac{28}{\delta} \left[ \log_+\left( \frac{\epsilon\sqrt{|\mathcal{S}|} 2^{T-1}}{\omega} \right) + 2 \right]^2 \right) \\
&\overset{(c)}{\leq} \sum_{\eta \in \mathcal{S}} 3G \sqrt{d |\mathcal{S}| \sum_{t=1}^T \|w_t^\eta\|^2 \log\left(\frac{4}{\delta} \left[ T + \log_+\left( \frac{4\epsilon\sqrt{|\mathcal{S}|}}{\omega} \right) \right]^2 \right)} \\
&\quad + \sum_{\eta \in \mathcal{S}} 24 dG \max\left( \omega, \max_{t \leq T} \|w_t^\eta\| \right) \log\left( \frac{28}{\delta} \left[ T + \log_+\left( \frac{2\sqrt{|\mathcal{S}|}\epsilon}{\omega} \right) \right]^2 \right)
\end{aligned}
\tag{13}
$$

where $(a)$ uses the fact that the base-algorithms $\mathcal{A}_\eta$ satisfy a comparator-adaptive guarantee, and hence have $\|w_t^\eta\| \leq \epsilon 2^{t-1}$ for any $w_t^\eta$ via Lemma I.5, and $(b)$ uses $\sum_{t=1}^T a^{t-1} = (a^T - 1)/(a-1) \leq a^T$ for $a \geq 2$ and $(c)$ uses $\log_+\left( c2^{T-2} \right) + 2 \leq (T-2)\log(e) + \log_+(c) + 2 = T + \log_+(c)$ for $c > 0$. Therefore, we set each $\varphi_t^\eta(w)$ using two components, one to cancel each of these summations. First, observe that with $r_t^\eta(w; c_1, \alpha_1, 2)$ in Lemma E.4 (defined w.r.t. $w_t^\eta$), we have

$$\sum_{t=1}^T r_t^\eta(w_t^\eta; c_1, \alpha_1, 2) \geq c_1 \sqrt{\alpha_1^2 + \sum_{t=1}^T \|w_t^\eta\|^2} - c_1 \alpha_1,$$

and so if we set $c_1 = 6G\sqrt{d |\mathcal{S}| \log\left( \frac{4}{\delta} \left[ T + \log_+\left( \frac{4\epsilon\sqrt{|\mathcal{S}|}}{\omega} \right) \right]^2 \right)}$ we will cancel the first part of $2\Sigma_T^\eta(w_{1:T}^\eta)$ for each $\eta$.

To cancel the second term of $2\Sigma_T^\eta(w_{1:T}^\eta)$ in Equation (13), suppose we add a term $r_t^\eta(w; c_2, \alpha_2, p_2)$ with $p_2 = \log(T+1)$.

Then again using Lemma E.4, we have

$$\sum_{t=1}^{T} r_t^\eta(w_t^\eta; c_2, \alpha_2, p_2) \geq c_2 \left( \alpha_2^{p_2} + \sum_{t=1}^{T} \|w_t^\eta\|^{p_2} \right)^{\frac{1}{p}} - c_2\alpha_2 \geq c_2 \max\left\{\alpha_2, \max_t \|w_t^\eta\|\right\} - \alpha_2 c_2$$

Therefore, setting $c_2 = 48dG \log\left(\frac{28}{\delta}\left[T + \log_+\left(\frac{2\epsilon\sqrt{|\mathcal{S}|}}{\omega}\right)\right]^2\right)$ and $\alpha_2 = \omega$, we cancel the remaining part of $2\Sigma_T^\eta(w_{1:T})$ for each $\eta$. Finally, plugging this all back into the regret bound and expanding $\Sigma_T(u_{1:T})$ as in Equation (13), and choosing $\alpha_1 = \epsilon$, we have that with probability at least $1 - 4\delta$

$$R_T(u_{1:T}) \leq \min\left\{8d\sqrt{(\Phi_T + P_T^\Phi)\sum_{t=1}^{T}\|\ell_t\|^2\|u_t\|}, \ 8G\sqrt{2M(\Phi_T + P_T^\Phi)}\left[\sqrt{dT} + d\sqrt{\log(1/\delta)}\right]\right\}$$

$$+ 2G\sqrt{d\sum_{t=1}^{T}\|u_t\|^2 \log\left(\frac{4}{\delta}\left[\log_+\left(\frac{\sqrt{\sum_{t=1}^{T}\|u_t\|^2}}{\omega}\right) + 2\right]^2\right)}$$

$$+ 4c_1\sqrt{\left(\epsilon^2 + \sum_{t=1}^{T}\|u_t\|^2\right)\log\left(e + \frac{e\sum_{t=1}^{T}\|u_t\|^2}{\epsilon^2}\right)}$$

$$+ 3c_2\log^2(T+1)\max\{\omega, M\}\left[\log_+\left(\frac{3M}{\omega}\right) + 3\right]$$

$$+ 24dG\max\left\{\omega, \max_{t\leq T}\|u_t\|\right\}\log\left(\frac{28}{\delta}\left[\log_+\left(\frac{\max_{t\leq T}\|u_t\|}{\omega}\right) + 2\right]^2\right)$$

$$+ 32(dG + H)(\epsilon|\mathcal{S}| + M + \Phi_T + P_T^\Phi)$$

$$+ c_1\epsilon + c_2\omega + 3dGP_T + 2G\omega\sqrt{2\log\left(\frac{16}{\delta}\right)}$$

$$\leq \widetilde{O}\left(\min\left\{d\sqrt{(\Phi_T + P_T^\Phi)\sum_{t=1}^{T}\|\ell_t\|^2\|u_t\|}, \ G\sqrt{M(\Phi_T + P_T^\Phi)}\left[\sqrt{dT} + d\sqrt{\log(1/\delta)}\right]\right\}\right.$$

$$\left. + G\sqrt{d\sum_{t=1}^{T}\|u_t\|^2\log(T/\delta)} + dG(\epsilon + M + \Phi_T + P_T^\Phi)\log(T/\delta)\right)$$

where we've used that $H = c_1p_1 + c_2p_2 = \widetilde{O}(dG)$ after hiding poly-logarithmic factors. $\qquad\square$

### D.3.1. EXTENSION TO ARBITRARY COMPARATOR SEQUENCES

Our high-probability guarantees in the main text are framed in terms of $(\mathcal{F}_t)_{t=0}^{T}$-adapted comparator sequences $(u_t)_{t\in[T]}$. However, the same results can be obtained up to constant factors for *arbitrary* comparator sequences as well by using the same ghost-iterate trick used in Proposition 2.3, though the statement becomes a bit more complicated to state because the "ghost-iterates" enter into the proposition statement. The following lemma provides a generalization of Proposition 4.1 to arbitrary comparator sequences.

**Proposition D.2.** *Let $\mathcal{A}$ be an OLO learner and let $w_t \in \mathbb{R}^d$ denote its output on round $t$. Let $\widehat{\mathcal{A}}$ be a virtual instance of $\mathcal{A}$, appearing only in the analysis, and let $\widehat{w}_t \in \mathbb{R}^d$ denote its output on round $t$. Let $\delta \in (0, 1/3]$, $\omega > 0$, and $\varepsilon^2 = \omega^2/T$, and let $u_{1:T} = (u_t)_{t=1}^{T}$ be an arbitrary sequence in $\mathbb{R}^d$. Then `PABLO` with $H_t = \frac{1}{d[\|w_t\|^2 \vee \varepsilon^2]}I_d$ guarantees that with probability at least $1 - 3\delta$,*

$$R_T(u_{1:T}) \leq \underbrace{\sum_{t=1}^{T}\left\langle\widetilde{\ell}_t, w_t - u_t\right\rangle + 2\Sigma_T(w_{1:T})}_{\widetilde{R}_T^{\mathcal{A}}(u_{1:T})} + \underbrace{\sum_{t=1}^{T}\left\langle\ell_t - \widetilde{\ell}_t, \widehat{w}_t - u_t\right\rangle + \Sigma_T(\widehat{w}_{1:T})}_{\widehat{R}_T^{\widehat{\mathcal{A}}}(u_{1:T})} + 2G\omega\sqrt{2\log\left(\frac{16}{\delta}\right)}$$

*where $\Sigma_T : (\mathbb{R}^d)^T \to \mathbb{R}_{\geq 0}$ is defined as in Proposition 4.1.*

*Proof.* Following a similar argument to Proposition 4.1, we can decompose

$$
R_T(u_{1:T}) = \sum_{t=1}^{T} \langle \ell_t, \widetilde{w}_t - u_t \rangle
$$

$$
= \sum_{t=1}^{T} \langle \ell_t, w_t - u_t \rangle + \sum_{t=1}^{T} \left\langle \ell_t, H_t^{\frac{1}{2}} s_t \right\rangle
$$

$$
= \underbrace{\sum_{t=1}^{T} \left\langle \widetilde{\ell}_t, w_t - u_t \right\rangle}_{\widetilde{R}_T^{\mathcal{A}}(u_{1:T})} + \sum_{t=1}^{T} \left\langle \ell_t - \widetilde{\ell}_t, w_t \right\rangle + \sum_{t=1}^{T} \left\langle \widetilde{\ell}_t - \ell_t, u_t \right\rangle + \sum_{t=1}^{T} \left\langle \ell_t, H_t^{-\frac{1}{2}} s_t \right\rangle \tag{14}
$$

$$
= \widetilde{R}_T^{\mathcal{A}}(u_{1:T}) + \underbrace{\sum_{t=1}^{T} \left\langle \ell_t - \widetilde{\ell}_t, w_t \right\rangle}_{\text{\textcircled{A}}} + \underbrace{\sum_{t=1}^{T} \left\langle \ell_t - \widetilde{\ell}_t, \widehat{w}_t \right\rangle}_{\text{\textcircled{B}}} + \underbrace{\sum_{t=1}^{T} \left\langle \widetilde{\ell}_t - \ell_t, \widehat{w}_t - u_t \right\rangle}_{\widehat{R}_T^{\widehat{\mathcal{A}}}(u_{1:T})} + \underbrace{\sum_{t=1}^{T} \left\langle \ell_t, H_t^{-\frac{1}{2}} s_t \right\rangle}_{\text{\textcircled{C}}}. \tag{15}
$$

We proceed by bounding each of the noise terms $\text{\textcircled{A}}$, $\text{\textcircled{B}}$, and $\text{\textcircled{C}}$ with high probability. The terms $\text{\textcircled{A}}$ and $\text{\textcircled{C}}$ can be bound exactly the same as in Proposition 4.1. The new term, $\text{\textcircled{B}}$, can be bound using the exact same argument as $\text{\textcircled{A}}$ but replacing $w_t$ with $\widehat{w}_t$. Overall, we get that with probability at least $1 - 3\delta$

$$
\text{\textcircled{A}} \leq \Sigma(w_{1:T}), \quad \text{\textcircled{B}} \leq \Sigma(\widehat{w}_{1:T}), \quad \text{\textcircled{C}} \leq 2G\omega\sqrt{2\log(\tfrac{16}{\delta})} + \Sigma_T(w_{1:T}),
$$

for an overall bound of

$$
R_T(u_{1:T}) \leq R_T^{\mathcal{A}}(u_{1:T}) + 2\Sigma_T(w_{1:T}) + \widehat{R}_T^{\widehat{\mathcal{A}}}(u_{1:T}) + \Sigma_T(\widehat{w}_{1:T}) + 2G\omega\sqrt{2\log(\tfrac{16}{\delta})}
$$

with probability at least $1 - 3\delta$. $\qquad\square$

Now with the above reduction in hand, proof of the main result for arbitrary comparator sequences follows using a similar strategy to Theorem 4.4: we choose $\mathcal{A}$ which provides an additional composite penalty $\sum_{t=1}^{T} \varphi_t(u_t) - \varphi_t(w_t)$, such that $\sum_{t=1}^{T} \varphi_t(w_t) \geq \Sigma_T(w_{1:T})$ while also ensuring that $\sum_{t=1}^{T} \varphi_t(u_t)$ is controlled. Importantly, the virtual version of $\mathcal{A}$ will likewise also generate such a term capable of cancelling $\Sigma_T(\widehat{w}_{1:T})$. The main difference is that we should increase the Lipschitz constant to $L = 3dG$, so that we have both $\|\widetilde{\ell}_t\| \leq 2dG \leq L$ and $\|\ell_t - \widetilde{\ell}_t\| \leq \|\ell_t\| + \|\widetilde{\ell}_t\| \leq G + 2dG \leq 3dG \leq L$ (via Corollary 2.2), in which case we can apply the same arguments as in Theorem 4.4 to bound each of the terms. The same arguments then lead to a bound that matches Theorem 4.4 up to constant factors. An analogous argument can be made for our high-probability static regret bound to generalize the result to an arbitrary fixed comparator norm $\|u\|$ but using the algorithm of Zhang & Cutkosky (2022) as the base algorithm $\mathcal{A}$.

## E. Results for Online Convex Optimization

In this section we present auxiliary results for OCO which drive our dynamic regret guarantees for uBLO in Section 3.2 and Section 4. We present these results separately here since these results are of independent interest in OCO.

In particular, in Appendix E.1 we show how a combination of composite regularization and implicit optimistic updates can introduce additional terms into the regret $\sum_{t=1}^{T} \varphi_t(u_t) - \varphi_t(w_t)$ which can be used as a means to control the stability of the iterates $\|w_t\|$. This is crucial in our high-probability results in Section 4, where controlling the bias of our loss estimates leads to penalties scaling with $\widetilde{\mathcal{O}}(\sqrt{\sum_t \|w_t\|^2})$. Our approach is inspired by the strategy used by Zhang & Cutkosky (2022), though our results require additional care to extend the approach to dynamic regret while also preserving the adaptivity to the individual comparator norms, which the optimistic reductions of Zhang & Cutkosky (2022); Cutkosky (2019) cannot account for. In Appendix E.2 we provide a refinement of the dynamic base algorithm of Jacobsen & Cutkosky (2022) which simplifies the analysis and exposes a more fine-grained measure of comparator variability than the $P_T \log(MT/\epsilon + 1)$ presented in the original work, while also avoiding the worst-case factors of $M = \max_t \|u_t\|$ from appearing in the main terms of the bound.

Throughout this section, we focus on a general OCO setting in which the losses $(f_t)_{t\in[T]}$ are arbitrary $L$-Lipschitz convex functions. We denote the regret as

$$R_T(u_{1:T}) = \sum_{t=1}^{T} f_t(w_t) - f_t(u_t)$$

and, when relevant, we denote the regret of a given algorithm $\mathcal{A}$ by $R_T^{\mathcal{A}}(u_{1:T})$.

### E.1. Online Learning with Optimistic Composite-penalty Cancellation

---

**Algorithm 4** Online Learning with Optimistic Composite-Penalty Cancellation

---

**Require:** $f_t : \mathbb{R}^d \to \mathbb{R}$ is convex and $L$-Lipschitz and $\varphi_t : \mathbb{R}^d \to \mathbb{R}$ is convex and $H$-Lipschitz for all $t$
**Input:** step-size $0 \le \eta \le \frac{1}{L+H}$, algorithms $\mathcal{A}_x$ and $\mathcal{A}_y$ defined on $\mathbb{R}^d$ and $\mathbb{R}_{\ge 0}$ respectively
**for** $t = 1 : T$ **do**
    Get $x_t \in \mathbb{R}^d$ from $\mathcal{A}_x$ and $y_t \in R_{\ge 0}$ from $\mathcal{A}_y$
    Solve for $w_t = x_t - y_t \eta \nabla \varphi_t(w_t)$
    Play $w_t$ and receive $g_t \in \partial f_t(w_t)$
    Send $g_t + \nabla \varphi_t(w_t)$ to $\mathcal{A}_x$ as the $t^{\text{th}}$ subgradient
    Send $-\eta \langle g_t + \nabla \varphi_t(w_t), \nabla \varphi_t(w_t) \rangle$ to $\mathcal{A}_y$ as the $t^{\text{th}}$ subgradient
**end for**

---

In this section we develop an algorithm for dynamic regret which introduces additional penalties to the regret of $\sum_{t=1}^{T} \varphi_t(u_t) - \varphi_t(w_t)$, which will let us cancel out the $\|w_t\|$-dependent terms from Proposition 4.1. The key difficulty is that this changes the OLO learner's feedback to be $g_t + \nabla \varphi_t(w_t)$ which is problematic in our application of interest because the gradients of the composite penalty in Equation (19) would again depend on $\|w_t\|$. Fortunately, we can remove this factor from the feedback by using an optimistic update by setting the optimistic hint to be $h_t = \nabla \varphi_t(w_t)$, so that $g_t + \nabla \varphi_t(w_t) - h_t = g_t$. Note that the optimistic reduction in Algorithm 4 incorporates the hints $h_t = \eta \nabla \varphi_t(w_t)$ by playing $w_t = x_t - y_t \eta h_t$, so when $h_t = \nabla \varphi_t(w_t)$ this generally requires solving a fixed-point equation. We assume that $\varphi_t$ is chosen in such a way that the solution to this fixed-point equation exists, and we note that this is indeed the case when $\varphi_t$ is set according to the Huber-like penalty in Appendix E.1.1, which is used in our high-probability results for uBLO (see Zhang & Cutkosky (2022, Lemma 6)).

The following theorem shows that we can extend the per-comparator dynamic regret guarantee of Jacobsen & Cutkosky (2022) to include additional terms $\sum_{t=1}^{T} \varphi_t(u_t) - \varphi_t(w_t)$ in the bound without otherwise changing the original regret guarantee significantly. We will instantiate the result more concretely in Theorem E.2.

**Theorem E.1.** *Let $\mathcal{A}_x$ and $\mathcal{A}_y$ be online learning algorithms defined on convex domains $\mathcal{W}_x = \mathbb{R}^d$ and $\mathcal{W}_y = \mathbb{R}_{\ge 0}$ respectively and let $w_t^{\mathcal{A}_x} \in \mathcal{W}_x$ and $w_t^{\mathcal{A}_y} \in \mathcal{W}_y$ denote their respective outputs on round $t$. Suppose that for each $z \in \{x, y\}$, $\mathcal{A}_z$ guarantees that for any sequence of $L$-Lipschitz convex loss functions $\widetilde{f}_1, \ldots, \widetilde{f}_T$ on $\mathcal{W}_z$ and any sequence $u_{1:T}$ in $\mathcal{W}_z$ that*

$$R_T^{\mathcal{A}_z}(u_{1:T}) \le A_T^{\mathcal{A}_z}(u_{1:T}) + \frac{P_T^{\mathcal{A}_z}(u_{1:T})}{2\eta} + \frac{\eta}{2} \sum_{t=1}^{T} \|\widetilde{g}_t\|^2 \|u_t\|,$$

*where $\widetilde{g}_t \in \partial \widetilde{f}_t(w_t^{\mathcal{A}_z})$, for some non-negative functions $A_T^{\mathcal{A}_z} : (\mathcal{W}_z)^T \to \mathbb{R}_{\ge 0}$ and $P_T^{\mathcal{A}_z} : (\mathcal{W}_z)^T \to \mathbb{R}_{\ge 0}$. Then for any sequence of convex $L$-Lipschitz losses $f_1, \ldots, f_T$ on $\mathbb{R}^d$ and any comparator sequence $u_{1:T} = (u_1, \ldots, u_T)$ in $\mathbb{R}^d$, Algorithm 4 guarantees*

$$R_T(u_{1:T}) \le A_T^{\mathcal{A}_x}(u_{1:T}) + A_T^{\mathcal{A}_y}(\|u\|_{1:T}) + \frac{P_T^{\mathcal{A}_x}(u_{1:T}) + P_T^{\mathcal{A}_y}(\|u\|_{1:T})}{2\eta} + \frac{\eta}{2} \sum_{t=1}^{T} \|g_t\|^2 \|u_t\|$$

$$+ \sum_{t=1}^{T} \varphi_t(u_t) - \varphi_t(w_t)$$

*where $g_t \in \partial f_t(w_t)$.*

*Proof.* We have by convexity of $f_t$ that

$$R_T(u_{1:T}) \leq \sum_{t=1}^{T} \langle g_t, w_t - u_t \rangle \tag{16}$$

$$= \sum_{t=1}^{T} \langle g_t, w_t - u_t \rangle \pm [\varphi_t(w_t) - \varphi_t(u_t)]$$

$$\leq \underbrace{\sum_{t=1}^{T} \langle g_t + \nabla\varphi_t(w_t), w_t - u_t \rangle}_{=:\widetilde{R}_T(u_{1:T})} + \sum_{t=1}^{T} \varphi_t(u_t) - \varphi_t(w_t) \tag{17}$$

where $g_t \in \partial f_t(w_t)$ and the last line uses convexity of $\varphi_t$. Focusing on the first term, denote $\widetilde{g}_t = g_t + \nabla\varphi_t(w_t)$ and observe that for $w_t = x_t - \eta y_t \nabla\varphi_t(w_t)$ and any arbitrary sequence $\mathring{y}_{1:T}$ in $\mathbb{R}_{\geq 0}$ we have

$$\widetilde{R}_T(u_{1:T}) = \sum_{t=1}^{T} \langle \widetilde{g}_t, w_t - u_t \rangle = \sum_{t=1}^{T} \langle \widetilde{g}_t, x_t - u_t \rangle + \sum_{t=1}^{T} \langle -\eta g_t, \nabla\varphi_t(w_t) \rangle y_t$$

$$= R_T^{\mathcal{A}_x}(u_{1:T}) + \sum_{t=1}^{T} (\langle -\eta\widetilde{g}_t, \nabla\varphi_t(w_t) \rangle y_t - \langle -\eta\widetilde{g}_t, \nabla\varphi_t(w_t) \rangle \mathring{y}_t) - \sum_{t=1}^{T} \langle \eta\widetilde{g}_t, \nabla\varphi_t \rangle \mathring{y}_t$$

$$= R_T^{\mathcal{A}_x}(u_{1:T}) + R_T^{\mathcal{A}_y}(\mathring{y}_{1:T}) - \eta \sum_{t=1}^{T} \langle \widetilde{g}_t, \nabla\varphi_t(w_t) \rangle \mathring{y}_t$$

$$\overset{(a)}{=} R_T^{\mathcal{A}_x}(u_{1:T}) + R_T^{\mathcal{A}_y}(\mathring{y}_{1:T}) + \frac{\eta}{2} \sum_{t=1}^{T} \left[ \|\widetilde{g}_t - \nabla\varphi_t(w_t)\|^2 - \|\widetilde{g}_t\|^2 - \|\nabla\varphi_t(w_t)\|^2 \right] \mathring{y}_t$$

$$\overset{(b)}{=} R_T^{\mathcal{A}_x}(u_{1:T}) + R_T^{\mathcal{A}_y}(\mathring{y}_{1:T}) + \frac{\eta}{2} \sum_{t=1}^{T} \left[ \|g_t\|^2 - \|\widetilde{g}_t\|^2 - \|\nabla\varphi_t(w_t)\|^2 \right] |\mathring{y}_t|$$

where $(a)$ uses the elementary identity $-2\langle x, y \rangle = \|x - y\|^2 - \|x\|^2 - \|y\|^2$ and $(b)$ recalls that $\widetilde{g}_t = g_t + \nabla\varphi_t(w_t)$ so that $\widetilde{g}_t - \nabla\varphi_t(w_t) = g_t$, and writes $\mathring{y}_t = |\mathring{y}_t|$ for $\mathring{y} \geq 0$. Now from the regret guarantee of $\mathcal{A}_x$ applied to the linear losses $w \mapsto \langle \widetilde{g}_t, w \rangle$, we have

$$R_T^{\mathcal{A}_x}(u_{1:T}) \leq A_T^{\mathcal{A}_x}(u_{1:T}) + \frac{P_T^{\mathcal{A}_x}(u_{1:T})}{2\eta} + \frac{\eta}{2} \sum_{t=1}^{T} \|\widetilde{g}_t\|^2 \|u_t\|,$$

for some $A_T^{\mathcal{A}_x}(u_{1:T}), P_T^{\mathcal{A}_x}(u_{1:T}) \geq 0$ and $0 < \eta \leq \frac{1}{L+H} \leq \frac{1}{\|\widetilde{g}_t\|}$. Then choosing $\mathring{y}_t = \|u_t\|$ for all $t$, we have

$$\widetilde{R}_T(u_{1:T}) \leq A_T^{\mathcal{A}_x}(u_{1:T}) + \frac{P_T^{\mathcal{A}_x}(u_{1:T})}{2\eta} + \frac{\eta}{2} \sum_{t=1}^{T} \|\widetilde{g}_t\|^2 \|u_t\| + R_T^{\mathcal{A}_y}(\mathring{y}_{1:T})$$

$$+ \frac{\eta}{2} \sum_{t=1}^{T} \left[ \|g_t\|^2 - \|\widetilde{g}_t\|^2 - \|\nabla\varphi_t(w_t)\|^2 \right] \|u_t\|$$

$$\leq A_T^{\mathcal{A}_x}(u_{1:T}) + \frac{P_T^{\mathcal{A}_x}(u_{1:T})}{2\eta} + \frac{\eta}{2} \sum_{t=1}^{T} \|g_t\|^2 \|u_t\| + R_T^{\mathcal{A}_y}(\mathring{y}_{1:T}) - \frac{\eta}{2} \sum_{t=1}^{T} \|\nabla\varphi_t(w_t)\|^2 \|u_t\|.$$

Likewise, from the regret guarantee of $\mathcal{A}_y$ applied to the linear losses $y \mapsto \langle -\eta\widetilde{g}_t, \nabla\varphi_t(w_t) \rangle y$, we get

$$R_T^{\mathcal{A}_y}(\mathring{y}_{1:T}) \leq A_T^{\mathcal{A}_y}(u_{1:T}) + \frac{P_T^{\mathcal{A}_y}(\mathring{y}_{1:T})}{2\eta} + \frac{\eta}{2} \sum_{t=1}^{T} \langle \eta\widetilde{g}_t, \nabla\varphi_t(w_t) \rangle^2 |\mathring{y}_t|$$

$$\leq A_T^{\mathcal{A}_y}(u_{1:T}) + \frac{P_T^{\mathcal{A}_y}(\mathring{y}_{1:T})}{2\eta} + \frac{\eta}{2} \sum_{t=1}^{T} \|\nabla\varphi_t(w_t)\|^2 \|u_t\|,$$

where we've used $\eta\|\widetilde{g}_t\| \leq 1$, so plugging this back in above we have

$$\widetilde{R}_T(u_{1:T}) \leq A_T^{\mathcal{A}_x}(u_{1:T}) + A_T^{\mathcal{A}_y}(\|u\|_{1:T}) + \frac{P_T^{\mathcal{A}_x}(u_{1:T}) + P_T^{\mathcal{A}_y}(\|u\|_{1:T})}{2\eta} + \frac{\eta}{2}\sum_{t=1}^{T}\|g_t\|^2\|u_t\| \qquad \Box$$

For concreteness, we instantiate this result with the algorithm characterized in Theorem E.6 for both $\mathcal{A}_x$ on $\mathbb{R}^d$ and $\mathcal{A}_y$ on $\mathbb{R}_{\geq 0}$ to immediately get the following result.

**Theorem E.2.** *Under the same assumptions as Theorem E.1, let both $\mathcal{A}_x$ and $\mathcal{A}_y$ be instances of Algorithm 5 with $\eta \leq 1/L$ and $k \geq 4$, applied on $\mathbb{R}^d$ and $\mathbb{R}_{\geq 0}$ respectively. Then for any sequence of $L$-Lipschitz convex functions $f_1, \ldots, f_T$ and any sequence $u_{1:T} = (u_1, \ldots, u_T)$ in $\mathbb{R}^d$, Algorithm 4 guarantees*

$$R_T(u_{1:T}) \leq 2(L+H)(\epsilon + \max_t \|u_t\|) + \frac{16\left[\Phi\left(\|u_T\|, \frac{T}{\epsilon}\right) + P_T^{\Phi}\left(\frac{4T^3}{\epsilon}\right)\right]}{2\eta} + \frac{\eta}{2}\sum_{t=1}^{T}\|g_t\|^2\|u_t\|$$

$$+ \sum_{t=1}^{T}\varphi_t(u_t) - \varphi_t(w_t),$$

*where $g_t \in \partial f_t(w_t)$ and we define $\Phi(x, \lambda) = x\log(\lambda x + 1)$ and $P_T^{\Phi}(\lambda) = \sum_{t=2}^{T}\Phi(\|u_t - u_{t-1}\|, \lambda)$.*

*Proof.* The algorithm characterized by Theorem E.6 satisfies the condition of Theorem E.1 with $A_T^{\mathcal{A}_x}(u_{1:T}) \leq L(\epsilon + \max_t \|u_t\|)$ and $P_T^{\mathcal{A}_x}(u_{1:T}) = 8\Phi\left(\|u_T\|, \frac{T}{\epsilon}\right) + 8P_T^{\Phi}\left(\frac{4T^3}{\epsilon}\right)$, where

$$\Phi(x, \lambda) = x\log(\lambda x + 1), \text{ and } P_T^{\Phi}(\lambda) = \sum_{t=2}^{T}\Phi(\|u_t - u_{t-1}\|, \lambda)$$

Likewise, $A_T^{\mathcal{A}_y}(\|u\|_{1:T}) = A_T^{\mathcal{A}_x}(u_{1:T})$ and using reverse triangle inequality we have for any $t$, $|\|u_t\| - \|u_{t-1}\|| \leq \|u_t - u_{t-1}\|$, so $P_T^{\mathcal{A}_y}(\|u\|_{1:T}) \leq P_T^{\mathcal{A}_x}(u_{1:T})$. Thus, applying Theorem E.2, we have

$$R_T(u_{1:T}) \leq A_T^{\mathcal{A}_x}(u_{1:T}) + A_T^{\mathcal{A}_y}(\|u\|_{1:T}) + \frac{P_T^{\mathcal{A}_x}(u_{1:T}) + P_T^{\mathcal{A}_y}(\|u\|_{1:T})}{2\eta} + \frac{\eta}{2}\sum_{t=1}^{T}\|g_t\|^2\|u_t\|$$

$$+ \sum_{t=1}^{T}\varphi_t(u_t) - \varphi_t(w_t)$$

$$\leq 2(L+H)\epsilon + \frac{16\left[\Phi\left(\|u_T\|, \frac{T}{\epsilon}\right) + P_T^{\Phi}\left(\frac{4T^3}{\epsilon}\right)\right]}{2\eta} + \frac{\eta}{2}\sum_{t=1}^{T}\|g_t\|^2\|u_t\| + \sum_{t=1}^{T}\varphi_t(u_t) - \varphi_t(w_t). \qquad \Box$$

Finally, we have the following simple hyperparameter tuning argument. The result simply shows that if we add the iterates of many instances of the above algorithm together with different step-sizes, we can get a regret guarantee which balances the trade-off in $\eta$. For ease of exposition, in what follows we denote this trade-off as

$$\mathcal{T}(\eta) \stackrel{\text{def}}{=} \frac{16\left[\Phi\left(\|u_T\|, \frac{T}{\epsilon}\right) + P_T^{\Phi}\left(\frac{4T^3}{\epsilon}\right)\right]}{2\eta} + \frac{\eta}{2}\sum_{t=1}^{T}\|g_t\|^2\|u_t\|. \qquad (18)$$

The terms are balanced, up to lower-order terms, by selecting $\eta^* = \arg\min_{\eta \in \mathcal{S}}\mathcal{T}(\eta)$, as shown by the following theorem.

**Theorem E.3.** *Let $\mathcal{S} = \left\{\eta_i : \eta_i = \frac{2^i}{(L+H)T} \wedge \frac{1}{L+H}, i = 0, 1, \dots\right\}$ and for each $\eta \in \mathcal{S}$, let $\mathcal{A}_\eta$ denote an instance of the algorithm characterized in [Theorem E.2](#), and let $w_t^\eta$ and $\varphi_t^\eta : \mathbb{R}^d \to \mathbb{R}$ denote its output and $H$-Lipschitz composite penalty on round $t$ respectively. Suppose that on each round we play $w_t = \sum_{\eta \in \mathcal{S}} w_t^\eta$. Then for any sequence of $L$-Lipschitz convex functions $f_1, \dots, f_T$ and any sequence $u_{1:T} = (u_1, \dots, u_T)$ in $\mathbb{R}^d$,*

$$R_T(u_{1:T}) \le 16(L+H)\left(\epsilon|\mathcal{S}| + M + \Phi(\|u_T\|, \tfrac{T}{\epsilon}) + P_T^\Phi\left(\tfrac{4T^3}{\epsilon}\right)\right) + 4\sqrt{\left[\Phi\left(\|u_T\|, \tfrac{T}{\epsilon}\right) + P_T^\Phi\left(\tfrac{4T^3}{\epsilon}\right)\right]\sum_{t=1}^T \|g_t\|^2 \|u_t\|}$$

$$+ \sum_{\eta \in \mathcal{S}}\sum_{t=1}^T \varphi_t^\eta\left(u_t \mathbb{I}(\eta = \eta^*)\right) - \varphi_t^\eta(w_t^\eta),$$

*where $\eta^* = \arg\min_{\eta \in \mathcal{S}} \mathcal{T}(\eta)$ with $\mathcal{T}$ defined in [Equation (18)](#), $g_t \in \partial f_t(w_t)$, $M = \max_t \|u_t\|$, $\Phi(x, \lambda) = x \log(\lambda x + 1)$, and $P_T^\Phi(\lambda) = \sum_{t=2}^T \Phi(\|u_t - u_{t-1}\|, \lambda)$.*

*Proof.* Observe that for any $\eta \in \mathcal{S}$, we have via convexity of $f_t$ that

$$R_T(u_{1:T}) \le \sum_{t=1}^T \langle g_t, w_t - u_t \rangle = \sum_{t=1}^T \langle g_t, w_t^\eta - u_t \rangle + \sum_{\widetilde{\eta} \ne \eta}\sum_{t=1}^T \left\langle g_t, w_t^{\widetilde{\eta}} \right\rangle$$

$$= R_T^{\mathcal{A}_\eta}(u_{1:T}) + \sum_{\widetilde{\eta} \ne \eta} R_T^{\mathcal{A}_{\widetilde{\eta}}}(\mathbf{0})$$

$$\overset{(a)}{\le} R_T^{\mathcal{A}_\eta}(u_{1:T}) + 2(L+H)\epsilon(|\mathcal{S}| - 1) + \sum_{\widetilde{\eta} \ne \eta}\sum_{t=1}^T \varphi_t^{\widetilde{\eta}}(0) - \varphi_t^{\widetilde{\eta}}(w_t^{\widetilde{\eta}})$$

$$\overset{(b)}{\le} 2(L+H)(|\mathcal{S}|\epsilon + M) + \underbrace{\frac{16\left[\Phi\left(\|u_T\|, \tfrac{T}{\epsilon}\right) + P_T^\Phi\left(\tfrac{4T^3}{\epsilon}\right)\right]}{2\eta} + \frac{\eta}{2}\sum_{t=1}^T \|g_t\|^2 \|u_t\|}_{=:\mathcal{T}(\eta)}$$

$$+ \sum_{t=1}^T \varphi_t^\eta(u_t) - \sum_{\widetilde{\eta} \in \mathcal{S}}\sum_{t=1}^T \varphi_t^{\widetilde{\eta}}(w_t^{\widetilde{\eta}})$$

where $g_t \in \partial f_t(w_t)$, $M = \max_t \|u_t\|$, $(a)$ applies [Theorem E.2](#) to bound $R_T^{\mathcal{A}_{\widetilde{\eta}}}(\mathbf{0})$ for each of the $\widetilde{\eta} \ne \eta$, and $(b)$ applies [Theorem E.2](#) for $\mathcal{A}_\eta$ against the comparator sequence $u_{1:T}$. Moreover, notice that the previous display holds for any arbitrary $\eta \in \mathcal{S}$. Therefore, applying [Lemma I.3](#) we have

$$R_T(u_{1:T}) \le 16(L+H)\left(\epsilon|\mathcal{S}| + M + \Phi(\|u_T\|, \tfrac{T}{\epsilon}) + P_T^\Phi\left(\tfrac{4T^3}{\epsilon}\right)\right) + 4\sqrt{\left[\Phi\left(\|u_T\|, \tfrac{T}{\epsilon}\right) + P_T^\Phi\left(\tfrac{4T^3}{\epsilon}\right)\right]\sum_{t=1}^T \|g_t\|^2 \|u_t\|}$$

$$+ \sum_{\eta \in \mathcal{S}}\sum_{t=1}^T \varphi_t^\eta\left(u_t \mathbb{I}(\eta = \eta^*)\right) - \varphi_t^\eta(w_t^\eta),$$

where $\eta^* = \arg\min_{\eta \in \mathcal{S}} \mathcal{T}(\eta)$. $\qquad\square$

### E.1.1. USEFUL LEMMAS FOR HUBER-LIKE PENALTIES

Our main application of interest for [Algorithm 4](#) in this paper is to leverage the terms $\sum_{t=1}^T \varphi_t(u_t) - \varphi_t(w_t)$ to control certain $\|w_t\|$-dependent penalties that result from applying concentration arguments in our high-probability bounds. To this end, in this section we collect some useful auxiliary results for the Huber-like penalty originally proposed by [Zhang & Cutkosky (2022)](#), which is key to achieving our our high-probability bounds in [Section 4](#). The crucial property of these penalties is that they let us cancel out certain *algorithm-dependent* penalties $\widetilde{\mathcal{O}}\left(\left(\sum_{t=1}^T \|w_t\|^p\right)^{1/p}\right)$ and replace them with

*problem-dependent* penalties on the order of $\widetilde{\mathcal{O}}\left(\left(\sum_{t=1}^{T}\|u_t\|^p\right)^{1/p}\right)$. The following lemma defines the composite penalty and provides upper bounds for the terms that the comparator and the learner will incur when adding these additional penalties to the objective. It is a mild generalization of Zhang & Cutkosky (2022, Theorem 13) to a dynamic comparator sequence.

**Lemma E.4.** *Let $c, \alpha > 0$, $p \geq 1$, and for all $t$ let*

$$r_t(w; c, \alpha, p) = \begin{cases} c\left(p\|w\| - (p-1)\|w_t\|\right) \dfrac{\|w_t\|^{p-1}}{\left(\alpha^p + \sum_{s=1}^{t}\|w_s\|^p\right)^{1-1/p}} & \text{if } \|w\| > \|w_t\| \\ c\dfrac{\|w\|^p}{\left(\alpha^p + \sum_{s=1}^{t}\|w_t\|^p\right)^{1-1/p}} & \text{if } \|w\| \leq \|w_t\|. \end{cases} \tag{19}$$

*Then*

$$\sum_{t=1}^{T} r_t(w_t) \geq c\left(\sum_{t=1}^{T}\|w_t\|^p + \alpha^p\right)^{1/p} - c\alpha$$

$$\sum_{t=1}^{T} r_t(u_t) \leq cp\left(\alpha^p + \sum_{t=1}^{T}\|u_t\|^p\right)^{1/p}\left[\log\left(1 + \frac{\sum_{t=1}^{T}\|u_t\|^p}{\alpha^p}\right)^{\frac{p-1}{p}} + 1\right]$$

*Proof.* Observe that

$$\sum_{t=1}^{T} r_t(w_t) = \sum_{t=1}^{T} c\frac{\|w_t\|^p}{\left(\alpha^p + \sum_{s=1}^{t}\|w_s\|^p\right)^{1-1/p}} \geq \sum_{t=1}^{T} c\frac{\|w_t\|^p}{\left(\alpha^p + \sum_{t=1}^{T}\|w_t\|^p\right)^{1-1/p}} \geq c\left(\alpha^p + \sum_{t=1}^{T}\|w_t\|^p\right)^{1/p} - c\alpha,$$

where the last line uses the fact that $\alpha^p/(\alpha^p + \sum_{t=1}^{T}\|w_t\|^p)^{1-1/p} \leq \alpha$. Moreover, for any sequence $u_{1:T}$ we can upper bound

$$\sum_{t=1}^{T} r_t(u_t) = \sum_{t:\|u_t\|\leq\|w_t\|} r_t(u_t) + \sum_{t:\|u_t\|>\|w_t\|} r_t(u_t)$$

$$\leq c\left[\sum_{t:\|u_t\|\leq\|w_t\|} \frac{\|u_t\|^p}{\left(\alpha^p + \sum_{s=1}^{t}\|w_s\|^p\right)^{1-1/p}} + \sum_{t:\|u_t\|>\|w_t\|} \frac{\|u_t\|\|w_t\|^{p-1}}{\left(\alpha^p + c\sum_{s=1}^{t}\|w_s\|^p\right)^{1-1/p}}\right]$$

$$\leq c\left[\underbrace{\sum_{t:\|u_t\|\leq\|w_t\|} \frac{\|u_t\|^p}{\left(\alpha^p + \sum_{s\leq t:\|u_s\|\leq\|w_s\|}\|w_s\|^p\right)^{1-1/p}}}_{\text{\textcircled{A}}} + \underbrace{\sum_{t:\|u_t\|>\|w_t\|} \frac{\|u_t\|\|w_t\|^{p-1}}{\left(\alpha^p + c\sum_{s\leq t:\|u_s\|>\|w_s\|}\|w_s\|^p\right)^{1-1/p}}}_{\text{\textcircled{B}}}\right]$$

The first term can be bounded as

$$\text{\textcircled{A}} = \sum_{t:\|u_t\|\leq\|w_t\|} \frac{\|u_t\|^p}{\left(\alpha^p + \sum_{s\leq t:\|u_s\|\leq\|w_s\|}\|w_s\|^p\right)^{1-1/p}}$$

$$\leq \sum_{t:\|u_t\|\leq\|w_t\|} \frac{\|u_t\|^p}{\left(\alpha^p + \sum_{s\leq t:\|u_s\|\leq\|w_s\|}\|u_s\|^p\right)^{1-1/p}}$$

$$\leq p\left(\alpha^p + \sum_{t:\|u_t\|\leq\|w_t\|}\|u_t\|^p\right)^{1/p} \leq p\left(\alpha^p + \sum_{t=1}^{T}\|u_t\|^p\right)^{1/p}$$

where the last line uses Lemma I.1. The other term can be bound using Hölder inequality with $q = p$ and $q' = p/(p-1)$ so

that $1/q + 1/q' = 1$ and

$$\text{(B)} = \sum_{t:\|u_t\|>\|w_t\|} \frac{\|u_t\|\|w_t\|^{p-1}}{\left(\alpha^p + \sum_{s\le t:\|u_s\|>\|w_s\|} \|w_s\|^p\right)^{1-1/p}}$$

$$\le \left(\sum_{t:\|u_t\|>\|w_t\|} \|u_t\|^p\right)^{1/p} \left(\sum_{t:\|u_t\|>\|w_t\|} \frac{\|w_t\|^{(p-1)\frac{p}{p-1}}}{\left(\alpha^p + \sum_{s\le t:\|u_s\|>\|w_s\|} \|w_s\|^p\right)^{\frac{p-1}{p}\frac{p}{p-1}}}\right)^{(p-1)/p}$$

$$= \left(\sum_{t:\|u_t\|>\|w_t\|} \|u_t\|^p\right)^{1/p} \left(\sum_{t:\|u_t\|>\|w_t\|} \frac{\|w_t\|^p}{\left(\alpha^p + \sum_{s\le t:\|u_s\|>\|w_s\|} \|w_s\|^p\right)}\right)^{(p-1)/p}$$

$$\overset{(*)}{\le} \left(\sum_{t:\|u_t\|>\|w_t\|} \|u_t\|^p\right)^{1/p} \log\left(1 + \frac{\sum_{t:\|u_t\|>\|w_t\|} \|w_t\|^p}{\alpha^p}\right)^{(p-1)/p}$$

$$\le \left(\sum_{t:\|u_t\|>\|w_t\|} \|u_t\|^p\right)^{1/p} \log\left(1 + \frac{\sum_{t:\|u_t\|>\|w_t\|} \|u_t\|^p}{\alpha^p}\right)^{(p-1)/p}$$

$$\le \left(\sum_{t=1}^{T} \|u_t\|^p\right)^{1/p} \log\left(1 + \frac{\sum_{t=1}^{T} \|u_t\|^p}{\alpha^p}\right)^{(p-1)/p}$$

where $(*)$ uses Lemma I.2. Combining these bounds and over-approximating yields

$$\sum_{t=1}^{T} r_t(u_t) \le cp \left(\alpha^p + \sum_{t=1}^{T} \|u_t\|^p\right)^{1/p} \left[\log\left(1 + \frac{\sum_{t=1}^{T} \|u_t\|^p}{\alpha^p}\right)^{(p-1)/p} + 1\right]$$

$\square$

Now using Lemma E.4, the following lemma shows how to bound the cumulative penalty of the comparator sequence for the composite penalty used in Algorithm 3, which sets $\varphi_t$ as the sum of two Huber-like penalties.

**Lemma E.5.** *For all $t$ let $r_t(w; c, \alpha, p)$ be defined as in Lemma E.4 wrt some sequence $w_1, \ldots, w_t$, and suppose we set $\varphi_t(w) = r_t(w; c_1, \alpha_1, p_1) + r_t(w; c_2, \alpha_2, p_2)$ with $p_1 = 2$ and $p_2 = \log(T+1)$. Then for any sequence $u_{1:T} = (u_1, \ldots, u_T)$ in $\mathbb{R}^d$,*

$$\sum_{t=1}^{T} \varphi_t(u_t) = r_t(w; c_1, \alpha_1, p_1) + r_t(w; c_2, \alpha_2, p_2)$$

$$\le 4c_1 \sqrt{\alpha_1^2 + \sum_{t=1}^{T} \|u_t\|^2 \log\left(e + \frac{e\sum_{t=1}^{T} \|u_t\|^2}{\alpha_1^2}\right)} + 3c_2 \log^2(T+1) \max\{\alpha_2, M\} \left[\log_+ (3M/\alpha_2) + 3\right].$$

*where $M = \max_t \|u_t\|$.*

*Proof.* with $p_1 = 2$, we have via Lemma E.4 that

$$\sum_{t=1}^{T} r_t(u_t; c_1, \alpha_1, p_1) \le c_1 2 \sqrt{\alpha_1^2 + \sum_{t=1}^{T} \|u_t\|^2} \left[\sqrt{\log\left(1 + \frac{\sum_{t=1}^{T} \|u_t\|^2}{\alpha_1^2}\right)} + 1\right]$$

$$\le 4c_1 \sqrt{\left(\alpha_1^2 + \sum_{t=1}^{T} \|u_t\|^2\right) \log\left(e + \frac{e\sum_{t=1}^{T} \|u_t\|^2}{\alpha_1^2}\right)}.$$

Likewise, for $p_2 = \log(T+1)$ we have via Lemma E.4 that

$$\sum_{t=1}^{T} r_t(u_t; c_2, \alpha_2, p_2) \leq p_2 c_2 \left(\alpha_2^{p_2} + \sum_{t=1}^{T} \|u_t\|^{p_2}\right)^{1/p_2} \left[\log\left(1 + \frac{\sum_{t=1}^{T} \|u_t\|^{p_2}}{\alpha_2^{p_2}}\right)^{(p_2-1)/p_2} + 1\right]$$

$$\leq p_2 c_2 \max\left\{\alpha_2, \max_t \|u_t\|\right\} (T+1)^{\frac{1}{\log(T+1)}} \left[\log\left(1 + \frac{\sum_{t=1}^{T} \|u_t\|^{p_2}}{\alpha_2^{p_2}}\right)^{(p_2-1)/p_2} + 1\right]$$

$$\leq p_2 e c_2 \max\{\alpha_2, M\} \left[\log\left(1 + T\left(\frac{M}{\alpha_2}\right)^{p_2}\right) + 1\right],$$

where we've abbreviated $M = \max_t \|u_t\|$. Now suppose that $T(M/\alpha_2)^{p_2} \leq e - 1$, then the logarithm term is bounded by $\log(1 + T(M/\alpha_2)^{p_2}) \leq \log(e) = 1$. Otherwise, if $T(M/\alpha_2)^{p_2} \geq e - 1$, then using the elementary identity $\log(1+x) = \log(x) + \log(1 + 1/x)$, we have

$$\log(1 + T(M/\alpha_2)^{p_2}) = \log\left(\left(T^{1/p_2} M/\alpha_2\right)^{p_2}\right) + \log\left(1 + \frac{1}{T(M/\alpha_2)^{p_2}}\right)$$

$$\leq p_2 \log(eM/\alpha_2) + \log(e) = 1 + \log(T+1)\log(eM/\alpha_2),$$

so we may bound the previous display as

$$\sum_{t=1}^{T} r_t(u_t; c_2, \alpha_2, p_2) \leq 3\log(T+1) c_2 \max\{\alpha_2, M\} \left[\log(T+1)\log_+(3M/\alpha_2) + 2\right]$$

$$\leq 3\log^2(T+1) c_2 \max\{\alpha_2, M\} \left[\log_+(3M/\alpha_2) + 3\right],$$

where we've used that $2/\log(T+1) \leq 3$ for $T \geq 1$. Hence, we have the stated bound:

$$\sum_{t=1}^{T} \varphi_t(u_t) = r_t(w; c_1, \alpha_1, p_1) + r_t(w; c_2, \alpha_2, p_2)$$

$$\leq 4c_1 \sqrt{\alpha_1^2 + \sum_{t=1}^{T} \|u_t\|^2 \log\left(e + \frac{e \sum_{t=1}^{T} \|u_t\|^2}{\alpha_1^2}\right)} + 3c_2 \log^2(T+1) \max\{\alpha_2, M\} \left[\log_+(3M/\alpha_2) + 3\right].$$

$\square$

## E.2. Refined Dynamic Regret Algorithm for OCO

In this section we provide a dynamic regret algorithm for OLO which satisfies the conditions of Theorem E.1. The result is modest adaptation of the dynamic base algorithm first proposed by Jacobsen & Cutkosky (2022), with minor adjustments in the constants to allow the desired cancellations required in Theorem E.1 to occur. We present the base algorithm for the general constrained case for generality, though in our applications we will simply instantiate the algorithm unconstrained ($\mathcal{W} = \mathbb{R}^d$) and with $w_1 = \mathbf{0}$, in which case the update in Algorithm 5 reduces to $w_{t+1} = \widetilde{w}_{t+1} = \frac{\theta_t}{\|\theta_t\|} \alpha \left[\exp\left(\frac{k}{\eta}\left[\|\theta_t\| - \frac{\eta}{2}\|g_t\|^2 - \gamma\right]\right) - 1\right]_+$ and the guarantees in Theorem E.6 scales with $M = \max_t \|u_t\|$ and $\sum_{t=1}^{T} \|g_t\|^2 \|u_t\|$, as in Jacobsen & Cutkosky (2022).

Our result has a few other qualities which might be of independent interest. We provide a mild refinement of their result which avoids most factors of $M = \max_t \|u_t\|$ and replaces the global $\mathcal{O}((M + P_T)\log(MT))$ penalties reported in Jacobsen & Cutkosky (2022) with a refined *log-linear* penalties $\Phi_T(\lambda) + P_T^\Phi(\lambda)$, where

$$\Phi_T(\lambda) = \|u_T\| \log(\lambda\|u_T\| + 1), \qquad P_T^\Phi(\lambda) := \sum_{t=2}^{T} \|u_t - u_{t-1}\| \log(\lambda\|u_t - u_{t-1}\| + 1).$$

**Algorithm 5** Refined Dynamic Base Algorithm for OLO

---

**Input:** Non-empty convex domain $\mathcal{W} \subseteq \mathbb{R}^d$, initial point $w_1 \in \mathcal{W}$, parameters $\alpha, \eta, \gamma > 0$ and $k \geq 1$
**Define:** Regularizer $\psi(w) = \frac{k}{\eta} \int_0^{\|w - w_1\|} \log\left(x/\alpha + 1\right) dx$ and threshold operation $[x]_+ = \max\{x, 0\}$
**for** $t = 1$ **to** $T$ **do**
   Play $w_t$, observe $g_t \in \partial f_t(w_t)$
   Set $\varphi_t(w) = (\frac{\eta}{2}\|g_t\|^2 + \gamma)\|w - w_1\|$
   Set $\theta_t = \frac{k}{\eta} \log\left(\|w_t - w_1\|/\alpha + 1\right) \frac{w_t - w_1}{\|w_t - w_1\|} - g_t$
   Update

$$\widetilde{w}_{t+1} = \arg\min_{w \in \mathbb{R}^d} \langle g_t, w \rangle + \varphi_t(w) + D_\psi(w|w_t)$$
$$= w_1 + \frac{\theta_t}{\|\theta_t\|} \alpha \left[\exp\left(\frac{\eta}{k}\left[\|\theta_t\| - \frac{\eta}{2}\|g_t\|^2 - \gamma\right]\right) - 1\right]_+$$
$$w_{t+1} = \arg\min_{w \in \mathcal{W}} D_\psi(w|\widetilde{w}_{t+1})$$

**end for**

---

Our analysis also streamlines the analysis of Jacobsen & Cutkosky (2022) by using simple fixed hyperparameter settings for $\alpha$ and $\gamma$, contrasting the time-varying choices used in the original work. The main appeal of their time-varying hyperparameter choices is that they result in a horizon-independent base algorithm; however, this benefit is limited in application since full algorithm has to maintain a collection of learners $\{\mathcal{A}_{\eta_i}\}$ for $\eta_i = 2^i/L\sqrt{T}$, which makes the full algorithm horizon-dependent either way. Hence, here we focus on a simple fixed hyperparameter setting of $\alpha$ and $\gamma$.

**Theorem E.6.** *For any sequence $f_1, \ldots, f_T$ of L-Lipschitz convex loss functions and any sequence $u_{1:T} = (u_1, \ldots, u_T)$ in $\mathcal{W}$, Algorithm 5 with $\eta \leq 1/L$ and $k \geq 4$ guarantees*

$$R_T(u_{1:T}) \leq \frac{2k\left[\Phi\left(\|u_T - w_1\|, \frac{1}{\alpha}\right) + P_T^\Phi\left(\frac{k}{\eta\alpha\gamma}\right)\right]}{2\eta} + \frac{\eta}{2}\sum_{t=1}^{T}\|g_t\|^2\|u_t - w_1\| + \gamma\sum_{t=1}^{T}\|u_t - w_1\| + \eta\alpha\sum_{t=1}^{T}\|g_t\|^2,$$

*where $g_t \in \partial f_t(w_t)$ for all $t$ and we define $\Phi(x, \lambda) = x\log\left(\lambda x + 1\right)$ and the $\Phi$-path-length $P_T^\Phi(\lambda) = \sum_{t=2}^{T} \Phi(\|u_t - u_{t-1}\|, \lambda)$. Moreover, for any $\epsilon > 0$, setting $\alpha = \frac{\epsilon}{T}$, $\gamma = \frac{L}{T}$, $k = 4$, and $\frac{1}{LT} \leq \eta \leq \frac{1}{L}$ ensures that*

$$R_T(u_{1:T}) \leq L(M + \epsilon) + \frac{8\left[\Phi\left(\|u_T - w_1\|, \frac{T}{\epsilon}\right) + P_T^\Phi\left(\frac{4T^3}{\epsilon}\right)\right]}{2\eta} + \frac{\eta}{2}\sum_{t=1}^{T}\|g_t\|^2\|u_t - w_1\|,$$

*where $M = \max_t \|u_t - w_1\|$.*

*Proof.* We have via the dynamic regret guarantee of mirror descent (see, e.g., Jacobsen & Cutkosky (2022, Lemma 1)) that

$$R_T(u_{1:T}) \leq \sum_{t=1}^{T} \langle g_t, w_t - u_t \rangle$$
$$\leq \psi(u_T) + \sum_{t=1}^{T} \varphi_t(u_t) + \sum_{t=2}^{T} \langle \nabla\psi(w_t) - \nabla\psi(w_1), u_t - u_{t-1} \rangle$$
$$+ \sum_{t=1}^{T} \langle g_t, w_t - w_{t+1} \rangle + D_\psi(w_{t+1}|w_t) - \varphi_t(w_{t+1})$$
$$= \psi(u_T) + \sum_{t=1}^{T} \varphi_t(u_t) + \sum_{t=2}^{T} \underbrace{\langle \nabla\psi(w_t), u_t - u_{t-1} \rangle - \gamma\|w_t - w_1\|}_{=:\rho_t}$$
$$+ \sum_{t=1}^{T} \underbrace{\langle g_t, w_t - w_{t+1} \rangle + D_\psi(w_{t+1}|w_t) - \eta\|g_t\|^2\|w_{t+1} - w_1\|}_{=:\delta_t}$$

where $g_t \in \partial \ell_t(w_t)$ for each $t$. The terms $\rho_t$ can be bound using Fenchel-Young inequality. Let $f(x) = \frac{k}{\eta} \int_0^x \log\left(\frac{kv}{\alpha\eta\gamma} + 1\right) dv$; by direct calculation we have $f^*(\theta) = \alpha\gamma\left(\exp\left(\frac{\eta}{k}\theta\right) - \frac{\eta}{k}\theta\right)$. Hence, by Fenchel-Young inequality we have

$$\sum_{t=2}^T \rho_t \leq \sum_{t=2}^T \|\nabla\psi(w_t)\| \|u_t - u_{t-1}\| - \gamma\|w_t - w_1\| \leq \sum_{t=2}^T f(\|u_t - u_{t-1}\|) + f^*(\|\nabla\psi(w_t)\|) - \gamma\|w_t - w_1\|$$

$$\leq \sum_{t=2}^T f(\|u_t - u_{t-1}\|) + \alpha\gamma\left(\exp\left(\frac{\eta}{k}\frac{k}{\eta}\log\left(\|w_t - w_1\|/\alpha + 1\right)\right) - \frac{\eta}{k}\|\nabla\psi(w_t)\|\right) - \gamma\|w_t - w_1\|$$

$$\leq \sum_{t=2}^T f(\|u_t - u_{t-1}\|) + \alpha\gamma\left(\frac{\|w_t - w_1\|}{\alpha} + 1\right) - \gamma\|w_t - w_1\| = \sum_{t=2}^T \left(f(\|u_t - u_{t-1}\|) + \alpha\gamma\right)$$

$$\leq (T-1)\alpha\gamma + \sum_{t=2}^T \frac{k\|u_t - u_{t-1}\| \log\left(\frac{k\|u_t - u_{t-1}\|}{\alpha\eta\gamma} + 1\right)}{\eta}$$

where the last inequality uses $\int_0^x F(v)dv \leq xF(x)$ for non-decreasing $F(x)$.

For the stability terms $\sum_{t=1}^T \delta_t$, first observe that the regularizer is $\psi(w) = \Psi(\|w - w_1\|) = \frac{k}{\eta}\int_0^{\|w-w_1\|} \log(x/\alpha + 1)dx$, where $\Psi$ satisfies

$$\Psi'(x) = \frac{k}{\eta}\log\left(x/\alpha + 1\right)$$

$$\Psi''(x) = \frac{k}{\eta(x+\alpha)}$$

$$\Psi'''(x) = \frac{-k}{\eta(x+\alpha)^2},$$

hence, for $k \geq 4$ we have $|\Psi'''(x)| \leq \frac{\eta/2}{2}\Psi''(x)^2$ for all $x \geq 0$, so by Jacobsen & Cutkosky (2022, Lemma 2) with $\eta_t(\|w - w_1\|) := \eta/2$ we have that

$$\sum_{t=1}^T \delta_t = \sum_{t=1}^T \langle g_t, w_t - w_{t+1}\rangle - D_\psi(w_{t+1}|w_t) - \eta\|g_t\|^2\|w_{t+1} - w_1\| \leq \frac{2\|g_t\|^2}{\Psi''(0)} = \sum_{t=1}^T \frac{2\eta\alpha}{k}\|g_t\|^2 \leq \frac{\eta\alpha}{2}\sum_{t=1}^T \|g_t\|^2.$$

Plugging the bounds for $\sum_{t=1}^T \rho_t$ and $\sum_{t=1}^T \delta_t$ back into the regret bound and expanding the definition of $\varphi_t(u_t)$ yields

$$R_T(u_{1:T}) \leq \psi(u_T) + \sum_{t=1}^T \varphi_t(u_t) + \sum_{t=2}^T \rho_t + \sum_{t=1}^T \delta_t$$

$$\leq \frac{k\|u_T - w_1\|\log\left(\frac{\|u_T - w_1\|}{\alpha} + 1\right) + k\sum_{t=2}^T \|u_t - u_{t-1}\|\log\left(\frac{k\|u_t - u_{t-1}\|}{\alpha\eta\gamma} + 1\right)}{\eta}$$

$$+ \frac{\eta}{2}\sum_{t=1}^T \|g_t\|^2\|u_t - w_1\| + \gamma\sum_{t=1}^T \|u_t - w_1\| + \frac{\eta\alpha}{2}\sum_{t=1}^T \|g_t\|^2 + (T-1)\alpha\gamma$$

$$\leq \frac{k\left[\Phi\left(\|u_T - w_1\|, \frac{1}{\alpha}\right) + P_T^\Phi\left(\frac{k}{\alpha\eta\gamma}\right)\right]}{\eta}$$

$$+ \frac{\eta}{2}\sum_{t=1}^T \|g_t\|^2\|u_t - w_1\| + \gamma\sum_{t=1}^T \|u_t - w_1\| + \frac{\eta\alpha}{2}\sum_{t=1}^T \|g_t\|^2 + (T-1)\alpha\gamma,$$

where we've defined $\Phi(x, \lambda) = x\log\left(\lambda x + 1\right)$ and the $\Phi$-path-length $P_T^\Phi(\lambda) = \sum_{t=2}^T \Phi(\|u_t - u_{t-1}\|, \lambda) = \sum_{t=2}^T \|u_t - u_{t+1}\|\log\left(\lambda\|u_t - u_{t+1}\| + 1\right)$. Moreover, for any $\epsilon > 0$, setting $\alpha = \epsilon/T$, $\gamma = L/T$, and $\frac{1}{LT} \leq \eta \leq \frac{1}{L}$, we have

---

**Algorithm 6** Dynamic Algorithm for Unconstrained OLO

---

**Input:** $\epsilon > 0$, $\mathcal{S} = \left\{ \eta_i = \frac{2^i}{TL} \wedge \frac{1}{L} : i = 0, 1, \ldots \right\}$

**Initialize:** $\mathcal{A}_\eta$ implementing Algorithm 5 on $\mathcal{W} = \mathbb{R}^d$ with $\alpha = \epsilon/T$ and $\gamma = L/T$ for each $\eta \in \mathcal{S}$

**for** $t = 1$ **to** $T$ **do**

    Get output $w_t^\eta \in \mathbb{R}^d$ from $\mathcal{A}_\eta$ for all $\eta \in \mathcal{S}$

    Play $w_t = \sum_{\eta \in \mathcal{S}} w_t^\eta$, observe $g_t \in \partial f_t(w_t)$

    Pass $g_t$ to $\mathcal{A}_\eta$ for all $\eta \in \mathcal{S}$

**end for**

---

$k/\alpha\eta\gamma \le 4T^3/\epsilon$ and

$$R_T(u_{1:T}) \le \frac{k\left[\Phi(\|u_T\|, \frac{\epsilon}{T}) + P_T^\Phi\left(\frac{kT^3}{\epsilon}\right)\right]}{\eta} + \frac{\eta}{2}\sum_{t=1}^T \|g_t\|^2\|u_t - w_1\| + \frac{L}{T}\sum_{t=1}^T \|u_t - w_1\| + \frac{\epsilon}{2LT}\sum_{t=1}^T \|g_t\|^2 + \frac{\epsilon}{T}\frac{L}{T}(T-1)$$

$$\le L(M + \epsilon)$$

$$+ \frac{8\left[\|u_T\|\log\left(\frac{\|u_T\|T}{\epsilon} + 1\right) + \sum_{t=1}^{T-1}\|u_t - u_{t+1}\|\log\left(\frac{4\|u_t - u_{t+1}\|T^3}{\epsilon} + 1\right)\right]}{2\eta} + \frac{\eta}{2}\sum_{t=1}^T \|g_t\|^2\|u_t - w_1\|. \quad \square$$

For completeness we also provide the tuned guarantee for the unconstrained setting, obtained by running Algorithm 5 with step-size $\eta$ for each $\eta \in \left\{2^i/LT \wedge 1/L, i = 0, 1, \ldots\right\}$ and adding the resulting iterates together.

**Theorem E.7.** *For any sequence of $L$-Lipschitz convex functions $f_1, \ldots, f_T$ and any sequence $u_{1:T} = (u_1, \ldots, u_T)$ in $\mathbb{R}^d$, Algorithm 6 guarantees*

$$R_T(u_{1:T}) \le 4L\left(|\mathcal{S}|\epsilon + M + \Phi\left(\|u_T\|, \frac{T}{\epsilon}\right) + P_T^\Phi\left(\frac{4T^3}{\epsilon}\right)\right) + 2\sqrt{2\left(\Phi\left(\|u_T\|, \frac{T}{\epsilon}\right) + P_T^\Phi\left(\frac{4T^3}{\epsilon}\right)\right)\sum_{t=1}^T \|g_t\|^2\|u_t\|}$$

*where $\Phi(x, \lambda) = x\log(\lambda x + 1)$ and $P_T^\Phi(\lambda) = \sum_{t=2}^T \Phi(\|u_t - u_{t-1}\|, \lambda)$.*

*Proof.* Observe that for any $\eta_i \in \mathcal{S}$, we have

$$R_T(u_{1:T}) \le \sum_{t=1}^T \langle g_t, w_t - u_t \rangle = \sum_{t=1}^T \langle g_t, w_t^{\eta_i} - u_t \rangle + \sum_{\eta_j \ne \eta_i}\sum_{t=1}^T \langle g_t, w_t^{\eta_j} \rangle$$

$$= R_T^{\mathcal{A}_{\eta_i}}(u_{1:T}) + \sum_{\eta_j \ne \eta_i} R_T^{\mathcal{A}_j}(\mathbf{0}),$$

where $g_t \in \partial\ell_t(w_t)$ for all $t$ and $R_T^{\mathcal{A}_j}(u_{1:T}) = \sum_{t=1}^T \langle g_t, w_t^{\eta_j} - u_t \rangle$ denotes the dynamic regret of $\mathcal{A}_j$. Hence, applying Theorem E.6 and observing that $R_T^{\mathcal{A}_j}(\mathbf{0}) \le L\epsilon$ for any $\eta_j$, we have

$$R_T(u_{1:T}) \le R_T^{\mathcal{A}_i}(u_{1:T}) + (|\mathcal{S}| - 1)L\epsilon$$

$$\le L(|\mathcal{S}|\epsilon + M) + \frac{8(\Phi_T + P_T^\Phi)}{2\eta_i} + \frac{\eta_i}{2}\sum_{t=1}^T \|g_t\|^2\|u_t\|,$$

where we denote $\Phi_T = \Phi(\|u_T\|, T/\epsilon)$ and $P_T^\Phi = P_T^\Phi\left(\frac{4T^3}{\epsilon}\right) = \sum_{t=2}^T \Phi(\|u_t - u_{t-1}\|, 4T^3/\epsilon)$. Now applying Lemma I.3

we have

$$R_T(u_{1:T}) \leq L(|\mathcal{S}| \epsilon + M) + 2\sqrt{2(\Phi_T + P_T^\Phi) \sum_{t=1}^T \|g_t\|^2 \|u_t\|} + \frac{8(\Phi_T + P_T^\Phi)}{2\eta_{\max}} + \frac{\eta_{\min}}{2} \sum_{t=1}^T \|g_t\|^2 \|u_t\|$$

$$\leq L(|\mathcal{S}| \epsilon + M) + 2\sqrt{2(\Phi_T + P_T^\Phi) \sum_{t=1}^T \|g_t\|^2 \|u_t\| + 4L(\Phi_T + P_T^\Phi) + LM}$$

$$\leq 4L\left(|\mathcal{S}| \epsilon + M + \Phi_T + P_T^\Phi\right) + 2\sqrt{2(\Phi_T + P_T^\Phi) \sum_{t=1}^T \|g_t\|^2 \|u_t\|} \qquad \square$$

## F. Proof of Theorem 5.2

We recall the theorem before detailing its proof.

**Theorem 5.2** (Lower bound on the unit ball). *Assume $T \geq 4d$. Then, for any algorithm $\mathcal{A}$ playing actions $z_1, \ldots, z_T$ in the unit Euclidean ball it holds that*

1. *There exists a parameter $\theta \in \mathbb{R}^d$ and a sub-Gaussian distribution $\mathbb{P}_\theta$, such that $\mathbb{E}_{\ell \sim \mathbb{P}_\theta}[\ell] = \theta$, $\mathbb{E}_{\ell \sim \mathbb{P}_\theta}[\|\ell\|^2] \leq 1$, for which it holds that*

$$R_T^{sto}(\mathcal{A}, \theta) := \mathbb{E}_{(\ell_t)_{t=1}^T \sim \mathbb{P}_\theta^T}\left[R_T^{\mathcal{Z}}\left(\frac{\theta}{\|\theta\|}\right)\right] \geq \frac{\sqrt{dT}}{64} \wedge \frac{T}{12d} \ .$$

2. *There exists a sequence of losses $\ell_1, \ldots, \ell_T$ satisfying $\|\ell_t\| \leq 1$ for all $t \geq 1$, a comparator $u \in \mathbb{B}_d$, and an absolute constant $C$ such that*

$$R_T^{\mathcal{Z}}(u) \geq C \cdot \left\{\sqrt{dT} \wedge \frac{T}{d}\right\} \cdot \sqrt{\frac{d}{d \vee \log(T)}} \ .$$

*Proof.* The proof follows the general outline of Theorem 24.2 of (Lattimore & Szepesvári, 2020), which proves a $\Omega(d\sqrt{T})$ lower bound on the regret for a slightly different feedback model. In their setting, at time $t \geq 1$, after playing $x_t \in \mathbb{B}_d$ the learner receives

$$\ell_t(x_t) = \langle \theta, x_t \rangle + \epsilon_t, \quad \text{where } \epsilon_t \sim \mathcal{N}(0, 1),$$

and $\theta \in \Theta$ is a fixed parameter from some class of parameters $\Theta$. The proof uses that, up to horizon $T$, it is difficult for the learner to distinguish parameters $\theta$ from $\theta'$, if $\Theta$ is a small hypercube centered in the origin. In the following, we keep a similar class of parameters $\Theta$, but introduce key changes in the arguments to tackle different feedback models and constraints on the losses.

**Stochastic model** We fix a constant $\Delta = \frac{1}{8\sqrt{T}}$, and consider the class of parameters $\Theta = \{\pm\Delta\}^d$. By assumption, $T \geq 4d$, so $\|\theta\|^2 \leq \frac{1}{2}$ for any $\theta \in \Theta$. To satisfy the stochastic assumptions of the theorem, we assume that losses are generated as follows:

1. Before the interaction, the adversary chooses at random a parameter $\theta \in \Theta$.

2. for each time step $t \geq 1$, the adversary samples $\ell_t = \theta + \epsilon_t$, with $\epsilon_t \sim \mathcal{N}\left(0, \frac{1}{2d} I_d\right)$, and the learner observes feedback $\langle \ell_t, x_t \rangle$.

By construction, losses are sub-Gaussian and satisfy $\mathbb{E}\left[\|\ell_t\|^2\right] \leq \|\theta\|^2 + \mathbb{E}\left[\|\epsilon_t\|^2\right] \leq 1$ for all $t \geq 1$.

Then, similarly to (Lattimore & Szepesvári, 2020), for any $i \in [d]$ we define the stopping time

$$\tau_i := T \wedge \min\left\{t \geq 1 : \sum_{s=1}^t x_{si}^2 \geq \frac{T}{d} - 1\right\}.$$

In their proof, the threshold $\frac{T}{d}$ intuitively represents the quantity of information necessary to confidently identify the sign of $\theta_i$ when $|\theta_i| \propto \sqrt{\frac{d}{T}}$. In our case, the same intuition holds with a gap proportional to $1/\sqrt{T}$ because the variance is smaller by a factor $d$, so we don't have to modify the definition of $\tau_i$ (we just added $-1$ to simplify computations).

For any algorithm $\mathcal{A}$ and $\theta \in \Theta$, we denote by $R_T(\mathcal{A}, \theta)$ the regret of $\mathcal{A}$ against the comparator $u_\theta = -\frac{\mathrm{sgn}(\theta)}{\sqrt{d}}$. It holds that

$$
\begin{aligned}
R_T(\mathcal{A}, \theta) &= \Delta \, \mathbb{E}_\theta \left[ \sum_{t=1}^{T} \sum_{i=1}^{d} \left( \frac{1}{\sqrt{d}} + x_{ti} \, \mathrm{sgn}(\theta_i) \right) \right] \\
&\geq \frac{\Delta \sqrt{d}}{2} \, \mathbb{E}_\theta \left[ \sum_{t=1}^{T} \sum_{i=1}^{d} \left( \frac{1}{\sqrt{d}} + x_{ti} \, \mathrm{sgn}(\theta_i) \right)^2 \right] \\
&\geq \frac{\Delta \sqrt{d}}{2} \sum_{i=1}^{d} \mathbb{E}_\theta \left[ \sum_{t=1}^{\tau_i} \left( \frac{1}{\sqrt{d}} + x_{ti} \, \mathrm{sgn}(\theta_i) \right)^2 \right],
\end{aligned}
\tag{20}
$$

where the first inequality comes from the fact that for all steps $t \geq 1$,

$$
\begin{aligned}
\sum_{i=1}^{d} \left( \frac{1}{\sqrt{d}} + x_{ti} \, \mathrm{sgn}(\theta_i) \right)^2 &= 1 + \sum_{i=1}^{d} \frac{2}{\sqrt{d}} x_{ti} \, \mathrm{sgn}(\theta_i) + \|x_t\|^2 \\
&\leq 2 + \sum_{i=1}^{d} \frac{2}{\sqrt{d}} x_{ti} \, \mathrm{sgn}(\theta_i) \\
&= \frac{2}{\sqrt{d}} \sum_{i=1}^{d} \left( \frac{1}{\sqrt{d}} + x_{t,i} \, \mathrm{sgn}(\theta_i) \right) .
\end{aligned}
$$

Note that this inequality is an equality if $\|x_t\| = 1$.

Then, for any $i \in [d]$ and $\sigma \in \{\pm 1\}$ we define

$$
U_i(\sigma) := \sum_{t=1}^{\tau_i} \left( \frac{1}{\sqrt{d}} + \sigma \cdot x_{ti} \right)^2,
$$

and we verify that

$$
\begin{aligned}
U_i(\sigma) &\leq 2 \sum_{t=1}^{\tau_i} \frac{1}{d} + 2 \sum_{t=1}^{\tau_i} x_{ti}^2 \\
&\leq 2 \left( \frac{\tau_i}{d} + \frac{T}{d} \right) \leq 4 \frac{T}{d} .
\end{aligned}
$$

Let $\theta' \in \Theta$ be such that $\theta_j = \theta'_j$ for $j \neq i$ and $\theta'_i = -\theta_i$. Assume without loss of generality that $\theta_i > 0$. Let $\mathbb{P}$ and $\mathbb{P}'$ be the laws of $U_i(1)$ under the bandit/learner interaction induced by $\theta$ and $\theta'$, respectively. Then, using Pinsker inequality we obtain that

$$
\mathbb{E}_\theta[U_i(1)] \geq \mathbb{E}_{\theta'}[U_i(1)] - \mathrm{essup}\, U_i(1) \cdot \mathrm{TV}(\mathbb{P}, \mathbb{P}')
\tag{21}
$$

$$
\geq \mathbb{E}_{\theta'}[U_i(1)] - \frac{4T}{d} \sqrt{\frac{1}{2} \mathrm{KL}(\mathbb{P}, \mathbb{P}')}
\tag{22}
$$

Then, we use that at each step $t$ the distribution of the observation under $\mathbb{P}$ follows a Gaussian distribution $\mathcal{N}\left( \langle \theta, x_t \rangle, \frac{\|x_t\|^2}{2d} \right)$, while it follows a Gaussian distribution $\mathcal{N}\left( \langle \theta', x_t \rangle, \frac{\|x_t\|^2}{2d} \right)$ under model $\mathbb{P}'$. Thus, using the chain rule for the relative entropy up to a stopping time, we have

$$
\mathrm{KL}(\mathbb{P}, \mathbb{P}') = \mathbb{E}_\theta \left[ \sum_{t=1}^{\tau_i} \frac{d}{\|x_t\|^2} \langle \theta - \theta', x_t \rangle^2 \right] = 4 \Delta^2 d \cdot \mathbb{E}_\theta \left[ \sum_{t=1}^{\tau_i} \frac{x_{ti}^2}{\|x_t\|^2} \right] .
$$

We can identify two differences compared to the analogous proof step for Theorem 24.2 of (Lattimore & Szepesvári, 2020) (Eq. 24.4). First, we can observe a supplementary $d$ factor due to the $d^{-1}$ term in the noise variance, that will cause the scaling $\sqrt{dT}$ instead of $d\sqrt{T}$ in the final result. Secondly, the norm $\|x_t\|^2$ prevents us for showing that the expectation term scales in $T/d$ by using the definition of $\tau_i$ directly. However, it is clear that in this bounded setting playing an action with a small norm is sub-optimal.

We thus introduce $S_{\tau_i} = \sum_{i=1}^{\tau_i} \mathbb{I}(\|x_t\|^2 \leq 2/3)$. On each round where $\|x_t\|^2 \leq \frac{2}{3}$, the instantaneous regret against the comparator $u_\theta$ must be at least $1/6$: the comparator gets a reward of 1, while the learner gets a reward upper bounded by $\sqrt{2/3} \leq 0.82 \leq 1 - 1/6$. So, it must hold that $R_T(\mathcal{A}, \theta) \geq \frac{1}{6}\mathbb{E}_\theta[S_{\tau_i}]$. Meanwhile, using the definition of $S_{\tau_i}$ we can obtain that

$$\text{KL}(\mathbb{P}, \mathbb{P}') \leq 4\Delta^2 d \cdot \left(\frac{3}{2}\mathbb{E}\left[\sum_{t=1}^{\tau_i} x_{ti}^2\right] + \mathbb{E}[S_{\tau_i}]\right) \leq 4\Delta^2 d \cdot \left(\frac{3}{2}\frac{T}{d} + \mathbb{E}_\theta[S_{\tau_i}]\right) .$$

Combining these two results, we can use that either $\mathbb{E}_\theta[S_{\tau_i}] \geq \frac{T}{2d}$, in which case it holds that $R(T, \theta) \geq \frac{T}{12d}$, or it must hold that

$$\text{KL}(\mathbb{P}, \mathbb{P}') \leq 8\Delta^2 d\frac{T}{d} .$$

The first case yields the term $\frac{T}{12d}$ in the theorem, corresponding to an algorithm that would achieve linear regret because it can consistently play actions with too small of a norm under some instances. For the remainder of the proof, we focus on the second case. Plugging the above result in Eq. (22), we obtain that

$$\mathbb{E}_\theta[U_i(1)] \geq \mathbb{E}_{\theta'}[U_i(1)] - 8\Delta\frac{T}{d}\sqrt{T}$$

It follows that

$$\mathbb{E}_\theta[U_i(1)] + \mathbb{E}_{\theta'}[U_i(-1)] \geq \mathbb{E}_{\theta'}[U_i(1) + U_i(-1)] - 8\Delta\frac{T}{d}\sqrt{T}$$

$$= 2\,\mathbb{E}_{\theta'}\left[\frac{\tau_i}{d} + \sum_{t=1}^{\tau_i} x_{ti}^2\right] - 8\Delta\frac{T}{d}\sqrt{T}$$

$$\geq 2\left(\frac{T}{d} - 1\right) - 8\Delta\frac{T}{d}\sqrt{T} = \frac{T}{d} - 2, \tag{23}$$

since $\Delta = \frac{1}{8\sqrt{T}}$ and

$$U_i(1) + U_i(-1) = \sum_{t=1}^{\tau_i}\left[\left(\frac{1}{\sqrt{d}} + x_{ti}\right)^2 + \left(\frac{1}{\sqrt{d}} + x_{ti}\right)^2\right] = 2\sum_{t=1}^{\tau_i}\left(\frac{1}{d} + x_{ti}^2\right),$$

and that $\frac{\tau_i}{d} + \sum_{t=1}^{\tau_i} x_{ti}^2 \geq \frac{T}{d}$ by the definition of $\tau_i$. The proof is completed using the randomisation hammer,

$$\sum_{\theta \in \{\pm\Delta\}^d} R_T(\mathcal{A}, \theta) \geq \frac{\Delta\sqrt{d}}{2}\sum_{i=1}^{d}\sum_{\theta \in \{\pm\Delta\}^d} \mathbb{E}_\theta[U_i(\text{sgn}(\theta_i))]$$

$$= \frac{\Delta\sqrt{d}}{2}\sum_{i=1}^{d}\sum_{\theta_{-i} \in \{\pm\Delta\}^{d-1}}\sum_{\theta_i \in \{\pm\Delta\}} \mathbb{E}_\theta[U_i(\text{sgn}(\theta_i))]$$

$$\geq \frac{\Delta\sqrt{d}}{2}\sum_{i=1}^{d}\sum_{\theta_{-i} \in \{\pm\Delta\}^{d-1}}\left(\frac{T}{d} - 2\right) = 2^{d-2}(T - 2d)\Delta\sqrt{d} .$$

Hence, assuming that $T \geq 4d$, there exists $\theta \in \{\pm\Delta\}^d$ such that

$$R_T(\mathcal{A}, \theta) \geq \frac{T}{2} \cdot \frac{\Delta\sqrt{d}}{4} = \frac{\sqrt{dT}}{64}.$$

This gives the second lower bound on the constant $C_{d,T}$ in the statement of the theorem.

**Adversarial environment with bounded losses** The proof for this case is largely adapted from the previous proof, that we refer to as the "stochastic case" in the following for simplicity, although the proof still builds on stochastically generated losses. We still assume that

1. The adversary selects a parameter $\theta \in \Theta$ before the interaction.

2. At step $t$, it draws a loss $\widetilde{\ell}_t = \theta + \epsilon_t$, where $\epsilon_t \sim \mathcal{N}(0, \sigma_d^2 I_d)$, for some $\sigma_d > 0$.

but we make two changes. First we change the noise level $\sigma_d^2$ from $\frac{1}{2d}$ to something smaller. Secondly, we make the adversary select a *clipped* version of this random loss $\ell_t = \widetilde{\ell}_t \mathbb{I}\big(\|\widetilde{\ell}_t\| \leq 1\big)$. The intuition is that clipping will enforce a bounded norm almost surely.

However, a core ingredient of the proof we will be to calibrate the noise level $\sigma_d^2$ in order to make clipping very unlikely, so that the statistical properties of this model will be very close to the Gaussian stochastic model that we already studied. Furthermore, we will use that any rescaling of the variance propagates easily in the previous proof, as it only appears in the KL term induced after using Pinsker inequality, and thus propagates naturally to the choice of the gap $\Delta$.

To start the proof, we first show that we can express the regret $\widetilde{R}_T(\mathcal{A}, \theta)$ on the clipped environment, against the comparator $u_\theta$, as a function of the regret $R_T(\mathcal{A}, \theta)$ as defined in the unclipped environment. Assume this time that $\|\theta\|^2 \leq \frac{1}{4}$. Under this condition, we can write that

$$\widetilde{R}_T(\mathcal{A}, \theta) = \mathbb{E}_\theta \left[ \sum_{t=1}^T \left\langle x_t - u_\theta, \widetilde{\ell}_t \mathbb{I}\big(\|\widetilde{\ell}_t\| \leq 1\big) \right\rangle \right]$$

$$= \mathbb{E}_\theta \left[ \sum_{t=1}^T \left\langle x_t - u_\theta, \widetilde{\ell}_t \right\rangle \right] + \mathbb{E}_\theta \left[ \sum_{t=1}^T \left\langle x_t - u_\theta, \widetilde{\ell}_t \mathbb{I}\big(\|\widetilde{\ell}_t\| \geq 1\big) \right\rangle \right]$$

$$= R_T(\mathcal{A}, \theta) + \mathbb{E}_\theta \left[ \sum_{t=1}^T \left\langle x_t - u_\theta, \widetilde{\ell}_t \mathbb{I}\big(\|\widetilde{\ell}_t\| \geq 1\big) \right\rangle \right]$$

$$\geq R_T(\mathcal{A}, \theta) - 4\mathbb{E}_\theta \left[ \sum_{t=1}^T \|\widetilde{\ell}_t\|^2 \mathbb{I}\big(\|\widetilde{\ell}_t\| \geq 1\big) \right]$$

$$\geq R_T(\mathcal{A}, \theta) - 8\mathbb{E}_\theta \left[ \sum_{t=1}^T (\|\theta\|^2 + \|\epsilon_t\|^2) \mathbb{I}\big(\|\widetilde{\ell}_t\| \geq 1\big) \right]$$

$$\geq R_T(\mathcal{A}, \theta) - 16T \cdot \mathbb{E}_\theta \left[ \|\epsilon_1\|^2 \mathbb{I}\big(\|\epsilon_1\|^2 \geq \frac{1}{4}\big) \right],$$

where in the last line we used that all terms of the sum have the same expectation, and that $\mathbb{I}\big(\|\widetilde{\ell}_1\| \geq 1\big) \leq \mathbb{I}\big(\|\epsilon_1\|^2 \geq \frac{1}{4}\big)$ and that under this event $\|\theta\| \leq \|\epsilon_1\|$. We leave this term for now, and focus on lower bounding $R_T(\mathcal{A}, \theta)$ by using the proof outline introduced for the first lower bound we proved (in the stochastic model).

Then, we can again lower bound the term $R_T(\mathcal{A}, \theta)$ with Eq. (20) and use the same terms $U_i(\sigma)$. However, some care is needed to adapt Equation (21).

We introduce the notation $\mathbb{E}_{\widetilde{\theta}}[U_i(\sigma)]$ to denote the expectation of $U_i(\sigma)$ if the learner was provided the *untruncated* losses $(\widetilde{\ell}_t)_{t \geq 1}$ at each time step under the environment defined by $\theta$, and similarly for $\mathbb{E}_{\widetilde{\theta'}}[U_i(\sigma)]$.

Using these definitions, our goal is to obtain an inequality involving $\mathbb{E}_\theta[U_i(1)]$ and $\mathbb{E}_{\theta'}[U_i(1)]$ is a similar way as Equation (22). However, a subtlety is that algorithm $\mathcal{A}$ may not be able to handle unbounded values for $x_t^\top \widetilde{\ell}_t$. Thus, we need to further define an extension of algorithm $\mathcal{A}$, that we denote by $\overline{\mathcal{A}}$.

We define $\overline{\mathcal{A}}$ as follows: whenever $x_t^\top \widetilde{\ell}_t > 1$, the algorithm skips its update and defines $x_{t+1} = x_t$, and otherwise uses the same update rule as $\mathcal{A}$. We then denote by $\mathcal{G}$ the event that no loss is clipped during the interaction: $\mathcal{G} = \{\forall t \in [T] : \ell_t = \widetilde{\ell}_t\}$. Under $\mathcal{G}$, it further holds that the outputs of algorithms $\mathcal{A}$ and $\overline{\mathcal{A}}$ match, so $\mathbb{E}_{\theta, \mathcal{A}}[U_i(\sigma)\mathbb{I}(\mathcal{G})] = \mathbb{E}_{\theta, \overline{\mathcal{A}}}[U_i(\sigma)\mathbb{I}(\mathcal{G})]$, and thus

$$\mathbb{E}_{\theta, \mathcal{A}}[U_i(\sigma)] \geq \mathbb{E}_{\theta, \overline{\mathcal{A}}}[U_i(\sigma)\mathbb{I}(\mathcal{G})] \,.$$

Then, using that $\mathbb{E}_{\theta,\overline{\mathcal{A}}}[U_i(\sigma)\mathbb{I}(\mathcal{G})] = \mathbb{E}_{\widetilde{\theta},\overline{\mathcal{A}}}[U_i(\sigma)\mathbb{I}(\mathcal{G})]$ we can further obtain that

$$
\begin{aligned}
\mathbb{E}_{\theta,\mathcal{A}}[U_i(\sigma)] &\geq \mathbb{E}_{\widetilde{\theta},\overline{\mathcal{A}}}[U_i(\sigma)\mathbb{I}(\mathcal{G})] \\
&= \mathbb{E}_{\widetilde{\theta}',\overline{\mathcal{A}}}[U_i(\sigma)\mathbb{I}(\mathcal{G})] + \mathbb{E}_{\widetilde{\theta},\overline{\mathcal{A}}}[U_i(\sigma)\mathbb{I}(\mathcal{G})] - \mathbb{E}_{\widetilde{\theta}',\overline{\mathcal{A}}}[U_i(\sigma)\mathbb{I}(\mathcal{G})] \\
&\geq \mathbb{E}_{\widetilde{\theta}',\overline{\mathcal{A}}}[U_i(\sigma)\mathbb{I}(\mathcal{G})] + \underbrace{\mathbb{E}_{\widetilde{\theta},\overline{\mathcal{A}}}[U_i(\sigma)] - \mathbb{E}_{\widetilde{\theta}',\overline{\mathcal{A}}}[U_i(\sigma)]}_{-V} - \mathbb{E}_{\widetilde{\theta}',\overline{\mathcal{A}}}[U_i(\sigma)\mathbb{I}(\overline{\mathcal{G}})] \\
&= \mathbb{E}_{\theta',\mathcal{A}}[U_i(\sigma)\mathbb{I}(\mathcal{G})] - V - \frac{2T}{d}\mathbb{P}(\mathcal{G}^c) \\
&\geq \mathbb{E}_{\theta',\mathcal{A}}[U_i(\sigma)] - V - \frac{4T}{d}\mathbb{P}(\mathcal{G}^c) ,
\end{aligned}
$$

where we used that $U_i(\sigma)$ is non-negative and bounded by $\frac{T}{d}$. We now remark that the term $V$ can be upper bounded by following the exact same steps as in the unclipped Gaussian environment, since algorithm $\overline{\mathcal{A}}$ can process unbounded feedback and receives losses of the form $\langle x_t, \theta + \epsilon_t \rangle$, with $\epsilon_t \sim \mathcal{N}(0, \sigma_d^2 I_d)$. The only difference is that the scaling factor $\sigma_d^2$ will replace $\frac{1}{2d}$.

Furthermore, while in previous proof $\Delta$ was tuned to make this term smaller than $\frac{T}{d}$, here we can choose it to ensure that $V \leq \frac{T}{2d}$, and also choose $\sigma_d^2$ so that $\frac{4T}{d}\mathbb{P}(\mathcal{G}^c) \leq \frac{T}{2d}$ too, so we can exactly recover Equation (23). It is clear that, assuming that the later bound holds, the desired result can be obtained by simply multiplying $\Delta \approx 1/\sqrt{T}$ by a factor of order $\sqrt{2d\sigma_d^2}$.

It remains to calibrate the noise level. By independence between time steps and Gaussianity of the noise, we have that

$$
\begin{aligned}
\mathbb{P}(\mathcal{G}^c) &\leq T\mathbb{P}(\|\widetilde{\ell}_1\| \geq 1) \\
&= T\mathbb{P}(\|\theta + \epsilon_1\|^2 \geq 1) \\
&\leq T\mathbb{P}(2\|\theta\|^2 + 2\|\epsilon_1\|^2 \geq 1) \\
&\leq T\mathbb{P}\left(\|\epsilon_1\|^2 \geq \frac{1}{8}\right),
\end{aligned}
$$

if we assume that $\|\theta\|^2 \leq \frac{1}{4}$. The last arguments of the proof rely on the Laurent-Massart inequality for chi-squared random variables Laurent & Massart (2000, Lemma 1). For all $t \geq 0$, it holds that

$$
\mathbb{P}\left(\|\varepsilon_1\|^2 \geq \sigma_d^2\left(d + 2\sqrt{dt} + 2t\right)\right) \leq e^{-t}. \tag{24}
$$

To convert this bound for a fixed threshold $x > 0$, we define $a := x/\sigma_d^2$, and remark that if $a \geq d$ we can set

$$
t_x := \left(\frac{\sqrt{2a-d} - \sqrt{d}}{2}\right)^2 , \quad \text{so that } a = d + 2\sqrt{dt_x} + 2t_x .
$$

Then, by (24),

$$
\mathbb{P}(\|\varepsilon\|^2 \geq x) \leq \exp(-t_x) = \exp\left(-\left(\frac{\sqrt{2\frac{x}{\sigma_d^2} - d} - \sqrt{d}}{2}\right)^2\right), \quad (\text{for } x \geq \sigma^2 d). \tag{25}
$$

We use this bound to first identify a variance level $\sigma_d^2$ guaranteeing $\frac{4T}{d}\mathbb{P}(\mathcal{G}^c) \leq \frac{T}{2d}$, so that this term fits easily in the proof framework of the stochastic case. To ensure this condition it suffices that $\mathbb{P}\left(\|\varepsilon\|^2 \geq \frac{1}{8}\right) \leq \frac{1}{8T}$, which by Eq. (25) can be achieved by choosing $\sigma_d^2$ as follows,

$$
\exp\left(-\left(\frac{\sqrt{\frac{1}{4\sigma_d^2} - d} - \sqrt{d}}{2}\right)^2\right) = \frac{1}{8T} \iff \sigma_d^2 = \frac{1}{4} \cdot \frac{1}{d + (\sqrt{d} + 2\sqrt{\log(8T)})^2} , \tag{26}
$$

which is of order $\frac{1}{d \vee \log(T)}$, yielding the rescaling factor introduced in the theorem.

Thus, to lower bound $\widetilde{R}_T(\mathcal{A}, \theta)$ it only remains to upper bound the term

$$E := 16T \cdot \mathbb{E}_\theta \left[ \|\epsilon_1\|^2 \mathbb{I}\left( \|\epsilon_1\|^2 \geq \frac{1}{4} \right) \right] .$$

We first rewrite the expectation as

$$\mathbb{E} \left[ \|\epsilon_1\|^2 \mathbb{I}\left( \|\epsilon_1\|^2 \geq \frac{1}{4} \right) \right] = \frac{1}{4} \mathbb{P}\left( \|\epsilon_1\|^2 \geq \frac{1}{4} \right) + \int_{\frac{1}{4}}^\infty \mathbb{P}(\|\epsilon_1\|^2 \geq u) \, \mathrm{d}u$$

$$\leq \frac{1}{32T} + \int_{\frac{1}{4}}^\infty \mathbb{P}(\|\epsilon_1\|^2 \geq u) \, \mathrm{d}u,$$

where we used that $\mathbb{P}\left( \|\epsilon_1\|^2 \geq \frac{1}{4} \right) \leq \mathbb{P}\left( \|\epsilon_1\|^2 \geq \frac{1}{8} \right) \leq \frac{1}{8T}$, by our design. Furthermore, the fact that we could already apply Eq. (24) with threshold $1/8$ guarantees that we can also apply it for any threshold $u$ larger than $1/4$, and the resulting concentration bound will be smaller than $\frac{1}{8T}$. Hence, for any threshold $u_0 \geq \frac{1}{4}$, we can upper bound the remaining integral as follows,

$$\int_{\frac{1}{4}}^\infty \mathbb{P}(\|\varepsilon_1\|^2 \geq u) \, \mathrm{d}u \leq \frac{u_0}{8T} + \int_{u_0}^\infty e^{-t_u} \, \mathrm{d}u$$

$$\leq \frac{u_0}{8T} + \int_{u_0}^\infty e^{-\frac{1}{4} \cdot \left( \sqrt{\frac{2u}{\sigma_d^2} - d} - \sqrt{d} \right)^2} \, \mathrm{d}u.$$

We then choose $u_0$ in order to simplify the integral computation. More explicitly, we choose $u_0$ to satisfy

$$\sqrt{\frac{2u}{\sigma_d^2} - d} - \sqrt{d} \geq \sqrt{\frac{u}{\sigma_d^2}}, \text{ for } u \geq u_0 .$$

We then solve, for $d > 0$ and $y \geq d/2$,

$$\sqrt{2y - d} - \sqrt{d} \geq \sqrt{y} \quad \Longleftrightarrow \quad \sqrt{2y - d} \geq \sqrt{y} + \sqrt{d}.$$

Squaring (both sides are nonnegative on the domain) gives

$$2y - d \geq y + d + 2\sqrt{yd} \quad \Longleftrightarrow \quad y - 2d \geq 2\sqrt{yd}.$$

In particular this forces $y \geq 2d$. Squaring again yields

$$(y - 2d)^2 \geq 4yd \quad \Longleftrightarrow \quad y^2 - 8dy + 4d^2 \geq 0.$$

Solving the quadratic equation $y^2 - 8dy + 4d^2 = 0$ gives the roots

$$y = \frac{8d \pm \sqrt{64d^2 - 16d^2}}{2} = d \left( 4 \pm 2\sqrt{3} \right).$$

Keeping the positive solution, we get

$$y \geq d(4 + 2\sqrt{3}) \Longleftrightarrow u \geq (4 + 2\sqrt{3}) \cdot d\sigma_d^2.$$

Using these results, we can choose $u_0 = 8d\sigma_d^2 \vee 4\sigma_d^2 \log(64T)$ and obtain that

$$E \leq \frac{1}{2} + 2u_0 + 16T \cdot \int_{u_0}^{+\infty} e^{-\frac{u}{4\sigma_d^2}} \, \mathrm{d}u$$

$$= \frac{1}{2} + 2u_0 + 64\sigma_d^2 T e^{-\frac{u_0}{4\sigma_d^2}}$$

$$\leq \frac{1}{2} + 16\sigma_d^2 \left\{ d \vee \frac{1}{2} \log(64T) \right\} + \sigma_d^2$$

$$\leq \frac{1}{2} + 4 \frac{\{d \vee \log(8T)\}}{d + 4\log(8T)} + \sigma_d^2$$

$$\leq 5 ,$$

where we used in the final step that $\sigma_d^2 \leq \frac{1}{2}$. Hence, we proved that the tuning of $\sigma_d^2$ from Equation (26) is sufficient to ensure that the bias term $E$ is upper bounded by a constant. This concludes the proof. $\qquad\square$

**Remark F.1.** *One might think that it could be possible to build hard instances based on simpler distributions, e.g. using Rademacher variables. However, the problem there is that simple constructions do not obtain the right properties. Everything is essentially in the balance between the maximum per-round regret/gain of the adversary and the difficulty to distinguish the instances (the KL term above).*

*For instance, if the adversary (1) sample a coordinate $I_t$ uniformly at random, and (2) returns a loss $\sigma e_{I_t}$ where $\sigma$ is a Rademacher variable with mean $\theta_i$ then:*

- *The KL term becomes $\mathcal{O}(\Delta^2 \tau_i / d) = \mathcal{O}(\Delta^2 T / d)$, while we had $(\Delta^2 d / T \sum_{t=1}^{\tau_i} x_{ti}^2 \leq \Delta^2 T$ above).*

- *But the gain of the adversary is defined by $d$ (one coordinate showed at a time, but the optimal comparator still plays $1/\sqrt{d}$ weight on each!).*

*So, overall balancing the two makes the $d$ cancel and we even just get $\sqrt{T}$. The same holds if instead of selecting a coordinate the adversary would just rescale the Rademacher variables by $1/\sqrt{d}$ because now the expected regret becomes multiplied by $1/\sqrt{d}$ (Eq. (20) has no more $\sqrt{d}$). Then, the KL becomes $Td\Delta^2$ essentially, so it's clear we get an even worse tradeoff.*

*And finally, Gaussian noise is the simplest distribution that allows us to use that $\sum_{t=1}^{\tau_i} x_{ti}^2 \leq \frac{T}{d}$*

## G. Regret of OSMD against a norm-adaptive adversary

In this section we develop the computations leading to our claim from Section 3.1 that OSMD only yields an $\mathcal{O}((dT)^{2/3})$ direction regret when used as a direction learner, under a norm-adaptive adversary. We can start the analysis from the first bound of their Theorem 6, which states that the regret of OSMD is upper bounded by

$$R_T \leq \gamma T + \frac{\log(\gamma^{-1})}{\eta} + \eta \sum_{t=1}^{T} \mathbb{E}\left[(1 - \|z_t\|)\|\widetilde{z}_t\|^2\right],$$

where $\gamma$ and $\eta$ are parameters chosen by the learner. When optimized, they yield $R_T \leq 3\sqrt{dT \log(T)}$ in the $\mathcal{F}_0$-measurable regime.

However, in the norm-adaptive case the norm $\|u\|$ must go inside the last expectation, giving a term $\eta \sum_{t=1}^{T} \mathbb{E}\left[\|u\|(1 - \|z_t\|)\|\widetilde{z}_t\|^2\right]$. This breaks the upper bound presented in the paper, because of the potential correlation between $\|u\|$ and each of the realizations $\widetilde{z}_t$. Because of this, we can only use the crude bound

$$\sum_{t=1}^{T} \mathbb{E}\left[\|u\|(1 - \|z_t\|)\|\widetilde{z}_t\|^2\right] \leq \mathbb{E}\left[\|u\| \sum_{t=1}^{T} \frac{d^2\|\ell_t\|^2}{1 - \|x_t\|}\right] \leq \frac{d^2 T}{\gamma}\mathbb{E}\left[\|u\|\right],$$

while the same term is upper bounded by $dT\|u\|$ in the $\mathcal{F}_0$-measurable case. In this case, choosing $\eta = \left(\frac{\log(T)}{dT}\right)^{\frac{2}{3}}$ and $\gamma = d\sqrt{\eta}$ give a regret bound of order $(dT)^{\frac{2}{3}}(\log(T))^{1/3}\mathbb{E}\left[\|u\|\right]$, which shows the degradation of the guarantees of OSMD as a direction learner, in the norm-adaptive setting. As a final remark, we highlight that this claim is based on plugging a conservative bound in the proof of (Bubeck et al., 2012), which doesn't prove that this result can't be improved with a more elaborate decomposition.

## H. Fenchel Conjugate Characterization of Comparator-Adaptive Bounds

The connection between Fenchel conjugates and regret bounds in online learning is well-established; see, e.g., Orabona (2019) for a textbook treatment and Cutkosky & Orabona (2018); Jacobsen & Cutkosky (2022); Zhang et al. (2022) for applications to parameter-free and comparator-adaptive algorithms.

Recall that the Fenchel conjugate of a function $f : \mathbb{R}^d \to \mathbb{R}$ is defined as

$$f^*(y) = \sup_{x \in \mathbb{R}^d} \left\{\langle x, y \rangle - f(x)\right\}.$$

Consider the regret defined as

$$R_T(u) = \sum_{t=1}^{T} \langle \ell_t, w_t - u \rangle = \sum_{t=1}^{T} \langle \ell_t^\top, w_t \rangle - \langle L_T, u \rangle,$$

where $L_T = \sum_{t=1}^{T} \ell_t$ denotes the cumulative loss vector.

Suppose we want to establish a comparator-adaptive bound of the form $R_T(u) \leq B_T(u)$ for all $u$, where $B_T : \mathbb{R}^d \to \mathbb{R}_+$ is some bound function (e.g., $B_T(u) = \mathcal{O}(\|u\|\sqrt{T \log(\|u\|T)})$).

The condition "$R_T(u) \leq B_T(u)$ for all $u$" can be rewritten as:

$$\forall u: \quad \sum_{t=1}^{T} \langle \ell_t, w_t \rangle - \langle L_T, u \rangle \leq B_T(u)$$

$$\iff \quad \sum_{t=1}^{T} \langle \ell_t, w_t \rangle \leq \inf_u \{ \langle L_T, u \rangle + B_T(u) \}$$

$$\iff \quad \sum_{t=1}^{T} \langle \ell_t, w_t \rangle \leq -\sup_u \{ \langle u, -L_T \rangle - B_T(u) \}$$

$$\iff \quad \sum_{t=1}^{T} \langle \ell_t, w_t \rangle \leq -B_T^*(-L_T).$$

Thus, the comparator-adaptive regret bound is equivalent to

$$\boxed{\sum_{t=1}^{T} \langle \ell_t, w_t \rangle \leq -B_T^*\left( -\sum_{t=1}^{T} \ell_t \right).}$$

Hence, the natural worst-case comparator in the unconstrained setting is $u \in \partial B_T^*(-\sum_t \ell_t)$, where $B_T^*$ is the Fenchel conjugate of $B_T$. If we for instance assume that the regret bound $B_T$ admits the form $B_T(u) = G\|u\|\sqrt{T \log\left(\|u\|\sqrt{T}/\epsilon + 1\right)}$ as in the unconstrained OLO setting, this translates into a worst-case comparator having norm

$$\|u\| \propto \epsilon \exp\left( \frac{\|\sum_{t=1}^{T} \ell_t\|^2}{G^2 T} \right),$$

see for instance McMahan & Orabona (2014, Section 6.1).

## I. Supporting Lemmas

For completeness, this section collects various well-known lemmas, borrowed results, or otherwise tedius calculations we do not wish to repeat.

The following lemma is standard and included for completeness.

**Lemma I.1.** *Let $(\alpha_t)_t$ be an arbitrary sequence of non-negative numbers and let $p \geq 1$. Then*

$$\sum_{t=1}^{T} \frac{\alpha_t}{\left( \sum_{s=1}^{t} \alpha_s \right)^{1-1/p}} \leq p \left( \sum_{t=1}^{T} \alpha_t \right)^{1/p}$$

*Proof.* Let $S_t = \sum_{s=1}^{t} \alpha_s$, and observe that by concavity of $x \mapsto x^{1/p}$ for any $p \geq 1$, we have

$$S_t^{1/p} - S_{t-1}^{1/p} \geq \frac{S_t - S_{t-1}}{p S_t^{1-1/p}} = \frac{\alpha_t}{p S_t^{1-1/p}} = \frac{\alpha_t}{p \left( \sum_{s=1}^{t} \alpha_s \right)^{1-1/p}}$$

Hence summing over $t$ yields

$$\sum_{t=1}^{T} \frac{\alpha_t}{\left(\sum_{s=1}^{t} \alpha_s\right)^{1-1/p}} \leq p \left(\sum_{t=1}^{T} S_t^{1/p} - S_{t-1}^{1/p}\right) = p S_T^{1/p} = p \left(\sum_{t=1}^{T} \alpha_t\right)^{1/p}$$

$\square$

We also use the following standard integral bound

**Lemma I.2.** *Let $(\alpha_t)_t$ be an arbitrary sequence of non-negative numbers. Then*

$$\sum_{t=1}^{T} \frac{\alpha_t}{\alpha_0 + \sum_{s=1}^{t} \alpha_s} \leq \log\left(1 + \frac{\sum_{t=1}^{T} \alpha_t}{\alpha_0}\right).$$

*Proof.* We have via a standard integral bound (see, e.g., Orabona (2019, Lemma 4.13))

$$\sum_{t=1}^{T} \frac{\alpha_t}{\alpha_0 + \sum_{s=1}^{t} \alpha_s} \leq \int_{\alpha_0}^{\alpha_0 + \sum_{t=1}^{T} \alpha_t} \frac{1}{t} dt = \log(x) \Big|_{x=\alpha_0}^{\alpha_0 + \sum_{t=1}^{T} \alpha_t}$$

$$= \log\left(\alpha_0 + \sum_{t=1}^{T} \alpha_t\right) - \log(\alpha_0) = \log\left(1 + \frac{\sum_{t=1}^{T} \alpha_t}{\alpha_0}\right).$$

$\square$

The following tuning lemma follows by observing that expressions of the form $P/\eta + \eta V$ are minimized at $\eta^* = \sqrt{P/V}$, and then applying simple case work to cover the edge cases $\eta^*$ is outside of the range of candidate step-sizes.

**Lemma I.3.** *Let $b > 1$, $0 < \eta_{\min} \leq \eta_{\max}$ and let $\mathcal{S} = \{\eta_i = \eta_{\min} b^i \wedge \eta_{\max} : i = 0, 1, \ldots\}$. Then for any $P, V \in \mathbb{R}_{\geq 0}$, there is an $\eta \in \mathcal{S}$ such that*

$$R(\eta) := \frac{P}{\eta} + \eta V \leq (b+1)\sqrt{PV} + \frac{P}{\eta_{\max}} + \eta_{\min} V$$

We borrow the following concentration result from (Zhang & Cutkosky, 2022).

**Theorem I.4.** *(Zhang & Cutkosky (2022)) Suppose $\{X_t, \mathcal{F}_t\}$ is a $(\sigma_t, b_t)$ sub-exponential martingale difference sequence. Let $\nu$ be an arbitrary constant. Then with probability at least $1 - \delta$, for all $t$ it holds that:*

$$\sum_{i=1}^{t} X_i \leq 2 \sqrt{\sum_{i=1}^{t} \sigma_i^2 \log\left(\frac{4}{\delta}\left[\log\left(\left[\sqrt{\sum_{i=1}^{t} \sigma_i^2/(2\nu^2)}\right]_1\right) + 2\right]^2\right)}$$

$$+ 8 \max\left(\nu, \max_{i \leq t} b_i\right) \log\left(\frac{28}{\delta}\left[\log\left(\frac{\max(\nu, \max_{i \leq t} b_i)}{\nu}\right) + 2\right]^2\right).$$

*where $[x]_1 = \max(1, x)$.*

We also use a mild modification of Zhang & Cutkosky (2022, Lemma 8) which corrects a minor discrepancy in the "units" of the quantities involved.

**Lemma I.5.** *(Adapted from Zhang & Cutkosky (2022, Lemma 8)) Suppose $\mathcal{A}$ is an arbitrary OLO algorithm that guarantees regret*

$$R_T^{\mathcal{A}}(0) = \sum_{t=1}^{T} \langle g_t, w_t \rangle \leq \epsilon G$$

*for $T \geq 1$ and all sequences $(g_t)_{t \in [T]}$ with $\|g_t\| \leq G$. Then it must hold that $\|w_t\| \leq \epsilon 2^{t-1}$ for all $t$.*

*Proof.* We first show that

$$G\|w_t\| \le G\epsilon - R_{t-1}^{\mathcal{A}}(0). \tag{27}$$

Indeed, suppose not; then on an arbitrary sequence of losses $g_1, \ldots, g_{t-1}, \frac{w_t}{\|w_t\|}G$, we would have

$$R_t^{\mathcal{A}}(0) = R_{t-1}^{\mathcal{A}}(0) + G\|w_t\| > G\epsilon,$$

contradicting the assumption $\mathcal{A}$ guarantees that $R_t^{\mathcal{A}}(0) \le G\epsilon$. Now with Equation (27) established, observe that

$$G\|w_t\| \le G\epsilon - R_{t-1}^{\mathcal{A}}(0) = G\epsilon - R_{t-2}^{\mathcal{A}}(0) - \langle g_{t-1}, w_{t-1}\rangle \le G\epsilon - R_{t-2}^{\mathcal{A}}(0) + G\|w_{t-1}\|$$
$$\le 2(G\epsilon - R_{t-2}^{\mathcal{A}}(0)) \le 2^2(G\epsilon - R_{t-3}^{\mathcal{A}}(0)) \le \ldots \le 2^{t-1}G\epsilon$$

hence dividing both sides by G we have $\|w_t\| \le 2^{t-1}\epsilon$ $\qquad\square$

