# OpenReview forum: "A Perturbation Approach to Unconstrained Linear Bandits"
_ICML.cc/2026/Conference — ICML 2026 regular_

### Official Review · Reviewer_s5B6 · 2026-03-05

**Soundness:** 3
**Presentation:** 3
**Significance:** 3
**Originality:** 3
**Overall Recommendation:** 4
**Confidence:** 3

**Summary:**

This paper studies unconstrained bandit linear optimization (uBLO) over $\mathbb{R}^d$ with bandit feedback $\langle \ell_t, w_t\rangle$, aiming for comparator-adaptive static and dynamic regret guarantees.
The authors propose a modular framework PABLO, whichallows any comparator-adaptive OLO algorithm to be used as a plug-in.
The mechanism of PABLO is simple and elegant: at each round,
PABLO perturbs an OLO output $w_t$ to $\tilde w_t=w_t+H_t^{-1/2}s_t$, plays $\tilde{\omega}_t$ and observes feedback $\langle \ell_t, \tilde w_t\rangle$, then computes an unbiased loss estimator $\hat \ell_t$ and feeds it back to the OLO algorithm.
By utilizing the ghost-iterate trick, the authors demonstrate that the expected regret of the PABLO framework can be reduced to the regret of the underlying OLO algorithm plus an additional noise term.

By plugging different OLO routines, the paper provides expected and high-probability guarantees for static/dynamic comparator-adaptive regret in uBLO, which are novel and interesting.
Moreover, for expected regret, the paper explicitly distinguishes between oblivious and data-adaptive comparator regimes, which is often left implicit in prior work.

**Compliance With Llm Reviewing Policy:**

Affirmed.

**Final Justification:**

This paper is technically solid. My concerns have been fully resolved. I keep the positive score.

**Key Questions For Authors:**

- Correct me if I am wrong.  The high-probability dynamic regret result in Theorem 4.3 is for the regime that $\mu_{t}$ is $\mathcal F_t$-measurable. Can this be extended to the data-adaptive comparator regime (analogous to the "otherwise, $\kappa=1$" case in Section 3)?

**Limitations:**

yes

**Strengths And Weaknesses:**

Strengths:

- The paper studies the problem of unconstrained bandit linear optimization, which is timely and interesting in online learning.
- The results are comprehensive and technically deep.
- The proposed PABLO framework is simple and elegant, which effectively reduces the problem of uBLO to OLO.
- The paper provides a perturbation-based estimator, which is unbiased and almost surely bounded, enabling use of OLO routines that assume bounded gradients.
- The paper provides a broad set of theoretical guarantees, including expected and high-probability results for both static and dynamic regret. For example,
	- new "norm-adaptive" results in the expected static/dynamic regret;
	- $\sqrt{P_T}$-adaptive dynamic regret without any prior knowledge of $P_T$;
	- the first high-probability bounds for both static and dynamic regret in the uBLO setting.

Weaknesses:

- The paper uses multiple "adaptive" notions, e.g., comparator-adaptive and norm/data-adaptive comparator norms. Their meanings can be easy to conflate. It would be helpful to explain "data-adaptive comparator norm" clearly in the text in Section 1.2.
- It would be helpful to provide more intuition or proof sketches for the main results.
- Since the paper contains many results, it is dense and may be difficult for readers to follow. It would be helpful to provide a summary table of the main results and their conditions.

---

> ### Author Rebuttal · Authors · 2026-03-30
>
> We thank the reviewer for the positive feedback and the careful review
> of our paper. We will elaborate on the suggested points for the
> camera-ready revision. More precisely, we will introduce a table
> summarizing our results in Appendix and detail the learner-environment
> interaction in each of the settings that we consider in the main paper.
>
> > Correct me if I am wrong. The high-probability dynamic regret result
> > in Theorem 4.3 is for the regime that is -measurable. Can this be
> > extended to the data-adaptive comparator regime (analogous to the
> >\"otherwise, " case in Section 3)?
>
> This is a great question. It is in fact possible, using a similar
> "ghost-iterate" trick that we used in our in-expectation results. It
> is a fairly straight-forward extension of the result, which we will add
> for the camera-ready revision of the paper. Thank you for the insightful
> suggestion!
>
> A rough sketch of the argument is as follows: similar to the
> in-expectation reduction (Proposition 2.3), the basic idea is that if
> $u_t$ is allowed to be fully data-adaptive, then might correlate with
> $g_t-\\hat{g}_t$, so we can't necessarily apply concentration results to control the
> term $-\\sum_t\\langle g_t-\\hat{g}_t,u_t\\rangle$.
> However, we can
> add/subtract the iterates of a "virtual" instance of our base
> algorithm which exists only in the analysis, leading to
> $$\\sum_t \\langle{g_t-\\hat{g}_t, \\tilde{w}_t-u_t}\\rangle + \\sum_t \\langle {g_t-\\hat{g}_t, -\\tilde{w}_t}\\rangle~.$$
>
> The $\\tilde{w}\_t$ in the second summation are $\\mathcal{F}_{t-1}$
> measurable so concentration can control this term using the same
> arguments as Theorem 4.1, while the first term can be bounded using the
> regret guarantee of the (virtual) base algorithm applied to losses
> $w\\mapsto \langle g_t-\\hat{g}_t,w\\rangle$, which also provides the
> crucial negative terms $-\\sum_t\\varphi_t(\\tilde w_t)$ which control the
> $\\|\tilde{w}_t\\|$-dependent terms that will result from the
> concentration argument, similar to the discussion in section 4.

---

> > ### Author Rebuttal · Reviewer_s5B6 · 2026-04-01
> >
> > Thank the authors for the clear response and the proof sketch addressing my question on extending to the data-adaptive comparator regime. My concerns have been addressed. I am inclined to raise my score.

---

### Official Review · Reviewer_4Rio · 2026-03-13

**Soundness:** 3
**Presentation:** 2
**Significance:** 3
**Originality:** 3
**Overall Recommendation:** 4
**Confidence:** 2

**Summary:**

This paper considers bandit linear optimization with unconstrained action sets (uBLO). The authors applied the standard perturbation-based approach to this setting and proposed PABLO. The authors derived the first high-probability regret bounds for static and dynamic uBLO settings, while the static regret bound matches the best result in bounded domain setting. Additionally, the authors established intermediate results on the lower regret bounds on static uBLO.

**Compliance With Llm Reviewing Policy:**

Affirmed.

**Final Justification:**

Most of my concerns are addressed. Therefore, I keep my positive score.

**Key Questions For Authors:**

**1.** How does the computational cost of the proposed algorithm compare with other algorithms in similar settings?

**Limitations:**

The limitations in this work are the same as listed in "Key Questions For Authors", addressing or clarifying these questions could strengthen this paper significantly.

**Strengths And Weaknesses:**

**Soundness**
*Strengths:* The claims in this paper are well supported by rigorous proofs.

**Presentation**
*Strengths:* The structure of the paper is generally clear, the algorithms are well explained.
*Weaknesses:* The problem setup is not formally defined, please add formal definitions of the setup and assumptions before introducing the algorithm.

**Significance and Originality**
This paper applied perturbation approach to the uBLO setting, discovering that we can reduce BLO to an OLO problem. The authors proposed novel algorithms and proved high-probability regret bounds. Additionally, this paper established progress in the lower bounds in the uBLO setting.
Overall, this paper is novel and significantly contributes to this field, while leaves several open questions.

---

> ### Author Rebuttal · Authors · 2026-03-30
>
> We thank the reviewer for their time and feedback.
>
> ### **Presentation**
>
> Following your remark and point 3. of reviewer XN7Y, we will clarify the
> learner-environment protocol in all settings considered in
> a paragraph at the beginning of Section 2.
>
> ### **Questions**
>
> > 1\. How does the computational cost of the proposed algorithm compare
> > with other algorithms in similar settings?
>
> Our approach is a reduction to OCO, so the complexity depends on both
> the cost of the reduction and the cost of the OCO algorithm applied.
>
> The complexity of the reduction depends on the choice of the
> perturbation matrix $H_t$. In all of our results, we choose an isotropic
> setting $H_t\propto \lambda_t I_d,$ in which case the perturbation is
> equivalent to randomly sampling a standard basis vector
> $v_t\in\{e_1,\ldots,e_d\}$ and setting
> $\tilde w_t =  w_t + \lambda_t v_t$, which amounts to an $O(d)$
> operation. Then, since our approach is a reduction to OCO, the remaining
> computation will match that of whatever OCO algorithm is chosen. Our
> static regret guarantee therefore takes $O(d)$ per-round, which is
> actually significantly better than the $O(d^2)$ that would typically be
> used by linear bandit algorithms employing a loss estimator based on a
> least-squares estimate.
>
> The closest comparison for our dynamic regret results would be
> $P_T$-adaptive results for *stochastic* linear bandits, which require
> the same $O(d\log(T))$ per-round computation as our approach. This is
> essentially the cost of not knowing $P_T$. The same computational
> bottleneck also exists in the OCO setting as well.

---

> > ### Author Rebuttal · Reviewer_4Rio · 2026-04-04
> >
> > My concerns are properly resolved in detail. Overall, this paper is technically solid and novel. Solving the open questions left in this work could significantly strengthen this paper. Therefore, I keep my original score.

---

### Official Review · Reviewer_jNjy · 2026-03-15

**Soundness:** 3
**Presentation:** 2
**Significance:** 3
**Originality:** 3
**Overall Recommendation:** 4
**Confidence:** 4

**Summary:**

The paper revisits the classical perturbation-based approach of Abernethy et al. (2008) — SCRiBLe — and reframes it as a modular reduction called PABLO (Perturbation Approach for Bandit Linear Optimization) for the unconstrained Bandit Linear Optimization (uBLO) setting.

The insight is that in the unconstrained domain, this perturbation scheme effectively reduces BLO to a standard Online Linear Optimization (OLO) problem, which enables the authors to plug in modern comparator-adaptive OLO algorithms as subroutines. The authors analyze the distinction between oblivious and data-adaptive comparator norms, which turns out to induce a \sqrt{d} separation in dimension dependence. The paper obtains expected-regret bounds for both static and dynamic regret, the first high-probability guarantees for uBLO, a proof of the standard lower bound on the unit ball, and a partial conjecture on the minimax rate for the full unconstrained setting.

**Compliance With Llm Reviewing Policy:**

Affirmed.

**Key Questions For Authors:**

See Weaknesses.

**Limitations:**

yes

**Strengths And Weaknesses:**

Strengths:
1) Modularity and Conceptual Clarity. The PABLO framework is elegant. By decoupling the OLO update from the perturbation mechanism the paper enables a clean plug-and-play interface with any OLO algorithm.

2) Novel Distinction on Comparator Norm Adaptivity:  The observation that a data-adaptive comparator norm (chosen after observing the trajectory) forces a \sqrt{d} degradation in guarantees  is a subtle and important point.

3) Dynamic Regret Result. Theorem 3.2 achieves the first \sqrt{P_T}-adaptive dynamic regret bound in uBLO without prior knowledge of P_T. This strictly improves on Rumi et al. (2025), who only achieved switching adaptivity (switching number), which is a weaker measure failing to account for increment magnitudes.

4) High-Probability Bounds. Theorems 4.2 and 4.3 are the first high-probability guarantees for uBLO, matching the best known rates from constrained settings.

5) Lower Bound. Theorem 5.2 provides a clean, self-contained proof of the folklore \sqrt{dT} minimax lower bound on the unit Euclidean ball, which was previously unproven in the literature. The key insight that the improved \sqrt{d} scaling arises from a tighter noise-variance constraint is clearly explained and of independent interest.


Weaknesses
1) Incomplete Lower Bound Theory: The lower bound contribution is partial. Conjecture 5.3, the minimax static regret for uBLO  when the comparator norm is oblivious — is left unproven, and the authors explicitly acknowledge that combining the scale and direction lower bounds seems to require fundamentally new techniques.


2) Norm-Adaptive Gap Unexplained. While the paper identifies a \sqrt{d} gap between the oblivious and norm-adaptive regimes, it does not resolve whether this gap is fundamental or an artifact of the current analysis. The discussion in Section 5 on this point is inconclusive.

3) Novelty and Heavy Reliance on  Zhang & Cutkosky (2022) : The high-probability guarantees in Section 4 rely very heavily on Zhang & Cutkosky (2022) — particularly the Huber-like composite penalty, optimistic updates, and concentration tools. While composing these correctly is non-trivial, the novelty of Section 4 relative to that prior work is somewhat limited to the reduction step in Proposition 4.1 and careful bookkeeping.

4) Presentation: The paper is technically dense, and the main text occasionally sacrifices intuition for brevity (ex: significance of $\kappa$ in Theorems 3.1 and 3.2). A dedicated discussion or table summarising the rate landscape across adversarial regimes would substantially improve readability.

5) Experimental Validation Absent. The paper is entirely theoretical, which is standard for this subfield. However, given that the key claim — the \sqrt{d} gap between oblivious and adaptive comparator norm regimes — is a subtle one that may affect practitioners, even a synthetic simulation demonstrating the gap would strengthen the paper's practical relevance.

---

> ### Author Rebuttal · Authors · 2026-03-30
>
> Thank you for carefully assessing our submission! Below we address some of the main questions and weaknesses discussed in the review
>
>
> > 3\. Novelty and Heavy Reliance on Zhang & Cutkosky (2022) : The
> > high-probability guarantees in Section 4 rely very heavily on Zhang &
> > Cutkosky (2022) --- particularly the Huber-like composite penalty,
> > optimistic updates, and concentration tools.
>
> We emphasize that our high-probability dynamic regret result is a
> significant and non-trivial extension of the existing black-box optimism
> approach: existing optimistic reductions (Cutkosky, 2019; Zhang &
> Cutkosky, 2022) *can not obtain the per-comparator adaptivity that we
> achieve* in our high-probability dynamic regret guarantees. In
> particular, our result required designing a novel variant of the
> reduction which preserves the multiplicative $\\|g_t\\|^2\\|u_t\\|$
> dependencies, while still ensuring that certain delicate cancellations
> occur. This is non-trivial and is the first time these per-comparator
> adaptive guarantees have been obtained in the context of a black-box
> optimistic reduction similar to Cutkosky (2019)'s original reduction,
> and is likely to be of independent interest. We discuss the novelty of
> the dynamic regret result in the main text, but will revise the
> discussion to more strongly highlight the novelty and significance of
> our new result.
>
> We would also like to gently push back on the phrasing that we *heavily
> rely* on the results of Zhang & Cutkosky (2022). While it is true that
> we borrow the huber-like penalty and some analysis techniques from Zhang
> & Cutkosky (2022) (though we generalize their arguments to dynamic
> comparator sequences), it should be noted that the concentration tools
> and optimistic arguments used are fairly standard in the literature. We
> respectfully point out that many significant contributions in the
> community consist in such non-trivial use of standard tools.
>
>
> > 3\. (continued) While composing these correctly is non-trivial, the
> > novelty of Section 4 relative to that prior work is somewhat limited
> > to the reduction step in Proposition 4.1
>
> We would also like to emphasize that this is exactly the essential
> insight of our paper---the fact that it is possible to reduce so
> directly to OCO via Propositions 2.3 and 4.1 is what allows us to
> immediately apply existing results for OCO to achieve several
> non-trivial results that were previously out of reach. We see this as
> one of the strengths of this work, rather a weakness. Please see also
> our reply to Reviewer XN7Y [here](https://openreview.net/forum?id=XSpBSHzJAg&noteId=Vhe7xjY1U3) for related remarks.
>
> > 2\. and 5. Norm-Adaptive Gap Unexplained. While the paper identifies a
> > $\sqrt{d}$ gap between the oblivious and norm-adaptive regimes, it
> > does not resolve whether this gap is fundamental or an artifact of the
> > current analysis. The discussion in Section 5 on this point is
> > inconclusive.
>
> One contribution of the paper is to identify and formalize the
> distinction between the norm-oblivious and norm-adaptive regimes, which
> is motivated by the fact that the adversary's optimal norm is naturally
> trajectory-dependent in unconstrained settings (see Appendix G).
>
> The key difficulty is that the guarantees in the two regimes are not
> directly comparable. In the norm-oblivious setting, the bound holds
> uniformly for every *fixed* comparator norm $\\|u\\|$ chosen before the
> interaction. In the norm-adaptive setting, by contrast, $\\|u\\|$ may
> depend on the realized trajectory, and the guarantee must hold against
> every *policy* that selects the norm after observing that trajectory.
> Thus, the latter benchmark class is strictly richer, and a numerical gap
> between the two bounds cannot be interpreted directly as a like-for-like
> comparison. Accordingly, the $\sqrt{d}$ factor should be viewed as
> evidence that post-hoc norm adaptivity **may** incur an additional
> dimension-dependent price, rather than as a definitive characterization
> of the optimal rate. Determining whether this separation is intrinsic,
> or whether sharper analysis can close it, is an interesting open
> question.
>
> Finally, we believe that it is already interesting that the
> gap between the norm-oblivious and norm-adaptive regime is *only* of
> order $\sqrt{d}$, and achieved *without any change in the algorithms*. In
> the paper (l. 242 left column), we discuss that OSMD requires a
> different tuning in the two regimes and its regret degrades to $T^{2/3}$
> in the norm-adaptive one, and improving this result (if possible) would
> require significant refinements to the original analysis.
>
>
>
> > 4\. Presentation:  A dedicated discussion or table summarising the
> > rate landscape across adversarial regimes would substantially improve
> > readability.
>
> Thank you for the suggestion, we will include this table in appendix and
> refer to it in Section 1.2 (contributions) to enhance clarity.

---

> > ### Author Rebuttal · Reviewer_jNjy · 2026-04-04
> >
> > I am keeping my score. I thank the authors for answering some of my concerns partially. However, some of my concerns were not addressed as well.

---

> > > ### Author Response · Authors · 2026-04-04
> > >
> > > We thank the reviewer for responding and for their effort in carefully assessing our response so far. We would very much appreciate it if the reviewer could elaborate on what they felt we did not address adequately, before the discussion period ends. From what we can tell, we directly addressed each of the points in detail, besides the point on experimental validation, which we thought had been agreed was not essential for a theory-focused paper

---

### Official Review · Reviewer_XN7Y · 2026-03-18

**Soundness:** 3
**Presentation:** 2
**Significance:** 3
**Originality:** 2
**Overall Recommendation:** 4
**Confidence:** 4

**Summary:**

This paper proposes a new algorithm PABLO, which is able to convert any algorithm for Online Linear Optimization(OLO) into an algorithm for unconstrained Bandit Linear Optimization(uBLO). Given the performance guarantee of an algorithm for OLO, PABLO is able to derive a corresponding result for uBLO. By choosing different OLO subroutines, they prove multiple upper bounds of different regret definitions, including static and dynamic regret. In addition, this paper also discusses the impact of oblivious comparator scale. At the end of the paper, they review possible ideas to prove lower bound and left them as open problem.

**Compliance With Llm Reviewing Policy:**

Affirmed.

**Final Justification:**

Rebuttal from the reviewer fully addressed my concern regarding the possibility of achieving both optimality on high probability bound and expected bound. I believe a stronger lower bound should be included for a spotlight level, but current presentation is good enough for acceptance. I will keep my score unchanged(4 - weak accept) with higher confidence (2 to 4).

**Key Questions For Authors:**

1. Is it possible to adopt a single routine algorithm, such that PABLO is able to derive both upper bounds for expected regret and high probability regret? If not, what is the main challenge?

2. This might be a dumb question, but I cannot follow your logic in the right half of page 8, line 395 to line 414. It seems you claim that if restricting choosing $u$ that is dependent to the algorithm design and realized trajectory with $\mathbb{E}\|u\|=M$, we can derive lower bound $MT$. Do you mean the lower bound $MT$ holds for all algorithms? Or do you mean the lower bound $MT$ only works for the algorithm that will adopt uniform exploration with probability at least $\gamma$ at each round? If one of these two conclusions are indeed what you try to deliver, I think you can rewrite it as a Theorem or a Lemma.

   Anyway, I think more explanation will be better, since this part is trying to present unpromising ideas to the followers. If the page limit makes the extra description unavailable, you can put it in the appendix.

3. I think it will be better if the paper can rigorously clarify the dynamics of uBLO. For example, providing pseudocode presenting the interaction between the agent and the environment. Then the reader can understand the difference between "static" "dynamic" "$\|u\_t\|$ is $\mathcal{F}\_t$ measurable"

4. In Proposition 2.1, do you require $H_t$ to be symmetric? This is not a huge issue, as we suffice to figure out one choice for $H_t$

**Limitations:**

I doubt whether there is a real-world motivation in the research, but I agree theoertical analysis is the most important. I don't find other limitations. Please check the questions subsection.

**Strengths And Weaknesses:**

**Strengths:** This paper provides a general algorithm PABLO, which is able to adapt multiple algorithms for a new problem formulation. What's more, PABLO is easy to implement and sufficient to derive multiple upper bounds for different definition of regret. This paper analyze the impact of comparator scale carefully, which complements the historical literature.

**Weaknesses:**

1. The algorithm design appears to be a direct adaptation of existing literature, making me worried about the algorithmic novelty.

2. As discussed in section 5.1, the lower bound of uBLO remains a mystery. It is still unclear how to prove conjecture 5.3 and the current lower bounds Theorem 5.1 and 5.2 don't fully utilize the difficulty in the uBLO setting.



Given the above weaknesses, I still believe this paper provides contribute some novel conclusions. I admit I am not an expert in uBLO and I will follow the comments from other reviewers who are familiar with the development on Online Linear Optimization and Bandit Linear Optimization.

---

> ### Author Rebuttal · Authors · 2026-03-30
>
> We thank the reviewer for the detailed review, and for their insightful questions and suggestions.
>
> ## **Response to the two weaknesses**
>
> ### **Algorithmic novelty**
>
> We would like to point out that algorithmic novelty is a somewhat
> misleading benchmark when assessing the significance of our results; one
> of the key insights of this work, and a consistent theme throughout the
> paper, is that the simple perturbation-based approach to BLO effectively
> allows us to reduce to an OCO problem, wherein existing algorithms can
> be applied. In this sense, one of the key contributions of our work is
> precisely that our approach allows us to achieve new and non-trivial
> guarantees by leveraging existing results from the OCO literature. This
> is an important insight which has allowed us to achieve several novel
> and important results using a relatively simple approach.
>
> Moreover, we point out that our high-probability dynamic regret result
> does in fact introduce algorithmic novelty. Indeed, this result required
> developing a new optimistic reduction that preserves the
> $\\|g_t\\|^2\\|u_t\\|$ dependencies, which is not possible in prior works
> such as Cutkosky (2019) and Zhang & Cutkosky (2022). See our reply to
> Reviewer jNjy [here](https://openreview.net/forum?id=XSpBSHzJAg&noteId=3ZjjNGi5EY) for more details.
>
> ### **Lower bound**
>
> We agree that proving Conjecture 5.3 remains out of reach, and that our
> current lower bounds do not yet fully characterize the difficulty of
> uBLO, and we are upfront about this limitation in the main text. That
> said, the purpose of Section 5 is precisely to make progress toward this
> picture by identifying concrete lower-bound mechanisms and gathering
> evidence for the conjectured rates. We believe this is already valuable:
> despite extensive work on lower bounds for constrained linear bandits,
> the unit-ball case remained open despite being highlighted in prior work
> \[1,2\]. From this perspective, Theorem 5.1 is already a non-trivial
> result and provides a useful milestone toward the harder unconstrained
> case.
>
> \[1\] \"On the Complexity of Bandit Linear Optimization.\" Shamir, 2012
>
> \[2\] \"Towards minimax policies for online linear optimization with
> bandit feedback.\" Bubeck, Cesa-Bianchi, and Kakade, 2012.
>
> ## **Questions**
>
> > 1\. Is it possible to adopt a single routine algorithm, such that
> > PABLO is able to derive both upper bounds for expected regret and high
> > probability regret? If not, what is the main challenge?
>
> Classically, one can convert an upper bound $B_T(\delta)$ valid with
> probability at least $1-\delta$ into an in-expectation bound by upper
> bounding the expected regret by
> $B_T(\delta) + \delta \cdot \overline B_T$, where $\overline B_T$ is an
> almost-sure upper bound on the regret. For instance, in the static
> regret setting we can use $\overline B_T = \\|u\\|T+\epsilon$, thanks
> to the origin-regret guarantee $R_T(0) \leq \epsilon$. Hence, setting
> e.g. $\delta=1/T$ gives the same bound $B_T(\delta)$ (up to a
> lower-order term) both in expectation and with probability $1-1/T$.
>
> However, as is typically the case, our high-probability bounds admit
> several additional penalties that could otherwise be avoided if one only
> cares about expected regret, since the high-probability bound requires
> *additionally* controlling certain $\\|w_t\\|$-dependent terms that appear
> after applying concentration arguments, as discussed in Section 4.
>
> > 2\. ...Do you mean the lower bound $MT$ holds for all algorithms? ...
>
> The lower bound $MT$ discussed in this part of the text applies to any
> algorithm satisfying a minimum per-round exploration condition $\gamma$
> (that is, algorithms that mix uniformly with probability at least
> $\gamma$ at each round). Many standard adversarial bandit methods,
> including OSMD-style algorithms (Bubeck et al., 2012) fall into this
> class, but the discussion here is *not* claimed to to hold for all
> algorithms.
>
> The purpose of this paragraph is not to present a general lower bound
> for all algorithms, but rather to *highlight a limitation of
> norm-adaptive formulations based on $\mathbb{E}[\\|u\\|]$*. In
> particular, an adversary can choose a comparator with very large norm on
> rare but highly unfavorable trajectories, which can make such guarantees
> misleading even when those trajectories have very small probability.
>
> In the revision, we will clarify this paragraph and encapsulate the main
> argument in a proposition.
>
> > 3\. I think it will be better if the paper can rigorously clarify the
> > dynamics of uBLO....
>
> Thank you for this suggestion, we will introduce a paragraph at the
> beginning of Section 2 to make the learner-environment interaction
> protocol clearer in each of the settings we consider.
>
> > 4\. In Proposition 2.1, do you require $H_t$ to be symmetric? This is
> > not a huge issue, as we suffice to figure out one choice for $H_t$.
>
> Proposition 2.1 assumes $H_t\in\mathbb{R}^{d\times d}$ is positive
> definite, which implies symmetric for real matrices

---

> > ### Author Rebuttal · Reviewer_XN7Y · 2026-04-02
> >
> > Thanks for your clarification. I admit improving lower bounds can be hard and I believe the current result is sufficient to get published. I will keep my score unchanged.
> >
> > I want to clarify my questions.
> >
> > 1. Regarding both high probability upper bound and expected upper bound, I am asking whether we can adapt an oracle such that it can achieve all the Theorems in section 3 and 4. I know we can convert a high probability upper bound to expected upper bound, but as you mentioned, this might be loose.
> >
> > 2. Regarding positive definite matrix, I doubt your definition is "We call $H$ is positive definite, if for any non-zero vector $x$, we have $x^THx > 0$". If so, a positive definite matrix might not be symmetric. For example, $H=[\begin{matrix}1 \& 1\\\\ -1 \& 1\end{matrix}]$, for real value $x,y$, we have $[x, y]H[\begin{matrix}x\\\\ y\end{matrix}]=x^2+y^2>0$, as long as one of $x,y$ is non-zero.
> >
> > Anyway, I think the current rebuttal is acceptable.

---

> > > ### Author Response · Authors · 2026-04-02
> > >
> > > > Regarding both high probability upper bound and expected upper bound...
> > >
> > >
> > > Thanks for further elaborating this point; we are unsure if it is possible to achieve the exact bounds in sections 3 and 4 using a single algorithm. As mentioned briefly in our earlier reply, the key issue is that the high-probability bounds require controlling additional terms that result from applying concentration inequalities to control the bias terms: the terms $\sum_t\langle g_t-\hat g_t,w_t\rangle$ end up bounded by $\tilde O(\sqrt{\sum_t\\|w_t\\|^2\log(1/\delta)})$ in the high probability bounds, which could be arbitrarily large in unbounded domains. To control this additional term requires adding an additional penalty $\varphi_t(\cdot)$ to the update to help cancel these $\\|w_t\\|$ dependencies out, but this comes at a cost of an additional penalty $\sum_t \varphi_t(u_t)$ appearing in the bound, which would not otherwise need to be there if we only wanted an in-expectation bound. Our intuition is that it is unlikely that a single algorithm can obtain the exact guarantees in both sections 3 and 4 in the unconstrained setting.
> > >
> > > To make this concrete, applying the algorithm from Theorem 4.2 in the context of Theorem 3.1 would result in a regret bound that scales with $\tilde O\left(\frac{d}{\kappa}\\|u\\|\sqrt{V_T}+\\|u\\|\sqrt{dT\log(T/\delta)}\right)$, and we know from Theorem 3.1 that this latter term does not need to appear if you only care about the in-expectation result.
> > >
> > > Note that these issues do not arise in bounded domains since the concentration term can instead be bounded as $\tilde O(D\sqrt{T\log(1/\delta)}$), so it's not necessary to add any additional  regularization to get the high-probability bound versus the in-expectation bound. Thus, in these settings it can be possible to use the same algorithm for both in-expectation and high-probability bounds.
> > >
> > > We agree that this is an interesting observation and will add additional discussion surrounding these points for the revision, thank you your insights!
> > >
> > > > Regarding positive definite matrix...
> > >
> > >
> > > We apologize for the confusing reply; real matrices are often defined to be positive definite if they are symmetric and satisfy $x^\top Ax>0$ (e.g., Definition 7.62 of Axler 2024).  You are correct to point out that not all texts treat symmetric as part of the definition, so it is confusing for us to simply say "positive definite" without specifying which definition is being used.
> > >
> > > We will define positive definite as meaning $x^\top Ax>0$ in the notation section, and specify "symmetric positive definite" in proposition 2.1 to improve the clarity here. Thank you for catching this subtlety!

---

### Official Review · Reviewer_3r5j · 2026-03-21

**Soundness:** 3
**Presentation:** 3
**Significance:** 3
**Originality:** 3
**Overall Recommendation:** 4
**Confidence:** 4

**Summary:**

This paper is on the unconstrained adversarial bandit linear optmization problem, where the learner may take any action in $\mathbb{R}^d$. The focus is on achieving both static dynamic regret bounds relative to unbounded comparators (and comparator sequences) that adapt to the scale of the comparators. The approach taken is reduction based, whereby the authors construct a generic reduction to OLO by randomising the action for the sake of approximating first-order information, as is classical at this point. The key innovation is that rather than coupling the perturbations to the underlying OLO method, the proposed method, PABLO, simply perturbs the actions isotropically, at a scale inversely proportional to the norm of the action proposed by the underlying OLO method. To my understanding, this aspect of the design is where the unconstrained nature of the problem being studied is key, in that in a constrained problem, the learner would instead need to ensure that the perturbed action is feasible under such flat perturbations, which is unnecessary here. Of course, the underlying OLO methods also rely on the lack of constraints. The main technical advantage of this change is very concrete bounds on the sizes of the loss estimates $\\|tl\\|_t$, both surely and in expectation given the history.

$\newcommand{\tl}{\tilde\ell}$

The first set of results uses the conditional expectation bounds to drive certain parameter-free algorithms of Jacobsen and Cutkosky to derive static regret bounds (of the scale $f\_d \\|u\\| \tilde{O}(\sqrt{V\_T}) )$ and dynamic regret bounds in terms of the comparator path length. It should be noted that the underlying OLO methods for these settings are distinct. There are two sources of overhead due to the bandit feedback in these bounds: first in the dynamic bounds, an extra addition $d P\_T$ term is accrued, and second, in both cases, the main square-root terms has a premultiplying $f\_d$ factor. This factor shows a certain subtlety, in that if the comparators are allowed to be picked acausally (i.e., $u$ depends on the realizations of the trajectory, or in the dynamic case $u_t$ can depend on realisations after time $t$), then $f\_d = d$, and otherwise $f\_d = \sqrt{d}$. This distinction is discussed carefully, and the fact that $O(\sqrt{T})$ bounds can be retained even against such nonoblivious comparators is highlighted.

The second set of results focuses on high-probability guarantees, which rely on the sure bounds on $\\|\tl\\|\_t$. The overall approach is similar, and based on a high-probability reduction to OLO, although it should be noted that this reduction is particular to the specific algorithm proposed. From this static regret bounds follow using an OLO procedure of Zhang and Cutkosky. For the dynamic regret bounds, the authors instead extend the procedure of Jacobsen and Cutkosky using the techniques of Zhang and Cutkosky, and drive this updated algorithm to obtain dynamic regret bounds for the causal case. These are qualitatively similar to the static bounds, modulo and additive $\sqrt{d P\_T \log(T/\delta)}$ term and various extra log factors, although it should be noted that even with these causal $u\_t$s in the worst case the main term of the dynamic bound is a $\sqrt{d}$ factor larger than in the expected regret case.

Finally, the authors consider lower bounds on the uBLO problem. Towards this direction, a lower bound for a related problem of direction-learning is shown, which argues that if the actions and comparator are restricted to have unit norm, then a $\sqrt{dT}$ regret is necessary in the worst case (at least for large $T$). This is used to justify a wider lower bound conjecture on the static regret, and various subtleties regarding both this and teasing out the difference between adaptive and oblivious choices of $\\|u\\|$ are discussed.

**Compliance With Llm Reviewing Policy:**

Affirmed.

**Key Questions For Authors:**

-

**Limitations:**

Yes

**Strengths And Weaknesses:**

I quite like this paper. The uBLO problem is certainly of nontrivial interest, and the paper provides both significant new results in the dynamic setting, and also captures a subtle dependence issue in the prior results (and addresses it), which I appreciate. I think this would certainly be of interest to the online learning community at ICML. The algorithmic structure of PABLO is almost classical, but it strongly exploits the unconstrained setting of the problem to yield significant advantages in terms of results relative to the literature. Of course, much of the analysis approach is known, but it still needs nontrivial work to execute properly (esp. Thm 4.3). I found the paper to be mostly well written, with clear contextualisation and explanation of approaches, although I found the lower bounds to be confusingly stated. I did not read the proof of Thm. 4.3 completely, but it appears to be right. The rest of the proofs I did read, and found to be correct barring occasional typos.

The main weakness, in my opinion, is the specialisation of the problem at (i.e., the unconstrained setting). This limits the broader scope for impact for the techniques and results of the paper. Perhaps the only significant technical weakness is that the lower bounds for uBLO remain out of reach with the techniques of the paper, but it is clear that the intention of section 5 is more to gather direct evidence towards what these should look like rather than address them completely, and certainly the algorithm and analysis are interesting.

Overall, I think this is a nice paper, albeit a bit niche, and with techniques that are mainly refinements of existing ones. This makes it hard to justify a clear accept recommendation for me, since it is unclear to what degree the impact will extend beyond uBLO. This drives my current recommendation of weak accept. However, I should mention that I do not publish in this area, and so potentially don't have my finger completely on its pulse. I would thus rely more on other reviewers to judge this, and if deemed sufficient I think this paper could also fall into a clear 'accept' category.

----

Minor points:

- The statement of Prop 2.3 is a bit confusing, because $R\_T$ is defined with respect to the losses $\ell$, whereas the condition you need on $R^{\mathcal{A}}\_T$ is low-regret relative to the sequence of losses $\{g\_t\}$. It would be clearer if you defined a generic regret relative to a loss sequence, and then mentioned that the regret we care about is relative to the losses $\{\ell\_t\}$.

- The statement of Thm 5.2 is very confusing: 1) is $C\_{d,T}$ is lower bounded by either something or something else is unclear. From reading the proof, it is evident that the minimum of these two is shown, and you should just say that. 2) In part , up to a factor is unclear. Do you mean $C\_{d,T} f $ or $C\_{d,T} /f ,$ where $f$ is the factor?

- I found the paragraph on line 396 col 1 very unclear. When you talk about factors and lower bounds here, what are you referring to? On the regret? That shouldn't be the case, because line 416 onwards is explicitly discussing that the two results don't directly combine (i.e., the positive correlation of scale and direction regret doesn't mean that the scale regret is even positive, as far as I understand).

- Some miscellaneous typos/recommended edits
    - In Prop 2.3, $B\_T$ should be a map from $(\mathbb{R}^d)^{2T}$ to the positive reals, rather than from $\mathbb{R}^{2T}$.
    - In line 645, writing $\hat{w} - \hat{w}$ would be clearer than $\pm \hat{w}$ (which I initially read as $\hat{w}$ but with a free choice of sign).
    - Lemma D.3 there's a typo in the definition of $\varphi\_t$ on line 1091
    - Should be $\sum r\_t$ instead of $\sum r\_t^{\eta}$ on line 1112
    - Also a factor od $p_2$ has disappeared in going from the above line to 1115/6. So there should probably be an additive $\log^2 T$ in the statement of Lemma D.3 instead of $\log T$.

---

> ### Author Rebuttal · Authors · 2026-03-30
>
> We thank the reviewer for their detailed review, positive assessment, and constructive feedback. Below, we address their questions, with particular emphasis on clarifying the lower-bound part.
>
> ### **Specialization of the problem**
>
> While uBLO is a specialized setting, we believe it captures
> broader issues of interest in online learning, including understanding
> unbounded decision sets, extending comparator-adaptive guarantees to
> partial feedback settings, and modular bandit reductions. Our paper
> provides several foundational and important results for this setting, a
> point also recognized by other reviewers, and we believe this makes it a
> solid conference contribution.
>
> ### **Clarifying some statements/typos (Minor points 1/2/4)**
>
> Thank you for your suggestions, we will clarify the statements of Prop.
> 2.3 and Theorem 5.2 in the revision. Thank you also for catching some
> typos.
>
> Regarding the theorem, for the first part $C_{d,T}$ is indeed the
> minimum between the two quantities introduced. For the second part, the
> lower bound becomes $\alpha \times \sqrt{\frac{d}{d\vee \log(T)}} \times C_{d,T}$ for some absolute
> constant $\alpha$.
>
> ### **Lower bound (Minor point 3)**
>
> > I found the paragraph on line 396 col 1 very unclear. When you talk
> > about factors and lower bounds here, what are you referring to? On the
> > regret? That shouldn't be the case, because line 416 onwards is
> > explicitly discussing that the two results don't directly combine
> > (i.e., the positive correlation of scale and direction regret doesn't
> > mean that the scale regret is even positive, as far as I understand).
>
> In Eq. (6) we introduce a standard regret decomposition for
> unconstrained online learning, as the sum of a *scale* regret and a
> *direction* regret, and use it to motivate the conjectured lower bound.
> In this paragraph we explain how Theorem 5.2, which focuses on a setting
> where actions belong to the unit Euclidean ball, **might** explain
> the $\sqrt{dT}$ scaling of the upper bounds (obtained on the *full* regret) in the
> unconstrained setting, because it is a valid lower bound on the
> *direction* component of the regret in Eq. (6).
>
> In the rest of the section, we indeed detail that this doesn't
> generalize to the full regret since the *scale* regret could potentially
> be negative. Therefore, we present our results in Section 5 as arguments
> that motivate Conjecture 5.3, which we explicitly state in the paper. We
> suggest that future lower bound proofs should involve constructions that
> explicitly prevent algorithms satisfying the origin-regret condition to
> make large bets, which doesn't seem possible with existing techniques.

---

> > ### Author Rebuttal · Reviewer_3r5j · 2026-04-05
> >
> > My main concern was with broader impact. The rebuttal asserts that the techniques presented could be pertinent to other unconstrained sequential decision making models, and shed broader light on modular bandit reductions. This addresses my concern, but not completely, in that it is natural to then ask the broader pertinence of, say, unconstrained partial monitoring et c., and further to inquire how addition of constraints would modify the particular design approach of the paper.
> >
> > Nevertheless, I think this is not something that needs to be addressed in a counter-rebuttal, in that I doubt that trying to communicate the nuances of this position in short replies is likely to be fruitful. I continue to see the submission positively, and I am willing to take a broader cue on this point from the AC and other reviewers.

---

### Decision · Program_Chairs · 2026-04-30

**Decision:**

Accept (regular)

**Comment:**

The submission provides a novel connection from unconstrained BLO (uBLO) to OLO, via a rather novel perturbation procedure.  The authors proposed the meta algorithm PABLO, which solves uBLO given an OLO algorithm as input. PABLO is shown to achieve a variety of SOTA regret bounds in data-adaptive and high-probability settings. The most valuable point is that the design and analysis of PABLO provide a clean and elegant connection from uBLO to OLO. In agreement with the reviewers, my opinion is that such connection will benefit the online learning community and inspire interesting developments. Thus I recommend acceptance.